# A Single-Loop First-Order Algorithm for Linearly Constrained Bilevel Optimization

**Wei Shen**
University of Virginia
zyy5hb@virginia.edu

**Jiawei Zhang**
University of Wisconsin-Madison
jzhang2924@wisc.edu

**Minhui Huang**
Meta
mhhuang@meta.com

**Cong Shen**
University of Virginia
cong@virginia.edu

## Abstract

We study bilevel optimization problems where the lower-level problems are strongly convex and have coupled linear constraints. To overcome the potential non-smoothness of the hyper-objective and the computational challenges associated with the Hessian matrix, we utilize penalty and augmented Lagrangian methods to reformulate the original problem as a single-level one. Especially, we establish a strong theoretical connection between the reformulated function and the original hyper-objective by characterizing the closeness of their values and derivatives. Based on this reformulation, we propose a single-loop, first-order algorithm for linearly constrained bilevel optimization (SFLCB). We provide rigorous analyses of its non-asymptotic convergence rates, showing an improvement over prior double-loop algorithms – form $O(\epsilon^{-3}\log(\epsilon^{-1}))$ to $O(\epsilon^{-3})$. The experiments corroborate our theoretical findings and demonstrate the practical efficiency of the proposed SFLCB algorithm. Simulation code is provided at https://github.com/ShenGroup/SFLCB.

## 1 Introduction

In recent years, bilevel optimization (BLO) has gained significant popularity for addressing a wide range of modern machine learning problems, such as hyperparameter optimization [36, 7, 33], data hypercleaning [38], meta learning [37, 13], reinforcement learning [41, 11] and neural architecture search [26, 23]; see survey papers [52, 27, 40] for additional discussions. While numerous works for unconstrained BLO problems have been proposed [9, 30, 15, 5, 11, 22], studies focusing on constrained BLO problems are relatively limited.

In this paper, we consider the following BLO problem where the lower-level (LL) problem has *coupled constraints*:

$$\min_{x \in \mathcal{X}} \quad \Phi(x) \triangleq f(x, y^*(x)) \tag{1}$$
$$\text{s.t.} \quad y^*(x) \in \arg\min_{y \in \mathcal{Y}(x)} g(x, y).$$

The upper-level (UL) objective function $f : \mathbb{R}^{d_x} \times \mathbb{R}^{d_y} \to \mathbb{R}$ and the lower-level objective function $g : \mathbb{R}^{d_x} \times \mathbb{R}^{d_y} \to \mathbb{R}$ are continuously differentiable. Moreover, we assume that $g(x, y)$ is strongly convex with respect to $y$. The feasible sets are defined as $\mathcal{X} = \mathbb{R}^{d_x}$, $\mathcal{Y}(x) = \{ y \in \mathbb{R}^{d_y} \mid h(x, y) \le 0 \}$ where $h : \mathbb{R}^{d_x} \times \mathbb{R}^{d_y} \to \mathbb{R}^{d_h}$.

39th Conference on Neural Information Processing Systems (NeurIPS 2025).

For this setting, we develop a single loop algorithm for the special case $h(x, y) = Bx + Ay - b$, where $B \in \mathbb{R}^{d_h \times d_x}$ and $A \in \mathbb{R}^{d_h \times d_y}$. This special class of constrained BLO problems covers a wide class of applications, including distributed optimization [48], hyperparameter optimization of constrained learning problems [46] and adversarial training [53] and draw significant attentions [42, 18, 20].

A popular approach for solving unconstrained BLO is implicit gradient descent [9, 15, 14, 4, 19]. For constrained BLO, several studies have extended this approach to accommodate different constraint settings [42, 18, 46, 45]. However, these implicit gradient-based methods in constrained BLO necessitate computing the Hessian matrix of the lower-level problem [42, 18, 46, 45]. The potential computational challenges associated with the Hessian matrix limit their practical applicability for large-scale problems.

Recently, some first-order methods [21, 50, 49, 16, 20] have been proposed for addressing constrained BLO problems. Most of those works considered transforming the original problem (1) into a single-level one and trying to find the stationary points of the reformulated problem. For example, [50, 49] reformulated the original problem into some approximated functions and proposed single-loop algorithms for finding the stationary point of the approximated problem. However, neither [50] nor [49] establishes clear relationships between the stationary points of their approximated problems and the original one.

Works most closely related to ours are those by [21] and [16], both of which reformulated the problem (1) as

$$\min_{x \in \mathcal{X}, y \in \mathcal{Y}(x)} f(x, y) \quad \text{s.t. } g(x, y) - \min_{z \in \mathcal{Y}(x)} g(x, z) \leq 0, \tag{2}$$

and considered optimizing the following function with a penalty parameter $\delta$:

$$\min_{x \in \mathcal{X}} [\Phi_\delta(x) \triangleq \min_{y \in \mathcal{Y}(x)} \max_{z \in \mathcal{Y}(x)} \Phi_\delta(x, y, z)] \tag{3}$$

where $\Phi_\delta(x, y, z) = f(x, y) + \frac{1}{\delta}[g(x, y) - g(x, z)]$. Based on this reformulation, [21] proposed algorithms for solving BLO with LL constraints $y \in \mathcal{Y}$ and [16] proposed algorithms for solving coupled LL constraints $\mathcal{Y}(x) = \{y \in \mathbb{R}^{d_y} | h(x, y) \leq 0\}$. However, [21] only considered the LL constraints $\mathcal{Y}$ that are independent of $x$ and their methods require projection oracle to $\mathcal{Y}$ at each iteration. Algorithms in [16] require complex double or triple loops, resulting in sub-optimal convergence rates and difficult implementation. Moreover, the connection between the stationary point of the reformulated function $\Phi_\delta$ and the original hyper-objective $\Phi$ is not discussed in [21, 16] for coupled constraints $\mathcal{Y}(x)$.

To address these limitations, in this paper, we establish a rigorous theoretical justification for this reformulation (3) and propose a single-loop Hessian-free algorithm for the linearly constrained cases. Our main contributions can be summarized as follows.

- We establish a rigorous theoretical connection between the reformulated function $\Phi_\delta$ and the original hyper-objective $\Phi$ by proving the closeness of their values and derivatives under coupled constraints $\mathcal{Y}(x) = \{y \in \mathbb{R}^{d_y} | h(x, y) \leq 0\}$ with certain assumptions, which provides strong justifications for the reformulation (3).
- Based on this reformulation and equipped with augmented Lagrangian methods, we proposed SFLCB, a single-loop, first-order algorithm for linearly constrained bilevel optimization problem, and provide rigorous analyses of its non-asymptotic convergence rates, achieving an improvement in the convergence rate from $O(\epsilon^{-3} \log(\epsilon^{-1}))$ to $O(\epsilon^{-3})$ compared to prior works (See Table 1 for a more comprehensive comparison of our work with previous studies). The simple single-loop structure also makes our algorithm easier to implement in practice compared to [16].
- Our experiments on hyperparameter optimization in the support vector machine (SVM) and transportation network design problems validate the practical effectiveness and efficiency of the proposed SFLCB algorithm.

## 2 Related works

**BLO without constraints.** One popular approach for solving unconstrained BLO is to use implicit gradient descent methods [36]. It is well established that when the LL problem is strongly convex

Table 1: Comparison of our paper with [21, 16]. More detailed introductions and discussions of other related works can be found in Section 2. Here, the "Complexity" means the iteration complexity needed to achieve the $\epsilon$-stationary point of $\Phi_\delta$ (3).

| Methods | LL Constraint | Complexity | Loops |
|---------|---------------|------------|-------|
| [21] | $y \in \mathcal{Y}$, $\mathcal{Y}$ is a convex and compact set | $O(\epsilon^{-3}\log(\epsilon^{-1}))$ | single/double |
| [16] | $h(x,y) \leq 0$, LICQ holds | $O(\epsilon^{-5}\log(\epsilon^{-1}))$ | triple |
| [16] | $B(x) + A(x)y \leq 0$, $A(x)$ is full row rank | $O(\epsilon^{-3}\log(\epsilon^{-1}))$ | double |
| SFLCB (ours) | $Bx + Ay - b \leq 0$, $A$ is full row rank | $O(\epsilon^{-3})$ | single |
| SFLCB (ours) | $Ay \leq 0$, LICQ holds at the initial points | $O(\epsilon^{-3})$ | single |
| SFLCB (ours) | $Ay \leq 0$ | $O(\epsilon^{-4})$ | single |

and unconstrained, $y^*(x) = \arg\min_y g(x,y)$ exists and is differentiable, and the gradient of the hyper-objective can be calculated by $\nabla\Phi(x) = \nabla_x f(x,y) + (\nabla y^*(x))^\top \nabla_y f(x, y^*(x))$ [9]. Later works improved the convergence rates and studied the gradient descent methods under various settings [15, 14, 4, 19, 47]. Another popular approach is based on iterative differentiation, which iteratively solves the LL problems and computes $\nabla y^*(x)$ to approximate the hypergradient [34, 10, 29, 2]. Recently, penalty-based methods have gained traction as a promising approach for solving BLO. Those works usually reformulate the original BLO as the single-level one and use the first-order methods to find the stationary point of the reformulated problems [28, 35, 25, 22, 39, 8, 31, 32].

**BLO with constraints.** There are two primary types of methods for solving constrained bilevel optimization problems. One is based on the implicit gradient method. Generally, when the LL problem has constraints, the differentiabilities of $y^*(x)$ and $\Phi(x)$ are not guaranteed [18]. [42] proved the existence of $\nabla\Phi(x)$ under additional assumptions for linearly constraint $Ay \leq b$ and proposed an implicit gradient-type double-loop algorithm. [18] proposed a perturbation-based smoothing technique to compute the approximate implicit gradient for linearly constraint $Ay \leq b$. [46] used Clarke subdifferential to approximate the non-differentiable implicit function $\Phi$. However, they only provided an asymptotic convergence analysis of their algorithm. [45] proved the existence of $\nabla\phi$ where the LL has equality constraints $Ay + H(x) = c$, and introduced an alternating projected SGD approach to solve this problem. However, these implicit gradient-type algorithms [42, 18, 46, 45] require the computations for the Hessian matrix of the LL problems, which potentially limit their practical applicability for large-scale problems.

Another commonly used approach for solving constrained BLO problems is based on penalty reformulation. For example, [32] reformulated unconstrained and constrained BLO problems as structured minimax problems and introduced first-order methods with guarantees for finding $\epsilon$-KKT solutions. [50] reformulated the original problem into a proximal Lagrangian value function and proposed a single-loop, first-order method to find the stationary points of the reformulated value function. However, their algorithm requires the implementation of the projection operator on $\mathcal{C} = \{x, y | h(x,y) \leq 0\}$ at each iteration, which can be potentially costly. [49] reformulated the original problem into a doubly regularized gap function and proposed a single-loop, first-order algorithm. Compared to [50], [49] did not need the projection operator to the coupled constraint set. However, both [50] and [49] did not establish very clear relationships between the stationary points of their approximated problems and the original one. For example, [49] only provided an asymptotic relationship between the original problem and their reformulated one, i.e., as their penalty parameter approaches infinite, their reformulated problem is equivalent to the original one. Recently, [43, 17] proposed algorithms based on barrier approximation approach for constrained BLO problems. However, their algorithms also require the computations for the Hessian matrix.

[21] and [16] considered the same reformulation as ours. [21] studied the case where the LL variables $y \in \mathcal{Y}$ are independent of $x$ and characterized the conditions under which the values and derivatives of $\Phi$ and $\Phi_\delta$ can be $O(\delta)$-close for $y \in \mathcal{Y}$ constraints. Compared with [21], we prove similar results under coupled constraints $\mathcal{Y}(x) = \{y \in \mathbb{R}^{d_y} | h(x,y) \leq 0\}$. Moreover, the algorithms in [21] require the implementation of the projection operator to $\mathcal{Y}$ at each iteration, which can be costly. [16] studied the coupled constraints $\mathcal{Y}(x) = \{y \in \mathbb{R}^{d_y} | h(x,y) \leq 0\}$. While [16] considered more general constraints than ours, however, it did not characterize the gap between the stationary point of the reformulated function $\Phi_\delta$ and the original hyper-objective $\Phi$. Our Theorem 4.9 provides further

justifications for their reformulation in coupled constraints. Moreover, compared with the double- and triple-loop algorithms in [21], we propose a single-loop algorithm SFLCB and prove an improvement in the convergence rate from $O(\epsilon^{-3} \log(\epsilon^{-1}))$ to $O(\epsilon^{-3})$.

Recently, [20] also proposed first-order methods for linearly constrained BLO. Especially, they proved a nearly optimal convergence rate $\tilde{O}(\epsilon^{-2})$ for linear equality constraints and proposed algorithms that can attain $(\delta, \epsilon)$-Goldstein stationarity for linear inequality constraints. However, their convergence rates for linear inequality constraints either have additional dependence on dimension $d$ (such as $\tilde{O}(d\delta^{-1}\epsilon^{-3})$) or need additional assumptions to access the *exact* optimal dual variable (such as $\tilde{O}(\delta^{-1}\epsilon^{-4})$), while we do not require the exact optimal dual variable assumption. Compared with the double-loop algorithms in [20], our proposed single-loop one is easier to implement in practice. Moreover, our techniques are also different from theirs under linear inequality constraints, thereby highlighting the distinct contributions and independent interests of our work.

## 3 Preliminaries

**Notation.** For vectors $a, b \in \mathbb{R}^d$, we denote $a \leq b$ if for all $i \in [d]$, $a_i \leq b_i$. We use $\| \cdot \|$ to denote the $l_2$ norm of a vector and the spectral norm of a matrix. We define the projection operator that project $x$ to a set $\mathcal{P}$ as $\Pi_{\mathcal{P}}(x) = \operatorname{argmin}_{x' \in \mathcal{P}} \frac{1}{2}\|x - x'\|^2$. We denote the projection operator that projects a $x \in \mathbb{R}^d$ to the set $\mathbb{R}^d_-$ as $\Pi_-(x)$.

We state the following assumptions for problem (1), which are commonly used in the theoretical studies of BLO.

**Assumption 3.1.** For any $x \in \mathcal{X}$, $\mathcal{Y}(x)$ is nonempty, closed, and convex and $\Phi(x)$ is lower bounded by a finite, $\Phi^* = \inf_{x \in \mathcal{X}} \Phi(x) \geq -\infty$.

**Assumption 3.2.** $f, \nabla f, \nabla g$ are Lipschitz continuous with $l_{f,0}, l_{f,1}, l_{g,1}$ respectively, jointly over $\mathcal{X} \times \mathcal{Y}(x)$.

**Assumption 3.3.** For any fixed $x \in \mathcal{X}$, $g(x, y)$ is $\mu_g$-strongly convex with respect to $y \in \mathcal{Y}(x)$.

We introduce the standard definition of the $\epsilon$-stationary point for a differentiable function.

**Definition 3.4.** We say $\hat{x}$ is an $\epsilon$-stationary point of a differentiable function $f$ if $\|\nabla f(\hat{x})\| \leq \epsilon$.

## 4 Reformulation

In this section, we provide a theoretical justification for our reformulation (3) and establish the conditions under which the function values and gradients of the reformulated function $\Phi_\delta$ and the original hyper-objective $\Phi$ become sufficiently close. Note that in this section, we considered general coupled constraints $\mathcal{Y}(x) = \{y \in \mathbb{R}^{d_y} | h(x, y) \leq 0\}$ which include, but are not limited to, the linear constraint case. The complete proofs for the lemmas and theorems in this section can be found in Appendix C.

First, we assume $\delta \leq \frac{\mu_g}{2l_{f,1}}$ and introduce the following notations:

$$y_\delta^*(x) = \operatorname*{argmin}_{y \in \mathcal{Y}(x)} \delta f(x, y) + g(x, y)$$

$$y^*(x) = z^*(x) = \operatorname*{argmin}_{y \in \mathcal{Y}(x)} g(x, y)$$

$$\phi_\delta(x, y, z) = \delta \Phi_\delta(x, y, z)$$

$$\phi_\delta(x) = \phi_\delta(x, y_\delta^*(x), z^*(x)).$$

Similar to Theorem 3.8 in [21], we have the following theorem to bound the difference between $\Phi$ and $\Phi_\delta$, as well as $y^*(x)$ and $y_\delta^*(x)$ in the coupled constraints.

**Theorem 4.1.** *When Assumption 3.1, 3.2 and 3.3 hold, we have*

$$0 \leq \Phi(x) - \Phi_\delta(x) \leq \frac{\delta l_{f,0}^2}{2\mu_g}, \quad \|y_\delta^*(x) - y^*(x)\| \leq \frac{2\delta l_{f,0}}{\mu_g}.$$

Theorem 4.1 characterizes how the difference in function values and the optimal LL variables between the reformulated problem and the original one are controlled by the penalty parameter $\delta$. Therefore, by choosing a sufficiently small $\delta$, i.e., $\delta = O(\epsilon)$, we can treat the reformulated problem $\min_{x \in \mathcal{X}} \Phi_\delta(x)$ as an approximation of the original problem and solve this approximated problem instead. In the following lemmas, we will provide the conditions under which the reformulated function $\Phi_\delta(x)$ is differentiable. Before that, we first introduce the well-known and commonly used Linear Independence Constraint Qualification (LICQ) condition.

**Definition 4.2** (Active set). We denote $\mathcal{I}_y \subseteq [d_h]$ as the active set of $y$, i.e. $\mathcal{I}_y = \{i \in [d_h] \mid h_i(x,y) = 0\}$.

**Definition 4.3** (LICQ). We say a point $y$ satisfy the LICQ condition if, for all $i \in \mathcal{I}_y$, $\nabla_y h_i(x,y)$ are linearly independent.

Then, similar to Lemmas 2 and 3 in [16], we have the following lemma.

**Lemma 4.4.** *When Assumption 3.1, 3.2, 3.3 hold and $\delta \le \mu_g/(2l_{f,1})$, if, for all $x \in \mathcal{X}$, the LICQ condition (Definition 4.3) holds for $y^*(x)$ and $y^*_\delta(x)$, then there exist the corresponding unique Lagrangian multipliers $\lambda^*(x) \in \mathbb{R}^{d_h}$ and $\lambda^*_\delta(x) \in \mathbb{R}^{d_h}$ such that*

$$\lambda^*(x) = \underset{\lambda \in \mathbb{R}_+}{\arg\max} \ \min_{y \in \mathcal{Y}(x)} \ g(x,y) + \lambda^\top h(x,y) \tag{4}$$

$$\lambda^*_\delta(x) = \underset{\lambda \in \mathbb{R}_+}{\arg\max} \ \min_{y \in \mathcal{Y}(x)} \ \delta f(x,y) + g(x,y) + \lambda^\top h(x,y). \tag{5}$$

*Furthermore, we have*

$$\nabla \Phi_\delta(x) = \nabla_x f(x, y^*_\delta(x)) + \frac{1}{\delta}[\nabla_x g(x, y^*_\delta(x)) + \nabla_x h(x, y^*_\delta(x))\lambda^*_\delta(x)$$
$$- \nabla_x g(x, y^*(x)) - \nabla_x h(x, y^*(x))\lambda^*(x)].$$

While the gradients of $\Phi_\delta(x)$ exists under LICQ conditions, for general problem (1), $\Phi(x)$ is not guaranteed to be differentiable. For example, [18] provides an example where the LICQ condition holds, $\Phi(x)$ is non-differentiable at some points. However, if a given $x$ satisfies the following conditions, then $\nabla \Phi(x)$ exists at $x$.

**Assumption 4.5** (Strict Complementarity). Let $\lambda^*(x)$ be the Lagrange multipliers for $y^*(x)$ (4). For any $i \in \mathcal{I}_{y^*(x)}$, $[\lambda^*(x)]_i > 0$.

**Assumption 4.6.** $\nabla^2 f, \nabla^2 g$ are Lipschitz continuous with $l_{f,2}, l_{g,2}$ respectively, jointly over $\mathcal{X} \times \mathcal{Y}(x)$. For $i \in [d_h]$, $h_i(x,y)$ is convex with respect to $y$, $h_i, \nabla h_i, \nabla^2 h_i$ are respectively Lipschitz continuous with $l_{h,0}, l_{h,1}, l_{h,2}$ jointly over $\mathcal{X} \times \mathcal{Y}(x)$.

Note that Assumption 4.5, 4.6 are commonly used in constrained BLO literature [42, 18, 46, 17, 21] to ensure the existence of $\nabla \Phi(x)$.

**Lemma 4.7** (Theorem 2 in [46]). *When Assumption 3.1, 3.2, 3.3, 4.6 hold, if, for a given $x$, Assumption 4.5 and LICQ (Definition 4.3) condition hold for $y^*(x)$, then $\nabla \Phi(x)$ exists at $x$.*

Moreover, with additional assumptions, we can establish a non-asymptotic bound for $\|\nabla \Phi(x) - \nabla \Phi_\delta(x)\|$.

**Assumption 4.8.** For any $t \in [0, \delta]$,

(1) $y^*_t(x)$ satisfies the LICQ condition (Definition 4.3) with the same active set as $y^*(x)$. Denote this active set as $\mathcal{I}$. Let $\lambda^*_t(x)$ be the Lagrange multiplier for $y^*_t(x)$ in (5). For any $i \in \mathcal{I}$, $[\lambda^*_t(x)]_i > 0$ (Strict Complementarity). We assume $\|\lambda^*_t(x)\| \le \Lambda$, where $\Lambda$ is an $O(1)$ constant.

(2) Denote $\nabla_y \bar{h}(x, y^*_t(x)) = \nabla_y[h(x, y^*_t(x))]_\mathcal{I}$. The singular values of $\nabla_y \bar{h}(x, y^*_t(x))$ satisfy $\sigma_{\max}([\nabla_y \bar{h}(x, y^*_t(x))) \le s_{\max}$, $\sigma_{\min}(\nabla_y \bar{h}(x, y^*_t(x))) \ge s_{\min} > 0$, where $s_{\max}, s_{\min}$ are $O(1)$ constants.

Assumption 4.8 is made for $t \in [0, \delta]$. When $\delta$ is sufficiently small, i.e., $\delta = O(\epsilon)$, $y^*(x)$ and $y^*_t(x)$ are very close according to Theorem 4.1. Thus, we expect that for $t \in [0, \delta]$, $y^*_t(x)$ will have similar properties as $y^*(x)$. Similar assumptions have also been used in [21] to establish the non-asymptotic bound for $\|\nabla \Phi(x) - \nabla \Phi_\delta(x)\|$.

**Theorem 4.9.** *When Assumption 3.1, 3.2, 3.3, 4.6 hold and $\delta \leq \mu_g/(2l_{f,1})$, if Assumption 4.8 holds for a given $x$, we have*

$$\|\nabla \Phi(x) - \nabla \Phi_\delta(x)\| \leq O(\delta).$$

Similar non-asymptotic bound for $\|\nabla \Phi(x) - \nabla \Phi_\delta(x)\|$ has been established in [21]; however, their bound is established only for the LL constraints $\mathcal{Y}$ that do not depend on $x$. Our Theorem 4.9 provides a more general theoretical justification for the validity of the reformulation (3) for coupled constraints $\mathcal{Y}(x) = \{y \in \mathbb{R}^{d_y} | h(x, y) \leq 0\}$.

## 5 The SFLCB Algorithm

In the last section, we have justified the validity of our reformulation for coupled constrained BLO. In this section, we focus on a special and important case where the LL constraints are $h(x, y) = Bx + Ay - b$. This particular category of constrained BLO problems encompasses a broad range of applications, including distributed optimization [48, 18], adversarial training [53, 18], and hyperparameter optimization for constrained learning tasks such as hyperparameter optimization in SVM (see Section 6). For this special case $h(x, y) = Bx + Ay - b$, we introduce a novel single-loop, first-order algorithm SFLCB, which achieves an improvement in the convergence rate compared to prior works [21, 16].

First, we introduce the following slackness parameters $\alpha, \beta \in \mathbb{R}_-^{d_h}$ and define $y' = (y^\top, \alpha^\top)^\top$, $z' = (z^\top, \beta^\top)^\top$. With these slackness parameters, we can convert the original inequality constraints to equality constraints, i.e. we can reformulate $\min_{x \in \mathcal{X}, y \in \mathcal{Y}(x)} \max_{z \in \mathcal{Y}(x)} \phi_\delta(x, y, z)$ as:

$$\min_{x \in \mathcal{X}, y' \in \mathcal{S}_y(x)} \max_{z' \in \mathcal{S}_y(x)} \phi_\delta(x, y, z) \tag{6}$$

where $\mathcal{P}_y = \{y \in \mathbb{R}^{d_y}, \alpha \in \mathbb{R}_-^{d_h}\}$, $\mathcal{S}_y(x) = \{y, \alpha \in \mathcal{P}_y | h(x, y) - \alpha = 0\}$. The Lagrangian of (6) with multiplier $u, v \in \mathbb{R}^{d_h}$ is

$$L_\delta(x, y', z', u, v) = \phi_\delta(x, y, z) + u^\top (h(x, y) - \alpha) - v^\top (h(x, z) - \beta).$$

According to Proposition 5.3.4 in [1], we know that

$$\phi_\delta(x) = \min_{y' \in \mathcal{P}_y, v \in \mathbb{R}^{d_h}} \max_{z' \in \mathcal{P}_y, u \in \mathbb{R}^{d_h}} L_\delta(x, y', z', u, v).$$

Note that when $\delta \leq \mu_g/(2l_{f,1})$, $\phi_\delta$ is $\mu_g/2$-strongly convex with respect $y$. However, $L_\delta(x, y', z', u, v)$ is only convex with respect $y'$ and concave with respect $z'$. To make the objective function strongly convex with respect to $y'$ and strongly concave with respect to $z'$, we can construct an augmented Lagrangian $K$:

$$K(x, y', z', u, v) = L_\delta(x, y', z', u, v) + \frac{\rho_1}{2} \|h(x, y) - \alpha\|^2 - \frac{\rho_2}{2} \|h(x, z) - \beta\|^2.$$

With $0 \leq \rho_1 \leq \frac{\mu_g - \delta l_{f,1}}{\sigma_{\max}^2(A)}$ and $0 \leq \rho_2 \leq \frac{\mu_g}{\sigma_{\max}^2(A)}$, according to Lemma D.1, $K$ is strongly convex with respect to $y'$ and strongly concave with respect to $z'$. Moreover, we have

$$\min_{y' \in \mathcal{P}_y, v \in \mathbb{R}^{d_h}} \max_{z' \in \mathcal{P}_y, u \in \mathbb{R}^{d_h}} L_\delta(x, y', z', u, v) = \min_{y' \in \mathcal{P}_y, v \in \mathbb{R}^{d_h}} \max_{z' \in \mathcal{P}_y, u \in \mathbb{R}^{d_h}} K(x, y', z', u, v).$$

Note that $L_\delta$ and $K$ have the same optimal points and same optimal function value. Thus, we can reformulate the problem (6) to the minimax optimization problem over $K$:

$$\min_{x \in \mathcal{X}, y' \in \mathcal{P}_y, v \in \mathbb{R}^{d_h}} \max_{z' \in \mathcal{P}_y, u \in \mathbb{R}^{d_h}} K(x, y', z', u, v). \tag{7}$$

Motivated by these theoretical analyses, and applying gradient descent ascent (GDA) over problem (7), we propose SFLCB. A compact description can be found in Algorithm 1.

**Algorithm 1** SFLCB

> Input: $\delta, \rho_1, \rho_2, \eta_x, \eta_y, \eta_z, \eta_v, \eta_u, T$
> Initialize: $x_0 \in \mathcal{X}, y_0', z_0' \in \mathcal{P}_y, u_0, v_0 \in \mathbb{R}^{d_h}$
> **for** $t = 0, 1, ..., T-1$ **do**
> $\quad u_{t+1} = u_t + \eta_u(h(x_t, y_t) - \alpha_t)$
> $\quad v_{t+1} = v_t + \eta_v(h(x_t, z_t) - \beta_t)$
> $\quad x_{t+1} = x_t - \eta_x \nabla_x K(x_t, y_t', z_t', u_{t+1}, v_{t+1})$
> $\quad y_{t+1}' = \Pi_{\mathcal{P}_y}\{y_t' - \eta_y \nabla_y' K(x_t, y_t', z_t', u_{t+1}, v_{t+1})\}$
> $\quad z_{t+1}' = \Pi_{\mathcal{P}_y}\{z_t' + \eta_z \nabla_z' K(x_t, y_t', z_t', u_{t+1}, v_{t+1})\}$
> **end for**

## 5.1 Convergence results

In this section, we provide the non-asymptotic convergence results of SFLCB (Algorithm 1) for two constraint settings: 1) $h(x, y) = Bx + Ay - b$, where $A$ is full row rank, and 2) $h(y) = Ay - b$, where $A$ is not required to be full row rank.

Note that when the LICQ condition (Definition 4.3) holds for $y^*(x)$ and $y_\delta^*(x)$, the optimal Lagrangian multipliers of $y^*(x)$ and $y_\delta^*(x)$ are unique. Thus, we first introduce the following lemma and notations for these optimal Lagrangian multipliers.

**Lemma 5.1.** *When the LICQ condition (Definition 4.3) holds for $y^*(x)$ and $y_\delta^*(x)$, the optimal Lagrangian multipliers of $y^*(x)$ and $y_\delta^*(x)$ are unique, and we have*

$$u_\delta^*(x) = \underset{u \in \mathbb{R}_+}{\operatorname{argmax}} \min_{y \in \mathcal{Y}(x)} g_\delta(x, y) + u^\top h(x, y) = \underset{u \in \mathbb{R}^{d_h}}{\operatorname{argmax}} \min_{y' \in \mathcal{P}_y} K(x, y', z', u, v),$$

$$v^*(x) = \underset{v \in \mathbb{R}_+}{\operatorname{argmax}} \min_{z \in \mathcal{Y}(x)} g(x, z) + v^\top h(x, z) = \underset{v \in \mathbb{R}^{d_h}}{\operatorname{argmin}} \max_{z' \in \mathcal{P}_y} K(x, y', z', u, v).$$

The proof of Lemma 5.1 can be found in Appendix E.

Next, we introduce the following notations:

$$y_\delta'^*(x, u) = \underset{y' \in \mathcal{P}_y}{\operatorname{argmin}} K(x, y', z', u, v), \quad z'^*(x, v) = \underset{z' \in \mathcal{P}_y}{\operatorname{argmax}} K(x, y', z', u, v), \quad A' = (A, -I).$$

Then, we present the convergence results of SFLCB (Algorithm 1) for coupled constraints $h(x, y) = Bx + Ay - b$, where $A$ is full row rank. Note that BLOCC in [16] that achieves the complexity of $O(\epsilon^{-3} \log(\epsilon^{-1}))$ also needs the matrix $A$ to be full row rank (See Table 1). For $A$ that is not full row rank and $B = 0$, we provide the convergence results in Theorem 5.4 and Corollary 5.5.

**Theorem 5.2.** *When $h(x, y) = Bx + Ay - b$, $A$ is full row rank, Assumption 3.1, 3.2, 3.3 hold and $\delta = \Theta(\epsilon) \leq \mu_g/(2l_{f,1})$, if we apply Algorithm 1 with appropriate parameters (see Appendix E), then we can find an $\epsilon$-stationary point of $\Phi_\delta$ with a complexity of $O(\epsilon^{-4})$.*

*Moreover, if we have initial points $x_0, y_0, z_0, u_0, v_0$ such that*

$$\|y_0 - y_\delta^*(x_0)\| \leq O(\delta), \|u_0 - u_\delta^*(x_0)\| \leq O(\delta), \tag{8}$$

$$\|z_0 - z^*(x_0)\| \leq O(\delta), \|v_0 - v^*(x_0)\| \leq O(\delta), \tag{9}$$

*then we can find an $\epsilon$-stationary point of $\Phi_\delta$ with a complexity of $O(\epsilon^{-3})$.*

The formal statement and the complete proof of Theorem 5.2 can be found in Appendix E.

**Proof sketch for Theorem 5.2.** The key new idea in our proof is the construction of a novel potential function $V_t$ and prove the descent lemma of $V_t$ (Lemma E.4). $V_t$ is defined as:

$$V_t = \frac{1}{4} K(x_t, y_t', z_t', u_t, v_t) + 2q(x_t, v_t) - d(x_t, z_t', u_t, v_t)$$

where $d(x, z', u, v) = K(x, y_\delta'^*(x, u), z', u, v)$ and $q(x, v) = \phi_\delta(x, y_\delta^*(x), z^*(x, v)) - v^\top(A'z'^*(x, v) - b) - \frac{\rho_2}{2}\|A'z'^*(x, v) - b\|^2$. To prove Lemma E.4, we need to first prove several novel error bounds in Lemma D.2, Lemma E.3 and Lemma E.2. Those error bounds may be of

independent interest for solving other similar problems. The full row rank property of $A$ is used in Lemma E.2 to bound $\|u_{t+1} - u_\delta^*(x_t)\|$ and $\|v_{t+1} - v^*(x_t)\|$.

Since $A$ has full row rank, then according to Theorem 6 in [16], we can easily find initial points satisfying (8)-(9) with a complexity of $O(\log(\epsilon^{-1}))$ and we have the following corollary.

**Corollary 5.3.** *When $h(x, y) = Bx + Ay - b$, $A$ has full row rank, Assumption 3.1, 3.2, 3.3 hold, and $\delta = \Theta(\epsilon) \leq \mu_g/(2l_{f,1})$, if we apply projected gradient descent (PGD) for $\max_{v \in \mathbb{R}_+^{d_h}} \min_{z \in \mathbb{R}^{d_y}} g(x_0, z) + v^\top (Bx_0 + Az - b)$ with a fixed $x_0$, we can find $\hat{v}$, $\hat{z}$ such that $\|\hat{v} - v^*(x_0)\| \leq \delta$ and $\|\hat{z} - z^*(x_0)\| \leq \delta$ with a complexity of $O(\log(\epsilon^{-1}))$. Set $y_0 = z_0 = \hat{z}$, $u_0 = v_0 = \hat{v}$, $\alpha_0 = h(x_0, y_0)$, $\beta_0 = h(x_0, z_0)$. With $x_0, y_0', z_0', u_0, v_0$ as initial points and applying Algorithm 1, we can find an $\epsilon$-stationary point of $\Phi_\delta$ with a complexity of $O(\epsilon^{-3})$. Thus, the total complexity is $O(\epsilon^{-3} + \log(\epsilon^{-1})) = O(\epsilon^{-3})$.*

The proof of Corollary 5.3 can be found in Appendix E. Thus, compared to [16], we achieve an improvement in the convergence rate from $O(\epsilon^{-3}\log(\epsilon^{-1}))$ to $O(\epsilon^{-3})$ for the coupled linear constraint (See Table 1).

Additionally, we have the following convergence results for constraints $h(y) = Ay - b$, where $A$ is not required to have a full row rank.

**Theorem 5.4.** *When $h(x, y) = Ay - b$, Assumption 3.1, 3.2, 3.3 hold, and $\delta = \Theta(\epsilon) \leq \mu_g/(2l_{f,1})$, if we apply Algorithm 1 with appropriate parameters (see Appendix D), then we can find an $\epsilon$-stationary point of $\Phi_\delta$ with a complexity of $O(\epsilon^{-4})$.*

*Moreover, if we have initial points $x_0, y_0', z_0', u_0, v_0$ such that*

$$\|y_0 - y_\delta^*(x_0)\| \leq O(\delta), \|A'y_\delta'^*(x_0, u_0) - b\| \leq O(\delta), \|A'y_0' - b\| \leq O(\delta) \tag{10}$$

$$\|z_0 - z^*(x_0)\| \leq O(\delta), \|A'z'^*(x_0, v_0) - b\| \leq O(\delta) \|A'z_0' - b\| \leq O(\delta) \tag{11}$$

*then we can find an $\epsilon$-stationary point of $\Phi_\delta$ with a complexity of $O(\epsilon^{-3})$.*

The formal statement and the complete proof of Theorem 5.4 can be found in Appendix D.

**Proof sketch for Theorem 5.4**. The general proof flow of Theorem 5.4 is similar to that of Theorem 5.2. However, since here we do not have coupled constraints, $\nabla_x K$ has no relationship to $u$ or $v$, and according to the Danskin's theorem, $\nabla\phi_\delta(x) = \delta\nabla_x f(x, y_\delta^*(x)) + \nabla_x g(x, y_\delta^*(x)) - \nabla_x g(x, z^*(x))$ also has no relationship to $u_\delta^*(x)$ or $v^*(x)$. Thus, we do not require Lemma E.2 or the full row-rank assumption on $A$ in this setting.

Next, we show that, as long as the LICQ condition (Definition 4.3) holds for the initial $y^*(x_0)$ and $y_\delta^*(x_0)$, we can find initial points satisfying (10)-(11) with a complexity of $O(\epsilon^{-2})$ and we have the following corollary.

**Corollary 5.5.** *When $h(x, y) = Ay - b$, Assumption 3.1, 3.2, 3.3 hold, and $\delta = \Theta(\epsilon) \leq \mu_g/(2l_{f,1})$, for a given initial point $x_0$, if the LICQ condition (Definition 4.3) holds at $y^*(x_0)$ and $y_\delta^*(x_0)$, we can apply Algorithm 1 with fixed $x_0$. Then for a sufficiently small $\epsilon$ (see Appendix D), we can find $\hat{y}', \hat{z}', \hat{u}, \hat{v}$ such that*

$$\|\hat{y} - y_\delta^*(x_0)\| \leq O(\delta), \|A'y_\delta'^*(x_0, \hat{u}) - b\| \leq O(\delta), \|A'\hat{y}' - b\| \leq O(\delta), \|\hat{u} - u_\delta^*(x_0)\| \leq O(\delta)$$

$$\|\hat{z} - z^*(x_0)\| \leq O(\delta), \|A'z'^*(x_0, \hat{v}) - b\| \leq O(\delta), \|A'\hat{z}' - b\| \leq O(\delta), \|\hat{v} - v^*(x_0)\| \leq O(\delta)$$

*with a complexity of $O(\epsilon^{-2})$. Set $y_0' = \hat{y}', z_0' = \hat{z}', u_0 = \hat{u}, v_0 = \hat{v}$. With $x_0, y_0', z_0', u_0, v_0$ as initial points, we can find an $\epsilon$-stationary point of $\Phi_\delta$ with a complexity of $O(\epsilon^{-3})$. Thus, the total complexity is $O(\epsilon^{-3} + \epsilon^{-2}) = O(\epsilon^{-3})$.*

The proof of Corollary 5.5 can be found in Appendix D. The key to proving Corollary 5.5 lies in Lemma D.4. In Lemma D.4, we prove that, without the full row-rank assumption on $A$, we can bound $\|u_{t+1} - u_\delta^*(x_0)\|$ and $\|v_{t+1} - v^*(x_0)\|$ with a fixed $x_0$. Thus, we can use our algorithm SFLCB to find suitable initial points with a fixed $x_0$.

Note that in Corollary 5.5, we only need the LICQ condition holds for the initial $y^*(x_0)$ and $y_\delta^*(x_0)$, and we can achieve a total complexity of $O(\epsilon^{-3})$. Compared to the decoupled constrained setting in [21], we achieve an improvement in the convergence rate from $O(\epsilon^{-3}\log(\epsilon^{-1}))$ to $O(\epsilon^{-3})$ (See Table 1).

# 6 Experiments

In this section, we evaluate the performance of our SFLCB algorithm on three tasks: a toy example, hyperparameter optimization for SVM, and a transportation network design problem. These experiments demonstrate the practical effectiveness and efficiency of SFLCB. Additional hype-parameter sensitivity analysis experiments and detailed experimental settings can be found in Appendix F.

## 6.1 Toy example

We consider the same constrained BLO problem that was studied in [16], which is:

$$\min_{x \in [0,3]} f(x, y^*(x)) = \frac{e^{-y_g^*(x)+2}}{2 + \cos(6x)} + \frac{1}{2} \ln \left( (4x-2)^2 + 1 \right) \tag{12}$$

$$\text{s.t.} \quad y^*(x) \in \arg \min_{y \in \mathcal{Y}(x)} g(x, y) = (y - 2x)^2 \tag{13}$$

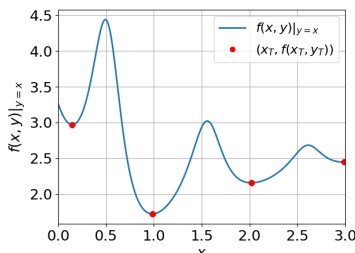

where $\mathcal{Y}(x) = \{y \in \mathbb{R} | y \leq x\}$. Note that this problem satisfies Assumption 3.1, 3.2, 3.3. Moreover, we have $y^*(x) = x$ and Equation (12) is equivalent to $\min_{x \in [0,3]} f(x, y)|_{y=x}$. In Figure 1, we plot the hyper-objective function $f(x, y^*(x))$. The red points indicate the converged solutions obtained by our algorithm with 200 different initialization values. We notice that SFLCB consistently finds the local minima of the hyper-objective function, which validates the effectiveness of SFLCB.

Figure 1: Toy example.

## 6.2 Hyperparameter optimization in SVM

Hyperparameter optimization in SVM is a well-known real-world application for constrained BLO problems that has been used in many prior works [46, 50, 49, 16]. Here we consider the same problem formulation as in [16], which formulates this problem as a coupled linearly constrained BLO problem. We conduct experiments comparing our SFLCB algorithm with GAM [46], LV-HBA [50], BLOCC [16], and BiC-GAFFA [49] on the diabetes dataset [6]. Results are plotted in Figure 2. We notice that our SFLCB algorithm converges significantly faster than other algorithms, which demonstrates the practical efficiency of the proposed SFLCB algorithm.

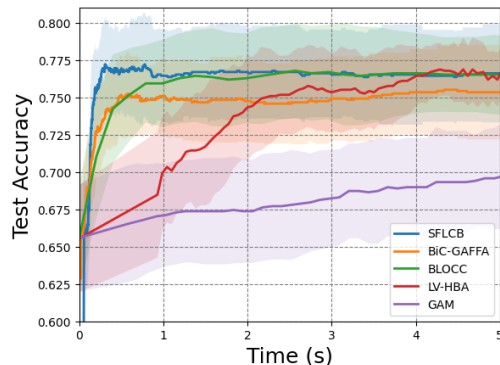

Figure 2: Hyperparameter optimization in SVM.

## 6.3 Transportation network design

We further conduct experiments on a transportation network design problem, following the same setting as in [16]. In this setting, we act as the operator, whose profit serves as the upper-level objective and is influenced by passenger behavior, which is modeled in the lower-level problem. Detailed formulations and settings can be found in Appendix F. We consider the two synthetic networks of 3 and 9 nodes, same as those considered in [16]. We compare SFLCB with BLOCC [16]. Results are plotted in Figure 3, which indicate that SFLCB significantly outperforms BLOCC on this network design task.

### 6.3.1 Sensitivity analysis of $\delta$

We also conduct the sensitivity analysis of the $\delta$ in SFLCB for the 3-node network. We set $\rho_1 = \rho_2 = 1000$, $\eta_x = \eta_y = \eta_z = \eta_u = \eta_v = 3e - 4$, and $T = 20000$. Then, we test different $\delta$ values from $\{0.01, 0.05, 0.1, 0.5, 1\}$. For each $\delta$, we test with three different random seeds. The final average results and one standard deviation are reported in Figure 4. As can be seen, larger values of

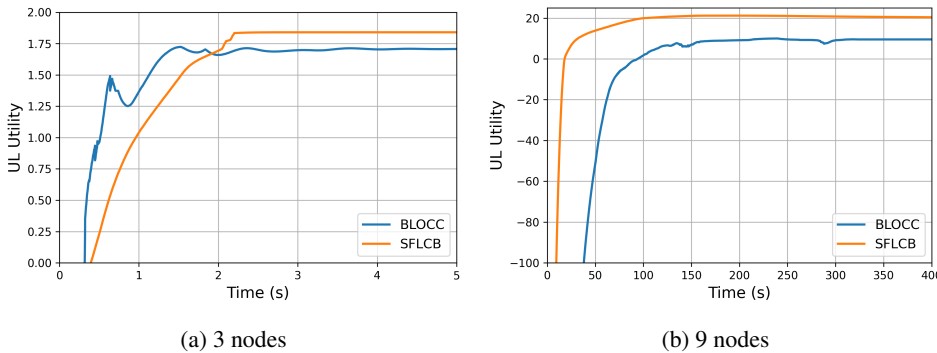

(a) 3 nodes

(b) 9 nodes

Figure 3: Results of the transportation experiments on 3 nodes and 9 nodes settings. Larger UL utility indicates better performance.

$\delta$ lead to a faster initial decrease in the loss (increase in UL utility). In contrast, very small $\delta$ (e.g., $\delta = 0.01$) results in significantly slower convergence overall. However, an overly large $\delta$ (e.g., $\delta = 1$) can lead to large approximation errors in later stages, causing deviation from the true optimization objective and ultimately poor performance. We observe that moderate values of $\delta$ (such as 0.05, 0.1, and 0.5) achieve relatively good final performance. These observations are consistent with our theoretical predictions. For example, our theory indicates that the convergence rate of SLFCB is inversely proportional to $\delta$: smaller $\delta$ leads to slower convergence but smaller approximation error, whereas larger $\delta$ improves convergence speed towards the approximate problem but incurs greater approximation error. Figure 4 indicates a properly chosen $\delta$ thus can balance convergence speed and approximation error.

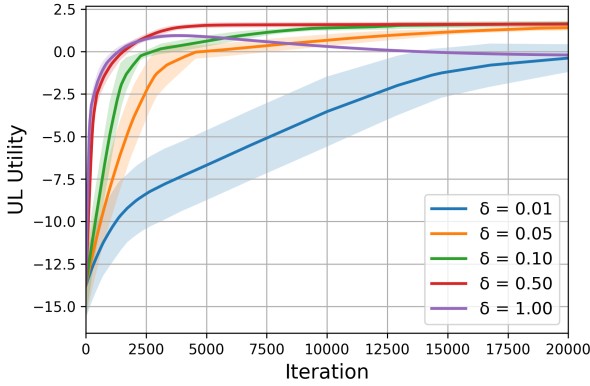

Figure 4: Comparison of different $\delta$ in SFLCB for the 3 nodes network.

## 7 Conclusions and future directions

In this paper, for coupled constrained BLO problem in Equation (1), we theoretically analyzed the relationship between the original hyper-objective $\Phi$ and the reformulated function $\Phi_\delta$ in Equation (3), providing a solid justification for the validity of the reformulation. Especially, for the linearly constrained case, we proposed SFLCB, a single-loop, Hessian-free algorithm, improving the convergence rate from $O(\epsilon^{-3} \log(\epsilon^{-1}))$ to $O(\epsilon^{-3})$ over previous works [21, 16]. Our experiments on hyperparameter optimization for SVM and the transportation network design problem validated the practical efficiency of the proposed SFLCB algorithm. One limitation of our work is that the analysis is restricted to deterministic and linearly constrained settings. A promising direction for future research is to extend the current results to stochastic environments or more general constraint structures. Moreover, since the best-known complexity for first-order methods in unconstrained BLO [12] is $O(\epsilon^{-2})$, it is also an interesting problem whether we can achieve this optimal rate in the constrained cases.

## Acknowledgements

The work of Wei Shen and Cong Shen was supported in part by the US National Science Foundation (NSF) under awards 2143559 and 2332060. The work of Jiawei Zhang was supported by the Office of the Vice Chancellor for Research and Graduate Education at the University of Wisconsin–Madison with funding from the Wisconsin Alumni Research Foundation.

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

# Appendix

The Appendix is organized as follows. In Appendix A, we introduce some useful lemmas that will be utilized in the subsequent proofs. In Appendix B, we introduce some notations that will be used throughout the Appendix. In Appendix C, we provide the proofs for the lemmas and theorems in Section 4. In Appendix D, we provide proofs for Theorem 5.4 and Corollary 5.5. In Appendix E, we provide proofs for Theorem 5.2 and Corollary 5.3. In Appendix F, we present the detailed experimental settings along with additional hyperparameter sensitivity analysis experiments.

## A  Useful Lemmas

**Lemma A.1** (Lemma 12 in [3]). *Suppose $f(\cdot)$ is $l$-smooth and $\mu$-strongly convex, $\mathcal{X}$ is a convex closed set, and $\eta \leq 1/l$. Define $x^* = \operatorname{argmin}_{x \in \mathcal{X}} f(x)$ and $x^+ = \Pi_{\mathcal{X}}(x - \eta \nabla f(x))$. Then, we have*

$$\|x - x^*\| \leq \frac{2}{\mu \eta} \|x - x^+\|$$

**Lemma A.2** (Lemma 23 in [24]). *Suppose for any fixed $y \in \mathcal{Y}$, $f(x, y)$ is $\mu$-strongly convex w.r.t. $x$ and suppose for any $y_1, y_2 \in \mathcal{Y}, x \in \mathcal{X}$, $\|\nabla_x f(x, y_1) - \nabla_x f(x, y_2)\| \leq l\|y_1 - y_2\|$. Define $x^*(y) = \operatorname{argmin}_{x \in \mathcal{X}} f(x, y)$. Then, we have*

$$\|x^*(y_1) - x^*(y_2)\| \leq \frac{l}{\mu} \|y_1 - y_2\|$$

**Lemma A.3** (Theorem 4.1 in [51]). *Suppose $f(x) : \mathbb{R}^{d_1} \to \mathbb{R}$ is $l$-smooth and $\mu$-strongly convex, $\mathcal{P} = \{x | Cx \leq e\}, \mathcal{S}(r) = \{x \in \mathcal{P} | Ax - b = r\}$, where $C, A \in \mathbb{R}^{d_2 \times d_1}$, $e, b, r \in \mathbb{R}^{d_2}$. Define*

$$L(x, u) = f(x) + u^\top (Ax - b)$$
$$x^*(u) = \operatorname*{argmin}_{x \in \mathcal{P}} L(x, u)$$
$$x_r^* = \operatorname*{argmin}_{x \in \mathcal{S}(r)} f(x)$$

*where $u \in \mathbb{R}^{d_2}$ is the Lagrange multiplier. We have*

$$\|x^*(u) - x_0^*\| \leq \sigma_x \|Ax^*(u) - b\|$$
$$\|x_r^* - x_0^*\| \leq \sigma_x \|r\|$$

*where*

$$\sigma_x = \frac{\sqrt{2}(\bar{\theta} l^2 + 1)}{\mu}$$

$$M = \begin{pmatrix} A^\top & C^\top \\ 0 & I \end{pmatrix}$$

$$\bar{\theta} = \max_{\bar{M} \in \mathcal{B}(M)} \sigma_{\max}^2(\bar{M}) / \sigma_{\min}^4(\bar{M})$$

*$\bar{M}$ is the set of all submatrices of $M$ with full row rank.*

**Lemma A.4.** *Suppose $f(x) : \mathbb{R}^{d_x} \to \mathbb{R}$ is $\mu$-strongly convex w.r.t. $x$. Define $L(x, \alpha, u) = f(x) + u^\top (Ax - b - \alpha) + \frac{\rho}{2}\|Ax - b - \alpha\|^2$, where $\alpha, u, b \in \mathbb{R}^{d_y}$, $A \in \mathbb{R}^{d_y \times d_x}$, $\rho \in (0, \mu/\sigma_{\max}^2(A))$. Denoting $x' = (x^\top, \alpha^\top)^\top$, $L(x', u) = L(x, \alpha, u)$, we have $L(x', u)$ is $\mu_x$-strongly convex w.r.t. $x'$, where $\mu_x = \min\{\mu - \rho\sigma_{\max}^2(A), \frac{\rho}{2}\}$.*

*Proof.* For any $x_1, x_2 \in \mathbb{R}^{d_x}, \alpha_1, \alpha_2 \in \mathbb{R}^{d_y}$, denoting $x_1' = (x_1^\top, \alpha_1^\top)^\top$, $x_2' = (x_2^\top, \alpha_2^\top)^\top$, $A' = (A, -I)$, we have

$$
\begin{aligned}
&L(x_1', u) - L(x_2', u) - \langle \nabla_{x'} L(x_2', u), x_1' - x_2' \rangle \\
=& f(x_1) - f(x_2) + u^\top A'(x_1' - x_2') + \frac{\rho}{2}\|A'x_1' - b\|^2 - \frac{\rho}{2}\|A'x_2' - b\|^2 \\
& - \langle \nabla_x f(x_2), x_1 - x_2 \rangle - u^\top A'(x_1' - x_2') - \rho(A'^\top A'x_2' - A'^\top b)^\top(x_1' - x_2') \\
\geq& \frac{\mu}{2}\|x_1 - x_2\|^2 + \frac{\rho}{2}\|A'(x_1' - x_2')\|^2 \\
\geq& \frac{\mu}{2}\|x_1 - x_2\|^2 + \frac{\rho}{4}\|\alpha_1 - \alpha_2\|^2 - \frac{\rho}{2}\|A(x_1 - x_2)\|^2 \\
\geq& \frac{\mu - \rho\sigma_{\max}^2(A)}{2}\|x_1 - x_2\|^2 + \frac{\rho}{4}\|\alpha_1 - \alpha_2\|^2 \\
\geq& \frac{\mu_x}{2}\|x_1' - x_2'\|^2
\end{aligned}
$$

where $\mu_x = \min\{\mu - \rho\sigma_{\max}^2(A), \frac{\rho}{2}\}$. $\qquad\square$

# B  Notations

Denote $l_\delta = \mu_g/2$. We introduce the following notations.

$$
\begin{aligned}
\Phi_\delta(x, y, z) &= \frac{1}{\delta}\left(\delta f(x, y) + g(x, y) - g(x, z)\right) \\
\phi_\delta(x, y, z) &= \delta f(x, y) + g(x, y) - g(x, z) \\
g_\delta(x, y) &= \delta f(x, y) + g(x, y) \\
y_\delta^*(x) &= \min_{y \in \mathcal{Y}(x)} g_\delta(x, y) \\
y^*(x) &= \min_{y \in \mathcal{Y}(x)} g(x, y) \\
L_g &= l_\delta + l_{g,1} \\
L_\phi &= l_\delta + 2l_{g,1} \\
\Phi(x) &= f(x, y^*(x)) \\
\Phi_\delta(x) &= \Phi_\delta(x, y_\delta^*(x), y^*(x)) \\
\phi_\delta(x) &= \phi_\delta(x, y_\delta^*(x), y^*(x))
\end{aligned}
$$

When $\delta \leq \mu_g/2l_{f,1}$, we have $\delta l_{f,1} \leq \mu_g/2 = l_\delta$ and thus, $g_\delta(x, y)$ is $L_g$-smooth and $\phi_\delta(x, y, z)$ is $L_\phi$-smooth.

# C  Reformulation

In this section, we provide the proofs for the lemmas and theorems in Section 4.

### Theorem 4.1

*When Assumption 3.1, 3.2 and 3.3 hold, we have*

$$
0 \leq \Phi(x) - \Phi_\delta(x) \leq \frac{\delta l_{f,0}^2}{2\mu_g}
$$

$$
\|y_\delta^*(x) - y^*(x)\| \leq \frac{2\delta l_{f,0}}{\mu_g}.
$$

*Proof.* Similar results and proofs of Theorem 4.1 can also be found in [21, 16]. For completeness, we also provide our proofs for Theorem 4.1 here.

Note that $y_\delta^*(x)$ satisfy

$$\Pi_{\mathcal{Y}(x)}(y_\delta^*(x) - [\delta\nabla_y f(x, y_\delta^*(x)) + \nabla_y g(x, y_\delta^*(x))]/L_\phi) = y_\delta^*(x)$$

Since $g(x, \cdot)$ is $\mu_g$-strongly concave and $l_{g,1}$-smooth and $L_\phi \geq l_{g,1}$, according to Lemma A.1, we have

$$
\begin{aligned}
\|y_\delta^*(x) - z^*(x)\| &\leq \frac{2L_\phi}{\mu_g}\|y_\delta^*(x) - \Pi_{\mathcal{Y}(x)}(y_\delta^*(x) - \nabla_y g(x, y_\delta^*(x))/L_\phi)\| \\
&= \frac{2L_\phi}{\mu_g}\|\Pi_{\mathcal{Y}(x)}(y_\delta^*(x) - [\delta\nabla_y f(x, y_\delta^*(x)) + \nabla_y g(x, y_\delta^*(x))]/L_\phi) \\
&\quad - \Pi_{\mathcal{Y}(x)}(y_\delta^*(x) - \nabla_y g(x, y_\delta^*(x))/L_\phi)\| \\
&\leq \frac{2\delta}{\mu_g}\|\nabla_y f(x, y_\delta^*(x))\| \\
&\leq \frac{2\delta l_{f,0}}{\mu_g}.
\end{aligned}
$$

For $\Phi_\delta(x)$, we have

$$
\begin{aligned}
\Phi_\delta(x) &= f(x, y_\delta^*(x)) + \frac{1}{\delta}(g(x, y_\delta^*(x)) - g(x, y^*(x))) \\
&\leq f(x, y^*(x)) + \frac{1}{\delta}(g(x, y^*(x)) - g(x, y^*(x))) \\
&= \Phi(x),
\end{aligned}
$$

and

$$
\begin{aligned}
\Phi_\delta(x) &= f(x, y_\delta^*(x)) + \frac{1}{\delta}[g(x, y_\delta^*(x)) - g(x, y^*(x))] \\
&\geq f(x, y_\delta^*(x)) + \frac{\mu_g}{2\delta}\|y_\delta^*(x) - y^*(x)\|^2 \\
&\geq f(x, y^*(x)) + \frac{\mu_g}{2\delta}\|y_\delta^*(x) - y^*(x)\|^2 - l_{f,0}\|y_\delta^*(x) - y^*(x)\| \\
&\geq \Phi(x) - \frac{\delta l_{f,0}^2}{2\mu_g},
\end{aligned}
$$

where the first equality is due to the quadratic growth of a strongly convex function, the last equality is due to $ax^2 + bx \geq -b^2/(4a)$.

$\square$

## Lemma 4.4

*When Assumption 3.1, 3.2, 3.3 hold and $\delta \leq \mu_g/(2l_{f,1})$, if, for all $x \in \mathcal{X}$, LICQ condition (Definition 4.3) hold for $y^*(x)$ and $y_\delta^*(x)$, then there exist the corresponding unique Lagrangian multipliers $\lambda^*(x) \in \mathbb{R}^{d_h}$ and $\lambda_\delta^*(x) \in \mathbb{R}^{d_h}$ such that*

$$\lambda^*(x) = \operatorname*{argmax}_{\lambda\in\mathbb{R}_+} \min_{y\in\mathcal{Y}(x)} g(x, y) + \lambda^\top h(x, y) \tag{14}$$

$$\lambda_\delta^*(x) = \operatorname*{argmax}_{\lambda\in\mathbb{R}_+} \min_{y\in\mathcal{Y}(x)} \delta f(x, y) + g(x, y) + \lambda^\top h(x, y). \tag{15}$$

*Furthermore, we have*

$$
\begin{aligned}
\nabla\Phi_\delta(x) =& \nabla_x f(x, y_\delta^*(x)) + \frac{1}{\delta}[\nabla_x g(x, y_\delta^*(x)) + \nabla_x h(x, y_\delta^*(x))\lambda_\delta^*(x) \\
& - \nabla_x g(x, y^*(x)) - \nabla_x h(x, y^*(x))\lambda^*(x)].
\end{aligned}
\tag{16}
$$

*Proof.* The uniqueness of Lagrangian multipliers is a direct result from the LICQ condition [44]. According to Lemmas 2 and 3 in [16]. We know that when Assumption 3.1, 3.2, 3.3 hold, $\delta \leq$

$\mu_g/(2l_{f,1})$, if, for all $x \in \mathcal{X}$, LICQ condition (Definition 4.3) holds for $y^*(x)$ and $y^*_\delta(x)$, defining $\psi(x) = g(x, y^*(x))$ and $\psi_\delta(x) = g_\delta(x, y^*_\delta(x))$, we have

$$\nabla_x \psi(x) = \nabla_x g(x, y^*(x)) + \nabla_x h(x, y^*(x))\lambda^*(x)$$
$$\nabla_x \psi_\delta(x) = \delta\nabla_x f(x, y^*_\delta(x)) + \nabla_x g(x, y^*_\delta(x)) + \nabla_x h(x, y^*_\delta(x))\lambda^*_\delta(x)$$

Since $\Phi_\delta(x) = f(x, y^*_\delta(x)) + \frac{1}{\delta}(g_\delta(x, y^*_\delta(x)) - g(x, y^*(x)))$, we have

$$\nabla\Phi_\delta(x) = \nabla_x f(x, y^*_\delta(x)) + \frac{1}{\delta}[\nabla_x g(x, y^*_\delta(x)) + \nabla_x h(x, y^*_\delta(x))\lambda^*_\delta(x)$$
$$- \nabla_x g(x, y^*(x)) - \nabla_x h(x, y^*(x))\lambda^*(x)].$$

$\square$

**Lemma 4.7**

*When Assumption 3.1, 3.2, 3.3, 4.6 hold, if, for a given $x$, Assumption 4.5 and the LICQ condition (Definition 4.3) holds for $y^*(x)$, then $\nabla\Phi(x)$ exists at $x$ and can be expressed as*

$$\nabla\Phi(x) = \nabla_x f(x, y) + (\nabla y^*(x))^\top \nabla_y f(x, y^*(x))$$

*where $\nabla y^*(x)$ can be calculated according to* (20).

*Proof.* According to Theorem 2 in [46], we know that when Assumption 3.1, 3.2, 3.3, 4.6 hold, if, for a given $x$, Assumption 4.5 and the LICQ (Definition 4.3) condition holds for $y^*(x)$, then $\nabla\Phi(x)$ exists at $x$.

Moreover, we can give the explicit expression of $\nabla\Phi(x)$.

With $\lambda \in \mathbb{R}^{d_h}$, we have the following Lagrangian function

$$\mathcal{L}(x, y, \lambda) = g(x, y) + \lambda^\top h(x, y)$$

Denote $\lambda^*(x)$ as the optimal Lagrangian multiplier, $\mathcal{I}_x \subseteq [d_h]$ as the active set of $y^*(x)$, i.e. $\mathcal{I}_x = \{i \in [d_h] | [h(x, y)]_i = 0\}$ Denote $\bar{h}(x, y^*(x)) = [h(x, y^*(x))]_{\mathcal{I}_x}$ and $\bar{\lambda}^*(x) = [\lambda^*(x)]_{\mathcal{I}_x}$. We have the following KKT conditions:

$$\nabla_y g(x, y^*(x)) + \nabla_y \bar{h}(x, y^*(x))^\top \bar{\lambda}^*(x) = 0$$
$$\bar{h}(x, y^*(x)) = 0.$$

Differentiating the KKT conditions with respect to $x$, we have

$$\nabla^2_{xy} g(x, y^*(x)) + \nabla^2_{yy} g(x, y^*(x))\nabla y^*(x) + \nabla^2_{xy} \bar{h}(x, y^*(x))^\top \bar{\lambda}^*(x)$$
$$+ \nabla^2_{yy} \bar{h}(x, y^*(x))^\top \bar{\lambda}^*(x)\nabla y^*(x) + \nabla_y \bar{h}(x, y^*(x))^\top \nabla\bar{\lambda}^*(x) = 0$$
$$\nabla_x \bar{h}(x, y^*(x)) + \nabla_y \bar{h}(x, y^*(x))\nabla y^*(x) = 0$$

Thus, $\nabla y^*(x)$ and $\lambda^*(x)$ satisfy the following equation.

$$\begin{bmatrix} \nabla^2_{yy} g(x, y^*(x)) + \nabla^2_{yy} \bar{h}(x, y^*(x))^\top \bar{\lambda}^*(x) & \nabla_y \bar{h}(x, y^*(x))^\top \\ \nabla_y \bar{h}(x, y^*(x)) & 0 \end{bmatrix} \begin{bmatrix} \nabla y^*(x) \\ \nabla\bar{\lambda}^*(x) \end{bmatrix} \quad (17)$$

$$= \begin{bmatrix} -\nabla^2_{xy} g(x, y^*(x)) - \nabla^2_{xy} \bar{h}(x, y^*(x))^\top \bar{\lambda}^*(x) \\ -\nabla_x \bar{h}(x, y^*(x)) \end{bmatrix} \quad (18)$$

Denote

$$H = \begin{bmatrix} \nabla^2_{yy} g(x, y^*(x)) + \nabla^2_{yy} \bar{h}(x, y^*(x))^\top \bar{\lambda}^*(x) & \nabla_y \bar{h}(x, y^*(x))^\top \\ \nabla_y \bar{h}(x, y^*(x)) & 0 \end{bmatrix} \quad (19)$$

Since $g$ is strongly convex, $h$ is convex, $\nabla^2_{yy} g(x, y^*(x)) \succ 0$, $\nabla^2_{yy}[\bar{h}(x, y^*(x))]_i \succeq 0$. Moreover, since $\bar{\lambda}^*(x) > 0$, we have $\nabla^2_{yy} g(x, y^*(x)) + \nabla^2_{yy} \bar{h}(x, y^*(x))^\top \bar{\lambda}^*(x) \succ 0$. Additionally,

$\nabla_y \bar{h}(x, y^*(x))$ has full row rank. Thus, according to Lemma A.2 in [20], $H$ is invertible. We can calculate $\nabla y^*(x)$ and $\lambda^*(x)$ with the following equation.

$$\begin{bmatrix} \nabla y^*(x) \\ \nabla \bar{\lambda}^*(x) \end{bmatrix} = H^{-1} \begin{bmatrix} -\nabla_{xy}^2 g(x, y^*(x)) - \nabla_{xy}^2 \bar{h}(x, y^*(x))^\top \bar{\lambda}^*(x) \\ -\nabla_x \bar{h}(x, y^*(x)) \end{bmatrix} \tag{20}$$

According to [9], we have

$$\nabla \Phi(x) = \nabla_x f(x, y) + (\nabla y^*(x))^\top \nabla_y f(x, y^*(x)).$$

$\square$

**Theorem 4.9**

*When Assumption 3.1, 3.2, 3.3, 4.6 hold, and $\delta \le \mu_g/(2l_{f,1})$, if Assumption 4.8 holds for a given $x$, we have*

$$\|\nabla \Phi(x) - \nabla \Phi_\delta(x)\| \le O(\delta).$$

*Proof.* With $\lambda \in \mathbb{R}^{d_h}$, we have the following Lagrangian function

$$\mathcal{L}_\delta(x, y, \lambda) = g_\delta(x, y) + \lambda^\top h(x, y).$$

Denote $\lambda_\delta^*(x)$ as the optimal Lagrangian multiplier. We have the following KKT conditions.

$$\nabla_y g_\delta(x, y_\delta^*(x)) + \nabla_y \bar{h}(x, y_\delta^*(x))^\top \bar{\lambda}_\delta^*(x) = 0$$
$$\bar{h}(x, y_\delta^*(x)) = 0.$$

Differentiating the KKT conditions with respect to $\delta$, we have

$$\nabla_y f(x, y_\delta^*(x)) + \nabla_{yy}^2 g_\delta(x, y_\delta^*(x)) \frac{d}{d\delta} y_\delta^*(x) + \nabla_{yy}^2 \bar{h}(x, y_\delta^*(x))^\top \bar{\lambda}_\delta^*(x) \frac{d}{d\delta} y_\delta^*(x)$$

$$+ \nabla_y \bar{h}(x, y_\delta^*(x))^\top \frac{d}{d\delta} \bar{\lambda}_\delta^*(x) = 0$$

$$\nabla_y \bar{h}(x, y_\delta^*(x)) \frac{d}{d\delta} y_\delta^*(x) = 0$$

We have the following equation:

$$\begin{bmatrix} \nabla_{yy}^2 g_\delta(x, y_\delta^*(x)) + \nabla_{yy}^2 \bar{h}(x, y_\delta^*(x))^\top \bar{\lambda}_\delta^*(x) & \nabla_y \bar{h}(x, y_\delta^*(x))^\top \\ \nabla_y \bar{h}(x, y_\delta^*(x)) & 0 \end{bmatrix} \begin{bmatrix} \frac{d}{d\delta} y_\delta^*(x) \\ \frac{d}{d\delta} \bar{\lambda}_\delta^*(x) \end{bmatrix}$$
$$= \begin{bmatrix} -\nabla_y f(x, y_\delta^*(x)) \\ 0 \end{bmatrix}. \tag{21}$$

Denote

$$H_\delta = \begin{bmatrix} \nabla_{yy}^2 g_\delta(x, y_\delta^*(x)) + \nabla_{yy}^2 \bar{h}(x, y_\delta^*(x))^\top \bar{\lambda}_\delta^*(x) & \nabla_y \bar{h}(x, y_\delta^*(x))^\top \\ \nabla_y \bar{h}(x, y_\delta^*(x)) & 0 \end{bmatrix}. \tag{22}$$

We can notice that $H_{\delta=0} = H$, where $H$ is defined in (19).

Then, we have

$$\lim_{\delta \to 0} \frac{[\nabla_x g(x, y_\delta^*(x)) + \nabla_x h(x, y_\delta^*(x))\lambda_\delta^*(x)] - [\nabla_x g(x, y^*(x)) + \nabla_x h(x, y^*(x))\lambda^*(x)]}{\delta}$$

$$= \begin{bmatrix} \nabla_{xy}^2 g(x, y^*(x)) + \nabla_{xy}^2 \bar{h}(x, y^*(x))^\top \bar{\lambda}^*(x) \\ \nabla_x \bar{h}(x, y^*(x)) \end{bmatrix}^\top \begin{bmatrix} \frac{d}{d\delta} y_\delta^*(x) \\ \frac{d}{d\delta} \bar{\lambda}_\delta^*(x) \end{bmatrix} \Bigg|_{\delta=0}$$

$$= \begin{bmatrix} \nabla_{xy}^2 g(x, y^*(x)) + \nabla_{xy}^2 \bar{h}(x, y^*(x))^\top \bar{\lambda}^*(x) \\ \nabla_x \bar{h}(x, y^*(x)) \end{bmatrix}^\top H^{-1} \begin{bmatrix} -\nabla_y f(x, y^*(x)) \\ 0 \end{bmatrix}$$

$$= T^\top H^{-1} \begin{bmatrix} -\nabla_y f(x, y^*(x)) \\ 0 \end{bmatrix}$$

where we denote

$$T = \begin{bmatrix} \nabla^2_{xy} g(x, y^*(x)) + \nabla^2_{xy} \bar{h}(x, y^*(x))^\top \bar{\lambda}^*(x) \\ \nabla_x \bar{h}(x, y^*(x)) \end{bmatrix}.$$

Note that according to (20), we have

$$\nabla y^*(x) = - \begin{bmatrix} 1 & 0 \end{bmatrix} H^{-1} T$$

$$(\nabla y^*(x))^\top \nabla_y f(x, y^*(x)) = T^\top H^{-1} \begin{bmatrix} -\nabla_y f(x, y^*(x)) \\ 0 \end{bmatrix}$$

Thus,

$$\lim_{\delta \to 0} \frac{[\nabla_x g(x, y^*_\delta(x)) + \nabla_x h(x, y^*_\delta(x))\lambda^*_\delta(x)] - [\nabla_x g(x, y^*(x)) + \nabla_x h(x, y^*(x))\lambda^*(x)]}{\delta}$$

$$= (\nabla y^*(x))^\top \nabla_y f(x, y^*(x)) \tag{23}$$

Note that, according to Lemma 4.4, we have

$$\nabla \Phi_\delta(x) = \frac{[\nabla_x g(x, y^*_\delta(x)) + \nabla_x h(x, y^*_\delta(x))\lambda^*_\delta(x)] - [\nabla_x g(x, y^*(x)) + \nabla_x h(x, y^*(x))\lambda^*(x)]}{\delta}$$

$$+ \nabla_x f(x, y^*_\delta(x)).$$

Then, we consider to bound $\nabla \Phi_\delta(x) - \nabla \Phi(x)$.

$$\nabla \Phi_\delta(x) - \nabla \Phi(x)$$
$$= \nabla_x f(x, y^*_\delta(x)) - \nabla_x f(x, y^*(x))$$
$$\quad + \frac{[\nabla_x g(x, y^*_\delta(x)) + \nabla_x h(x, y^*_\delta(x))\lambda^*_\delta(x)] - [\nabla_x g(x, y^*(x)) + \nabla_x h(x, y^*(x))\lambda^*(x)]}{\delta}$$
$$\quad - (\nabla y^*(x))^\top \nabla_y f(x, y^*(x))$$
$$= \nabla_x f(x, y^*_\delta(x)) - \nabla_x f(x, y^*(x))$$
$$\quad + \frac{[\nabla_x g(x, y^*_\delta(x)) + \nabla_x h(x, y^*_\delta(x))\lambda^*_\delta(x)] - [\nabla_x g(x, y^*(x)) + \nabla_x h(x, y^*(x))\lambda^*(x)]}{\delta}$$
$$\quad - [\nabla^2_{xy} g(x, y^*(x)) + \nabla^2_{xy} \bar{h}(x, y^*(x))^\top \bar{\lambda}^*(x)] \frac{d}{d\delta} y^*_\delta(x)|_{\delta=0} - \nabla_x \bar{h}(x, y^*(x)) \frac{d}{d\delta} \bar{\lambda}^*_\delta(x)|_{\delta=0} \tag{24}$$

where the last equality is due to (23).

For the first term in (24), we have

$$\|\nabla_x f(x, y^*_\delta(x)) - \nabla_x f(x, y^*(x))\| \le l_{f,1} \|y^*_\delta(x) - y^*(x)\| \le \frac{2\delta l_{f,0} l_{f,1}}{\mu_g}$$

For the remaindering terms in (24), we have

$$
\frac{\nabla_x g(x, y_\delta^*(x)) - \nabla_x g(x, y^*(x))}{\delta} - \nabla_{xy}^2 g(x, y^*(x)) \frac{d}{d\delta} y_\delta^*(x)|_{\delta=0}
$$

$$
+ \frac{\nabla_x h(x, y_\delta^*(x)) \lambda_\delta^*(x) - \nabla_x h(x, y^*(x)) \lambda^*(x)}{\delta} - \nabla_{xy}^2 h(x, y^*(x)) \frac{d}{d\delta} \lambda_\delta^*(x)|_{\delta=0}
$$

$$
- \nabla_x \bar{h}(x, y^*(x)) \frac{d}{d\delta} \bar{\lambda}_\delta^*(x)|_{\delta=0}
$$

$$
= \frac{1}{\delta} \int_{t=0}^\delta \left( \nabla_{xy}^2 g(x, y_t^*(x)) \frac{d}{dt} y_t^*(x) - \nabla_{xy}^2 g(x, y^*(x)) \frac{d}{d\delta} y_\delta^*(x)|_{\delta=0} \right) dt
$$

$$
+ \frac{1}{\delta} \int_{t=0}^\delta \nabla_{xy}^2 h(x, y_t^*(x)) \frac{d}{dt} y_t^*(x) + \nabla_x h(x, y_t^*(x)) \frac{d}{dt} \lambda_t^*(x) dt
$$

$$
- \frac{1}{\delta} \int_{t=0}^\delta \left( \nabla_{xy}^2 h(x, y^*(x)) \frac{d}{ds} y_s^*(x) + \nabla_x h(x, y^*(x)) \frac{d}{ds} \lambda_s^*(x) \right) |_{s=0} dt
$$

$$
= \frac{1}{\delta} \int_{t=0}^\delta \left[ \nabla_{xy}^2 g(x, y_t^*(x)) - \nabla_{xy}^2 g(x, y^*(x)) \right] \frac{d}{dt} y_t^*(x) dt
$$

$$
+ \frac{1}{\delta} \int_{t=0}^\delta \nabla_{xy}^2 g(x, y^*(x)) \left[ \frac{d}{dt} y_t^*(x) - \frac{d}{ds} y_s^*(x)|_{s=0} \right] dt
$$

$$
+ \frac{1}{\delta} \int_{t=0}^\delta \left[ \nabla_{xy}^2 h(x, y_t^*(x)) - \nabla_{xy}^2 h(x, y^*(x)) \right] \frac{d}{dt} y_t^*(x) dt
$$

$$
+ \frac{1}{\delta} \int_{t=0}^\delta \nabla_{xy}^2 h(x, y^*(x)) \left[ \frac{d}{dt} y_t^*(x) - \frac{d}{ds} y_s^*(x)|_{s=0} \right] dt
$$

$$
+ \frac{1}{\delta} \int_{t=0}^\delta \left[ \nabla_x h(x, y_t^*(x)) - \nabla_x h(x, y^*(x)) \right] \frac{d}{dt} \lambda_t^*(x) dt
$$

$$
+ \frac{1}{\delta} \int_{t=0}^\delta \nabla_x h(x, y^*(x)) \left[ \frac{d}{dt} \lambda_t^*(x) - \lambda_s^*(x)|_{s=0} \right] dt \tag{25}
$$

For the first term in (25), we have

$$
\left\| \frac{1}{\delta} \int_{t=0}^\delta \left[ \nabla_{xy}^2 g(x, y_t^*(x)) - \nabla_{xy}^2 g(x, y^*(x)) \right] \frac{d}{dt} y_t^*(x) dt \right\|
$$

$$
\leq \frac{1}{\delta} \int_{t=0}^\delta \left\| \frac{d}{dt} y_t^*(x) \right\| \cdot l_{g,2} \| y_t^*(x) - y^*(x) \| \cdot dt
$$

$$
\leq \frac{1}{\delta} \int_{t=0}^\delta l_{g,2} C_y^2 \cdot t \cdot dt
$$

$$
= \delta l_{g,2} C_y^2 / 2
$$

where the last equality is due to Lemma C.1.

Similarly, for the third, fifth terms in (25), we have

$$
\left\| \frac{1}{\delta} \int_{t=0}^\delta \left[ \nabla_{xy}^2 h(x, y_t^*(x)) - \nabla_{xy}^2 h(x, y^*(x)) \right] \frac{d}{dt} y_t^*(x) dt \right.
$$

$$
\left. + \frac{1}{\delta} \int_{t=0}^\delta \left[ \nabla_x h(x, y_t^*(x)) - \nabla_x h(x, y^*(x)) \right] \frac{d}{dt} \lambda_t^*(x) dt \right\|
$$

$$
\leq \delta (l_{h,2} + l_{h,1}) C_y^2 / 2
$$

For the second term in (25), we have

$$\|\frac{1}{\delta}\int_{t=0}^{\delta}\nabla_{xy}^2(x,y^*(x))\cdot\left(\frac{d}{dt}y_t^*(x)-\frac{d}{ds}y_s^*(x)|_{s=0}\right)dt\|$$

$$\leq\frac{1}{\delta}\|\nabla_{xy}^2(x,y^*(x))\|\cdot\int_{t=0}^{\delta}\int_{s=0}^{t}\|\frac{d^2}{ds^2}y_s^*(x)\|dsdt$$

$$\leq\frac{1}{\delta}\|\nabla_{xy}^2(x,y^*(x))\|\cdot\max_{s\in[0,\delta]}\|\frac{d^2}{ds^2}y_s^*(x)\|\cdot\delta^2$$

$$\leq\delta l_{g,1}L_y$$

where the last equality is due to Lemma C.1.

Similarly, for the fourth, sixth terms in (25), we have

$$\left\|\frac{1}{\delta}\int_{t=0}^{\delta}\nabla_{xy}^2g(x,y^*(x))\left[\frac{d}{dt}y_t^*(x)-\frac{d}{ds}y_s^*(x)|_{s=0}\right]dt\right.$$

$$\left.+\frac{1}{\delta}\int_{t=0}^{\delta}\nabla_xh(x,y^*(x))\left[\frac{d}{dt}\lambda_t^*(x)-\lambda_s^*(x)|_{s=0}\right]dt\right\|$$

$$\leq\delta(l_{h,1}+l_{h,0})L_y$$

Therefore,

$$\|\nabla\Phi_\delta(x)-\nabla\Phi(x)\|\leq O(\delta).$$

$\square$

**Lemma C.1.** *When Assumption 3.1, 3.2, 3.3, 4.6 hold, and $\delta\leq\mu_g/(2l_{f,1})$, if Assumption 4.8 holds for a given $x$, we have*

$$\left\|\begin{bmatrix}\frac{d}{d\delta}y_\delta^*(x)\\\frac{d}{d\delta}\bar\lambda_\delta^*(x)\end{bmatrix}\right\|\leq C_y$$

$$\left\|\begin{bmatrix}\frac{d^2}{d\delta^2}y_\delta^*(x)\\\frac{d}{d\delta^2}\bar\lambda_\delta^*(x)\end{bmatrix}\right\|\leq L_y$$

*where $C_y,L_y$ are $O(1)$ constants.*

*Proof.* According to (21), we have

$$\begin{bmatrix}\frac{d}{d\delta}y_\delta^*(x)\\\frac{d}{d\delta}\bar\lambda_\delta^*(x)\end{bmatrix}=H_\delta^{-1}p_\delta,$$

where

$$H_\delta=\begin{bmatrix}\nabla_{yy}^2g_\delta(x,y_\delta^*(x))+\nabla_{yy}^2\bar h(x,y_\delta^*(x))^\top\bar\lambda_\delta^*(x)&\nabla_y\bar h(x,y_\delta^*(x))^\top\\\nabla_y\bar h(x,y_\delta^*(x))&0\end{bmatrix},$$

$$p_\delta=\begin{bmatrix}-\nabla_yf(x,y_\delta^*(x))\\0\end{bmatrix}.$$

According to Lemma C.2, we have

$$\left\|\begin{bmatrix}\frac{d}{d\delta}y_\delta^*(x)\\\frac{d}{d\delta}\bar\lambda_\delta^*(x)\end{bmatrix}\right\|\leq\left\|H_\delta^{-1}p_\delta\right\|\leq C_Hl_{f,0}=C_y.$$

Denote

$$\|H_{\delta_2}-H_{\delta_1}\|=\left\|\begin{bmatrix}B&D^T\\D&0\end{bmatrix}\right\|$$

where
$$
\begin{aligned}
B =& \nabla_{yy}^2 g_{\delta_2}(x, y_{\delta_2}^*(x)) + \nabla_{yy}^2 \bar{h}(x, y_{\delta_2}^*(x))^\top \bar{\lambda}_{\delta_2}^*(x) - \nabla_{yy}^2 g_{\delta_1}(x, y_{\delta_1}^*(x)) - \nabla_{yy}^2 \bar{h}(x, y_{\delta_1}^*(x))^\top \bar{\lambda}_{\delta_1}^*(x) \\
=& \delta_2 \nabla_{yy}^2 f(x, y_{\delta_2}^*(x)) - \delta_1 \nabla_{yy}^2 f(x, y_{\delta_1}^*(x)) + \nabla_{yy}^2 g(x, y_{\delta_2}^*(x)) - \nabla_{yy}^2 g(x, y_{\delta_1}^*(x)) \\
& + \nabla_{yy}^2 \bar{h}(x, y_{\delta_2}^*(x))^\top \bar{\lambda}_{\delta_2}^*(x) - \nabla_{yy}^2 \bar{h}(x, y_{\delta_1}^*(x))^\top \bar{\lambda}_{\delta_1}^*(x) \\
=& \delta_1 (\nabla_{yy}^2 f(x, y_{\delta_2}^*(x)) - \nabla_{yy}^2 f(x, y_{\delta_1}^*(x))) + \nabla_{yy}^2 f(x, y_{\delta_2}^*(x))(\delta_2 - \delta_1) + \nabla_{yy}^2 g(x, y_{\delta_2}^*(x)) \\
& - \nabla_{yy}^2 g(x, y_{\delta_1}^*(x)) + [\nabla_{yy}^2 \bar{h}(x, y_{\delta_2}^*(x)) - \nabla_{yy}^2 \bar{h}(x, y_{\delta_1}^*(x))]^\top \bar{\lambda}_{\delta_2}^*(x) \\
& + \nabla_{yy}^2 \bar{h}(x, y_{\delta_1}^*(x))^\top [\bar{\lambda}_{\delta_2}^*(x) - \bar{\lambda}_{\delta_1}^*(x)].
\end{aligned}
$$

Thus
$$
\|B\| \le \left( \frac{2\mu_g l_{f,2}}{l_{f,1}} C_y + l_{f,1} + l_{g,2} C_y + \Lambda l_{h,2} C_y + l_{f,1} C_y \right) |\delta_2 - \delta_1|
$$

$$
\|D\| = \|\nabla_y \bar{h}(x, y_{\delta_2}^*(x)) - \nabla_y \bar{h}(x, y_{\delta_1}^*(x))\| \le l_{h,1}(y_{\delta_2}^*(x) - y_{\delta_1}^*(x)) \le l_{h,1} C_y |\delta_1 - \delta_2|
$$

Therefore, we have
$$
\|H_{\delta_2} - H_{\delta_1}\| \le M_H |\delta_1 - \delta_2|
$$

Moreover, we have
$$
\begin{aligned}
& \left\| \begin{bmatrix} \frac{d}{d\delta} y_{\delta_1}^*(x) \\ \frac{d}{d\delta} \lambda_{\delta_1}^*(x) \end{bmatrix} - \begin{bmatrix} \frac{d}{d\delta} y_{\delta_2}^*(x) \\ \frac{d}{d\delta} \lambda_{\delta_2}^*(x) \end{bmatrix} \right\| \\
=& \|H_{\delta_1}^{-1} p_{\delta_1} - H_{\delta_2}^{-1} p_{\delta_2}\| \\
=& \|(H_{\delta_1}^{-1} - H_{\delta_2}^{-1}) p_{\delta_1} + H_{\delta_2}^{-1}(p_{\delta_1} - p_{\delta_2})\| \\
\le& \|(H_{\delta_1}^{-1}(H_{\delta_2} - H_{\delta_1}) H_{\delta_2}^{-1})\| \|p_{\delta_1}\| + \|H_{\delta_2}^{-1}\| \|p_{\delta_1} - p_{\delta_2}\| \\
\le& \left( C_H^2 l_{f,0} M_H^2 + C_H l_{f,1} \right) |\delta_1 - \delta_2| = L_y |\delta_1 - \delta_2|
\end{aligned}
$$
where $L_y = \left( C_H^2 l_{f,0} M_H^2 + C_H l_{f,1} \right)$.

$\square$

**Lemma C.2.** *For $H_\delta$ defined in (22), we have*
$$
\|H_\delta^{-1}\| \le C_H,
$$
*where $C_H$ is an $O(1)$ constant depending on $\mu_g, l_{g,1}, l_{f,1}, d_h, l_{h,1}, s_{\min}, s_{\max}, \Lambda$.*

*Proof.* Denote $A = \nabla_{yy}^2 g_\delta(x, y_\delta^*(x)) + \nabla_{yy}^2 \bar{h}(x, y_\delta^*(x))^\top \bar{\lambda}_\delta^*(x)$, $C = \nabla_y \bar{h}(x, y_\delta^*(x))$. We have

$$
H_{\delta_1}^{-1} = \begin{bmatrix} A^{-1} + A^{-1} C^\top (CA^{-1} C^\top)^{-1} CA^{-1} & -A^{-1} C^\top (CA^{-1} C^\top)^{-1} \\ -(CA^{-1} C^\top)^{-1} CA^{-1} & (CA^{-1} C^\top)^{-1} \end{bmatrix}.
$$

According to Assumption 4.8, we know that $\|\lambda_\delta^*(x)\| \le \Lambda$. Thus, $0 \le [\lambda_\delta^*(x)]_i \le \Lambda$. We have
$$
\|A^{-1}\| \le \frac{2}{\mu_g}
$$
$$
\|A\| \le L_g + d_h l_{h,1} \Lambda
$$
$$
\|C\| \le s_{\max}
$$
$$
\|C^{-1}\| \le \frac{1}{s_{\min}}
$$

Denote
$$
H_{\delta_1}^{-1} = \begin{bmatrix} B & D^T \\ D & E \end{bmatrix}.
$$

we have $\|B\| \le C_B, \|D\| \le C_D, \|E\| \le C_E$ and so that $\|H_{\delta_1}^{-1}\| \le C_H$, where $C_B, C_D, C_E, C_H$ are $O(1)$ constants depending on $\mu_g, l_{g,1}, l_{f,1}, d_h, l_{h,1}, s_{\min}, s_{\max}, \Lambda$.

$\square$

# D Proofs of Theorem 5.4 and Corollary 5.5

In this section, we provide proofs for Theorem 5.4 and Corollary 5.5. We first introduce the additional notations and lemmas that will be used in this section.

**Notations**

$$K(x, y', z', u, v) = \phi_\delta(x, y, z) + u^\top (A'y' - b) - v^\top (A'z' - b) + \frac{\rho_1}{2}\|A'y' - b\|^2 - \frac{\rho_2}{2}\|A'z' - b\|^2$$

$$L_K = l_\delta + 2l_{g,1} + \max\{\rho_1, \rho_2\}\sigma_{\max}^2(A')$$

$$\mu_y = \min\{\mu_g - l_\delta - \rho_1\sigma_{\max}^2(A), \frac{\rho_1}{2}\}$$

$$\mu_z = \min\{\mu_g - \rho_2\sigma_{\max}^2(A), \frac{\rho_2}{2}\}$$

$$y_\delta^*(x) = \min_{y \in \mathcal{Y}(x)} g_\delta(x, y)$$

$$z^*(x) = \operatorname*{argmin}_{z \in \mathcal{Y}(x)} g(x, z)$$

$$y_\delta'^*(x, u) = \operatorname*{argmin}_{y' \in \mathcal{P}_y} K(x, y', z', u, v)$$

$$z'^*(x, v) = \operatorname*{argmax}_{z' \in \mathcal{P}_y} K(x, y', z', u, v)$$

$$[y_\delta^*(x, u)^\top, \alpha_\delta^*(x, u)^\top]^\top = y_\delta'^*(x, u)$$

$$[z^*(x, v)^\top, \beta^*(x, v)^\top]^\top = z'^*(x, v)$$

$$d(x, z', u, v) = K(x, y_\delta'^*(x, u), z', u, v)$$

$$q(x, v) = \phi_\delta(x, y_\delta^*(x), z^*(x, v)) - v^\top (A'z'^*(x, v) - b) - \frac{\rho_2}{2}\|A'z'^*(x, v) - b\|^2$$

$$V_t = \frac{1}{4}K(x_t, y_t', z_t', u_t, v_t) + 2q(x_t, v_t) - d(x_t, z_t', u_t, v_t)$$

**Lemma D.1.** *When $\delta \le \mu_g/(2l_{f,1})$, $0 \le \rho_1 \le \frac{\mu_g - \delta l_{f,1}}{\sigma_{\max}^2(A)}$ and $0 \le \rho_2 \le \frac{\mu_g}{\sigma_{\max}^2(A)}$, $K(x, y', z', u, v)$ is $\mu_y$-strongly convex w.r.t. $y'$, $\mu_z$-strongly concave w.r.t. $z'$, and $L_K$-smooth w.r.t. $x, y', z'$.*

*Proof.* According to Lemma A.4, we know that $K(x, y', z', u, v)$ is $\mu_y$-strongly convex w.r.t. $y'$, $\mu_z$-strongly concave w.r.t. $z'$. Moreover

$$\nabla_x K(x, y', z', u, v) = \nabla_x \phi_\delta(x, y, z)$$
$$\nabla_y K(x, y', z', u, v) = \nabla_y \phi_\delta(x, y, z) + A^\top u + \rho_1 A^\top (A'y' - b)$$
$$\nabla_z K(x, y', z', u, v) = \nabla_z \phi_\delta(x, y, z) - A^\top v - \rho_2 A^\top (A'z' - b)$$

Thus, $K(x, y', z', u, v)$ is $L_K$-smooth w.r.t. $x, y', z'$. □

**Lemma D.2.** *When* $\delta \leq \mu_g/(2l_{f,1})$, $0 \leq \rho_1 \leq \frac{\mu_g - \delta l_{f,1}}{\sigma_{\max}^2(A)}$ *and* $0 \leq \rho_2 \leq \frac{\mu_g}{\sigma_{\max}^2(A)}$, $\eta_y, \eta_z \leq 1/L_K$, *we have*

$$\|y_\delta'^*(x, u_1) - y_\delta'^*(x, u_2)\| \leq \sigma_{yu}\|u_1 - u_2\| \tag{26}$$

$$\|y_\delta'^*(x_1, u) - y_\delta'^*(x_2, u)\| \leq \sigma_{yx}\|x_1 - x_2\| \tag{27}$$

$$\|z'^*(x, v_1) - z'^*(x, v_2)\| \leq \sigma_{zv}\|v_1 - v_2\| \tag{28}$$

$$\|z'^*(x_1, v) - z'^*(x_2, v)\| \leq \sigma_{zx}\|x_1 - x_2\| \tag{29}$$

$$\|y_\delta^*(x_1) - y_\delta^*(x_2)\| \leq \sigma_{ys}\|x_1 - x_2\| \tag{30}$$

$$\|z^*(x_1) - z^*(x_2)\| \leq \sigma_{zs}\|x_1 - x_2\| \tag{31}$$

$$\|y_\delta'^*(x, u) - y_\delta'^*(x)\| \leq \sigma_y\|A'y_\delta'^*(x, u) - b\| \tag{32}$$

$$\|z'^*(x, v) - z'^*(x)\| \leq \sigma_z\|A'z'^*(x, v) - b\| \tag{33}$$

$$\|y_\delta'^*(x, u) - y'\| \leq \sigma_{ye}\|\nabla_y K(x, y', z', u, v)\| + \sigma_\alpha\|\alpha - \Pi_-(\alpha - \eta_y\nabla_\alpha K(x, y', z', u, v))\| \tag{34}$$

$$\|z'^*(x, v) - z'\| \leq \sigma_{ze}\|\nabla_z K(x, y', z', u, v)\| + \sigma_\beta\|\beta - \Pi_-(\beta + \eta_z\nabla_\beta K(x, y', z', u, v))\| \tag{35}$$

$$\|y_\delta'^*(x_t, u_{t+1}) - y_t'\| \leq \frac{2}{\mu_y\eta_y}\|y_{t+1}' - y_t'\| \tag{36}$$

$$\|z'^*(x_t, v_{t+1}) - z_t'\| \leq \frac{2}{\mu_z\eta_z}\|z_{t+1}' - z_t'\| \tag{37}$$

*where* $\sigma_{yu} = \frac{\sigma_{\max}(A')}{\mu_y}$, $\sigma_{yx} = \frac{L_K}{\mu_y}$, $\sigma_{zv} = \frac{\sigma_{\max}(A')}{\mu_z}$, $\sigma_{zx} = \frac{L_K}{\mu_z}$, $\sigma_{ye} = \frac{2}{\mu_y}$, $\sigma_{ze} = \frac{2}{\mu_z}$, $\sigma_\alpha = \frac{2}{\mu_y\eta_y}$, $\sigma_\beta = \frac{2}{\mu_z\eta_z}$, $\sigma_{ys} = \frac{2L_K}{\mu_g}$, $\sigma_{zs} = \frac{L_K}{\mu_g}$,

$$\sigma_y = \frac{\sqrt{2}(\bar{\theta}L_K^2 + 1)}{\mu_y}$$

$$\sigma_z = \frac{\sqrt{2}(\bar{\theta}L_K^2 + 1)}{\mu_z}$$

$$M = \begin{pmatrix} A'^\top & G^\top \\ 0 & I \end{pmatrix}$$

$$G = \begin{pmatrix} 0_{d_y \times d_h} & 0 \\ 0 & I_{d_h} \end{pmatrix}$$

$$\bar{\theta} = \max_{\bar{M} \in \mathcal{B}(M)} \sigma_{\max}^2(\bar{M})/\sigma_{\min}^4(\bar{M})$$

$\bar{M}$ *is the set of all submatrices of* $M$ *with full row rank.*

*Proof.* (26), (27), (28), (29), (30), (31) is due to Lemma A.2. (34), (35), (36), (37) is due to Lemma A.1. (32), (33) is due to Lemma A.3. $\square$

### D.1 Potential function

In this subsection, we will prove the following descent lemma for $V_t$.

**Lemma D.3.** *When* $\delta \leq \mu_g/(2l_{f,1})$, $0 \leq \rho_1 \leq \frac{\mu_g - \delta l_{f,1}}{\sigma_{\max}^2(A)}$, $0 \leq \rho_2 \leq \frac{\mu_g}{\sigma_{\max}^2(A)}$, $\eta_y = 1/(4L_K)$, $\eta_z = 2/(L_K + 4L_d)$, $\eta_x = \min\{\eta_y\mu_y^2/(512L_\phi^2), \eta_z\mu_z^2/(96L_\phi^2), \eta_u/(64\sigma_y^2 L_\phi^2), \eta_v/(4\sigma_z^2 L_\phi^2), 2/(L_K + 4L_d + 8L_q)\}$, $\eta_u = \eta_y\mu_y^2/(32\sigma_{\max}^2(A))$, $\eta_v = \eta_z\mu_z^2/(32\sigma_{\max}^2(A))$, *we have*

$$V_t - V_{t+1} \geq \frac{1}{4\eta_x}\|x_{t+1} - x_t\|^2 + \frac{1}{16\eta_y}\|y_{t+1}' - y_t'\|^2 + \frac{1}{8\eta_z}\|z_{t+1}' - z_t'\|^2$$

$$+ \frac{\eta_u}{4}\|A'y_\delta'^*(x_t, u_{t+1}) - b\|^2 + \frac{\eta_v}{2}\|A'z'^*(x_t, v_{t+1}) - b\|^2$$

$$+ \frac{\eta_u}{4}\|A'y_t' - b\|^2 + \frac{\eta_v}{4}\|A'z_t' - b\|^2 + \frac{\eta_x}{4}\|\nabla\phi_\delta(x_t)\|^2$$

*Thus,*

$$\frac{1}{T}\sum_{t=0}^{T-1}\|\nabla\phi_\delta(x_t)\|^2 \leq \frac{4}{T\eta_x}(V_0 - \min_t V_t) \tag{38}$$

*Proof.* First, for function $d$, we have

$$d(x_t, z_t', u_{t+1}, v_{t+1}) - d(x_t, z_t', u_t, v_t)$$
$$= K(x_t, y_\delta'^*(x_t, u_{t+1}), z_t', u_{t+1}, v_{t+1}) - K(x_t, y_\delta'^*(x_t, u_t), z_t', u_t, v_t)$$
$$\geq K(x_t, y_\delta'^*(x_t, u_{t+1}), z_t', u_{t+1}, v_{t+1}) - K(x_t, y_\delta'^*(x_t, u_{t+1}), z_t', u_t, v_t)$$
$$= (u_{t+1} - u_t)^\top (A'y_\delta'^*(x_t, u_{t+1}) - b) - (v_{t+1} - v_t)^\top (A'z_t' - b)$$
$$= -\eta_v\|A'z_t' - b\|^2 + \eta_u(A'y_\delta'^*(x_t, u_{t+1}) - b)^\top(A'y_t' - b)$$

Note that

$$\nabla_x d(x, z', u, v) = \nabla_x\phi_\delta(x, y_\delta^*(x, u), z)$$
$$\nabla_{z'} d(x, z', u, v) = \begin{bmatrix} \nabla_z g(x, z) - A^\top v - \rho_2 A^\top(A'z' - b) \\ v + (A'z' - b) \end{bmatrix}$$

Thus, according to Lemma D.2, we know that $\nabla_x d(x, z', u, v)$ is $(L_\phi + L_\phi\sigma_{yx})$-continuous w.r.t. $x, z'$ and $\nabla'_z d(x, z', u, v)$ is $L_K$-continuous w.r.t. $x, z'$. Define $L_d = \max\{L_\phi + L_\phi\sigma_{yx}, L_K\}$. We have

$$d(x_{t+1}, z_{t+1}', u_{t+1}, v_{t+1}) - d(x_t, z_t', u_{t+1}, v_{t+1})$$
$$\geq \langle\nabla_x K(x_t, y_\delta'^*(x_t, u_{t+1}), z_t', u_{t+1}, v_{t+1}), x_{t+1} - x_t\rangle$$
$$\quad + \langle\nabla_{z'} K(x_t, y_\delta'^*(x_t, u_{t+1}), z_t', u_{t+1}, v_{t+1}), z_{t+1}' - z_t'\rangle$$
$$\quad - \frac{L_d}{2}(\|x_{t+1} - x_t\|^2 + \|z_{t+1}' - z_t'\|^2)$$
$$\geq \langle\nabla_x\phi_\delta(x_t, y_\delta^*(x_t, u_{t+1}), z_t), x_{t+1} - x_t\rangle + \frac{1}{\eta_z}\|z_{t+1}' - z_t'\|^2$$
$$\quad - \frac{L_d}{2}(\|x_{t+1} - x_t\|^2 + \|z_{t+1}' - z_t'\|^2)$$

Then, for function $q$, we have

$$q(x_t, v_t) - q(x_t, v_{t+1})$$
$$\geq K(x_t, y_\delta'^*(x_t), z'^*(x_t, v_t), u_t, v_t) - K(x_t, y_\delta'^*(x_t), z'^*(x_t, v_{t+1}), u_t, v_{t+1})$$
$$\geq K(x_t, y_\delta'^*(x_t), z'^*(x_t, v_{t+1}), u_t, v_t) - K(x_t, y_\delta'^*(x_t), z'^*(x_t, v_{t+1}), u_t, v_{t+1})$$
$$\geq \eta_v(A'z'^*(x_t, v_{t+1}) - b)^\top(A'z_t' - b)$$

Note that

$$\nabla_x q(x, v) = \nabla_x\phi_\delta(x, y_\delta^*(x), z^*(x, v))$$

Thus, according to Lemma D.2, $q(\cdot, v)$ is $L_q = (L_\phi + L_\phi\sigma_{zx} + L_\phi\sigma_{ys})$-smooth. We have

$$q(x_t, v_{t+1}) - q(x_{t+1}, v_{t+1})$$
$$\geq \langle\nabla_x\phi_\delta(x_t, y_\delta^*(x_t), z^*(x_t, v_{t+1})), x_t - x_{t+1}\rangle - \frac{L_q}{2}(\|x_{t+1} - x_t\|^2)$$

Finally, for function $K$, we have

$$K(x_t, y_t', z_t', u_t, v_t) - K(x_t, y_t', z_t', u_{t+1}, v_{t+1}) = -\eta_u\|A'y_t' - b\|^2 + \eta_v\|A'z_t' - b\|^2$$

and

$$K(x_t, y_t', z_t', u_{t+1}, v_{t+1}) - K(x_{t+1}, y_{t+1}', z_{t+1}', u_{t+1}, v_{t+1})$$
$$\geq \frac{1}{\eta_x}\|x_{t+1} - x_t\|^2 + \frac{1}{\eta_y}\|y_{t+1}' - y_t'\|^2 - \frac{1}{\eta_z}\|z_{t+1}' - z_t'\|^2$$
$$\quad - \frac{L_K}{2}(\|x_{t+1} - x_t\|^2 + \|y_{t+1}' - y_t'\|^2 + \|z_{t+1}' - z_t'\|^2).$$

Thus, for $V_t$, we have

$$
\begin{aligned}
&V_t - V_{t+1} \\
\geq & \langle \nabla_x \phi_\delta(x_t, y_\delta^*(x_t, u_{t+1}), z_t), x_{t+1} - x_t \rangle + \frac{1}{\eta_z} \|z_{t+1}' - z_t'\|^2 \\
& + \eta_u (A' y_\delta'^*(x_t, u_{t+1}) - b)^\top (A' y_t' - b) - \eta_v \|A' z_t' - b\|^2 - \frac{L_d}{2}(\|x_{t+1} - x_t\|^2 + \|z_{t+1}' - z_t'\|^2) \\
& + 2\langle \nabla_x \phi_\delta(x_t, y_\delta^*(x_t), z^*(x_t, v_{t+1})), x_t - x_{t+1} \rangle \\
& + 2\eta_v (A' z'^*(x_t, v_{t+1}) - b)^\top (A' z_t' - b) - L_q(\|x_{t+1} - x_t\|^2) \\
& + \frac{1}{4\eta_x} \|x_{t+1} - x_t\|^2 + \frac{1}{4\eta_y} \|y_{t+1}' - y_t'\|^2 - \frac{1}{4\eta_z} \|z_{t+1}' - z_t'\|^2 - \frac{\eta_u}{4} \|A' y_t' - b\|^2 + \frac{\eta_v}{4} \|A' z_t' - b\|^2 \\
& - \frac{L_K}{8}(\|x_{t+1} - x_t\|^2 + \|y_{t+1}' - y_t'\|^2 + \|z_{t+1}' - z_t'\|^2) \\
\geq & - \|\nabla_x \phi_\delta(x_t, y_\delta^*(x_t, u_{t+1}), z_t) - \nabla_x \phi_\delta(x_t, y_\delta^*(x_t), z^*(x_t, v_{t+1}))\| \|x_{t+1} - x_t\| \\
& + \left(\frac{\eta_u}{2} - \frac{\eta_u}{4}\right) \|A' y_t' - b\|^2 + \frac{\eta_u}{2} \|A' y_\delta'^*(x_t, u_{t+1}) - b\|^2 - \frac{\eta_u}{2} \|A' y_t' - A' y_\delta'^*(x_t, u_{t+1})\|^2 \\
& + \left(\eta_v + \frac{\eta_v}{4} - \eta_v\right) \|A' z_t' - b\|^2 + \eta_v \|A' z'^*(x_t, v_{t+1}) - b\|^2 - \eta_v \|A' z_t' - A' z'^*(x_t, v_{t+1})\|^2 \\
& - \frac{\eta_x}{2} \|\nabla_x \phi_\delta(x_t, y_\delta^*(x_t), z^*(x_t, v_{t+1})) - \nabla_x \phi_\delta(x_t, y_t, z_t)\|^2 \\
& + \frac{1}{2\eta_x} \|x_t - x_{t+1}\|^2 + \frac{\eta_x}{2} \|\nabla_x \phi_\delta(x_t, y_\delta^*(x_t), z^*(x_t, v_{t+1}))\|^2 \\
& + \left(\frac{1}{4\eta_x} - \frac{L_K}{8} - \frac{L_d}{2} - L_q\right) \|x_{t+1} - x_t\|^2 \\
& + \left(\frac{1}{\eta_z} - \frac{1}{4\eta_z} - \frac{L_K}{8} - \frac{L_d}{2}\right) \|z_{t+1}' - z_t'\|^2 \\
& + \left(\frac{1}{4\eta_y} - \frac{L_K}{8}\right) \|y_{t+1}' - y_t'\|^2 \\
\geq & - \frac{1}{4\eta_x} \|x_{t+1} - x_t\|^2 - 2\eta_x L_\phi^2 \|y_\delta'^*(x_t, u_{t+1}) - y_\delta'^*(x_t)\|^2 - 2\eta_x L_\phi^2 \|z'^*(x_t, v_{t+1}) - z_t'\|^2 \\
& + \frac{\eta_u}{4} \|A' y_t' - b\|^2 + \frac{\eta_u}{2} \|A' y_\delta'^*(x_t, u_{t+1}) - b\|^2 - \frac{\eta_u \sigma_{\max}^2(A)}{2} \|y_\delta'^*(x_t, u_{t+1}) - y_t'\|^2 \\
& + \frac{\eta_v}{4} \|A' z_t' - b\|^2 + \eta_v \|A' z'^*(x_t, v_{t+1}) - b\|^2 - \eta_v \sigma_{\max}^2(A) \|z'^*(x_t, v_{t+1}) - z_t'\|^2 \\
& - 2\eta_x L_\phi^2 \|y_\delta'^*(x_t) - y_\delta'^*(x_t, u_{t+1})\|^2 - 2\eta_x L_\phi^2 \|y_\delta'^*(x_t, u_{t+1}) - y_t'\|^2 - \eta_x L_\phi^2 \|z'^*(x_t, v_{t+1}) - z_t'\|^2 \\
& + \frac{\eta_x}{4} \|\nabla_x \phi_\delta(x_t, y_\delta^*(x_t), z^*(x_t))\|^2 - \frac{\eta_x L_\phi^2}{2} \|z'^*(x_t) - z'^*(x_t, v_{t+1})\|^2 + \frac{1}{2\eta_x} \|x_t - x_{t+1}\|^2 \\
& + \left(\frac{1}{4\eta_x} - \frac{L_K}{8} - \frac{L_d}{2} - L_q\right) \|x_{t+1} - x_t\|^2 \\
& + \left(\frac{3}{4\eta_z} - \frac{L_K}{8} - \frac{L_d}{2}\right) \|z_{t+1}' - z_t'\|^2 \\
& + \left(\frac{1}{4\eta_y} - \frac{L_K}{8}\right) \|y_{t+1}' - y_t'\|^2,
\end{aligned}
$$

and

$$V_t - V_{t+1}$$

$$\geq \left( \frac{1}{2\eta_x} + \frac{1}{4\eta_x} - \frac{1}{4\eta_x} - \frac{L_K}{8} - \frac{L_d}{2} - L_q \right) \|x_{t+1} - x_t\|^2$$

$$+ \left( \frac{1}{4\eta_y} - \frac{L_K}{8} \right) \|y'_{t+1} - y'_t\|^2 - \left( 4\eta_x L_\phi^2 + \frac{\eta_u \sigma_{\max}^2(A)}{2} \right) \|y_\delta'^*(x_t, u_{t+1}) - y'_t\|^2$$

$$+ \left( \frac{3}{4\eta_z} - \frac{L_K}{8} - \frac{L_d}{2} \right) \|z'_{t+1} - z'_t\|^2 - \left( 3\eta_x L_\phi^2 + \eta_v \sigma_{\max}^2(A) \right) \|z'^*(x_t, v_{t+1}) - z'_t\|^2$$

$$+ \frac{\eta_u}{4} \|A' y'_t - b\|^2 + \frac{\eta_u}{2} \|A' y_\delta'^*(x_t, u_{t+1}) - b\|^2 - 4\eta_x L_\phi^2 \|y_\delta'^*(x_t, u_{t+1}) - y_\delta'^*(x_t)\|^2$$

$$+ \frac{\eta_v}{4} \|A' z'_t - b\|^2 + \eta_v \|A' z'^*(x_t, v_{t+1}) - b\|^2 - \frac{\eta_x L_\phi^2}{2} \|z'^*(x_t, v_{t+1}) - z'^*(x_t)\|^2$$

$$+ \frac{\eta_x}{4} \|\nabla \phi_\delta(x_t)\|^2$$

$$\geq \left( \frac{1}{2\eta_x} - \frac{L_K}{8} - \frac{L_d}{2} - L_q \right) \|x_{t+1} - x_t\|^2$$

$$+ \left( \frac{1}{4\eta_y} - \frac{L_K}{8} - \frac{16\eta_x L_\phi^2 + 2\eta_u \sigma_{\max}^2(A)}{\mu_y^2 \eta_y^2} \right) \|y'_{t+1} - y'_t\|^2$$

$$+ \left( \frac{3}{4\eta_z} - \frac{L_K}{8} - \frac{L_d}{2} - \frac{12\eta_x L_\phi^2 + 4\eta_v \sigma_{\max}^2(A)}{\mu_z^2 \eta_z^2} \right) \|z'_{t+1} - z'_t\|^2$$

$$+ \left( \frac{\eta_u}{2} - 4\sigma_y^2 \eta_x L_\phi^2 \right) \|A' y_\delta'^*(x_t, u_{t+1}) - b\|^2$$

$$+ \left( \eta_v - \frac{\sigma_z^2 \eta_x L_\phi^2}{2} \right) \|A' z'^*(x_t, v_{t+1}) - b\|^2$$

$$+ \frac{\eta_u}{4} \|A' y'_t - b\|^2 + \frac{\eta_v}{4} \|A' z'_t - b\|^2 + \frac{\eta_x}{4} \|\nabla \phi_\delta(x_t)\|^2$$

where the last equality is due to Lemma D.2.

Thus, when $\delta \leq \mu_g/(2l_{f,1}), 0 \leq \rho_1 \leq \frac{\mu_g - \delta l_{f,1}}{\sigma_{\max}^2(A)}, 0 \leq \rho_2 \leq \frac{\mu_g}{\sigma_{\max}^2(A)}, \eta_y = 1/(4L_K), \eta_z = 2/(L_K + 4L_d), \eta_x = \min\{\eta_y \mu_y^2/(512L_\phi^2), \eta_z \mu_z^2/(96L_\phi^2), \eta_u/(64\sigma_y^2 L_\phi^2), \eta_v/(4\sigma_z^2 L_\phi^2), 2/(L_K + 4L_d + 8L_q)\}, \eta_u = \eta_y \mu_y^2/(32\sigma_{\max}^2(A)), \eta_v = \eta_z \mu_z^2/(32\sigma_{\max}^2(A))$, we have

$$V_t - V_{t+1}$$

$$\geq \frac{1}{4\eta_x} \|x_{t+1} - x_t\|^2 + \frac{1}{16\eta_y} \|y'_{t+1} - y'_t\|^2 + \frac{1}{8\eta_z} \|z'_{t+1} - z'_t\|^2$$

$$+ \frac{\eta_u}{4} \|A' y_\delta'^*(x_t, u_{t+1}) - b\|^2 + \frac{\eta_v}{2} \|A' z'^*(x_t, v_{t+1}) - b\|^2$$

$$+ \frac{\eta_u}{4} \|A' y'_t - b\|^2 + \frac{\eta_v}{4} \|A' z'_t - b\|^2 + \frac{\eta_x}{4} \|\nabla \phi_\delta(x_t)\|^2 \tag{39}$$

Thus,

$$\frac{1}{T} \sum_{t=0}^{T-1} \|\nabla \phi_\delta(x_t)\|^2 \leq \frac{4}{T\eta_x} (V_0 - \min_t V_t) \tag{40}$$

$\square$

**Proof of Theorem 5.4**

When $\delta \leq \mu_g/(2l_{f,1})$, $0 \leq \rho_1 \leq \frac{\mu_g - \delta l_{f,1}}{\sigma_{\max}^2(A)}$, $0 \leq \rho_2 \leq \frac{\mu_g}{\sigma_{\max}^2(A)}$, $\eta_y = 1/(4L_K)$, $\eta_z = 2/(L_K + 4L_d)$, $\eta_x = \min\{\eta_y\mu_y^2/(512L_\phi^2), \eta_z\mu_z^2/(96L_\phi^2), \eta_u/(64\sigma_y^2 L_\phi^2), \eta_v/(4\sigma_z^2 L_\phi^2), 2/(L_K + 4L_d + 8L_q)\}$, $\eta_u = \eta_y\mu_y^2/(32\sigma_{\max}^2(A))$, $\eta_v = \eta_z\mu_z^2/(32\sigma_{\max}^2(A))$, according to (40), we have

$$\frac{1}{T}\sum_{t=0}^{T-1}\|\nabla\phi_\delta(x_t)\|^2 \leq \frac{4}{T\eta_x}(V_0 - \min_t V_t) \tag{41}$$

Note that

$$V_t = \frac{1}{4}K(x_t, y_t', z_t', u_t, v_t) + 2q(x_t, v_t) - d(x_t, z_t', u_t, v_t)$$

$$\geq 2q(x_t, v_t) - \frac{3}{4}d(x_t, z_t', u_t, v_t) \geq \frac{5}{4}q(x_t, v_t) \geq \frac{5}{4}\phi_\delta(x_t) \geq \frac{5\delta\Phi^*}{4} - \frac{5\delta^2 l_{f,0}^2}{8\mu_g}$$

Therefore, when $\delta = \Theta(\epsilon)$, with $T = O(\epsilon^{-4})$, we have $t \in [T]$, such that $\|\nabla\Phi_\delta(x_t)\| = \|\frac{1}{\delta}\nabla\phi_\delta(x_t)\| \leq \epsilon$.

Moreover, if we have $x_0, y_0, z_0, u_0, v_0$ such that

$$\|y_0 - y_\delta^*(x_0)\| \leq O(\delta)$$
$$\|A'y_\delta'^*(x_0, u_0) - b\| \leq O(\delta)$$
$$\|A'y_0' - b\| \leq O(\delta)$$
$$\|z_0 - z^*(x_0)\| \leq O(\delta)$$
$$\|A'z'^*(x_0, v_0) - b\| \leq O(\delta)$$
$$\|A'z_0' - b\| \leq O(\delta)$$

Then, we have

$$V_0$$
$$= \frac{1}{4}\Bigg[\delta f(x_0, y_0) + (g(x_0, y_0) - g(x_0, z_0)) + u_0^\top(A'y_0' - b) - v_0^\top(A'z_0' - b)$$

$$+ \frac{\rho_1}{2}\|A'y_0' - b\|^2 - \frac{\rho_2}{2}\|A'z_0' - b\|^2\Bigg]$$

$$+ 2\Bigg[\delta f(x_0, y_\delta^*(x_0)) + (g(x_0, y_\delta^*(x_0)) - g(x_0, z^*(x_0, v_0))) - v_0^\top(A'z'^*(x_0, v_0) - b)$$

$$- \frac{\rho_2}{2}\|A'z'^*(x_0, v_0) - b\|^2\Bigg]$$

$$- \Bigg[\delta f(x_0, y_\delta^*(x_0, u_0)) + (g(x_0, y_\delta^*(x_0, u_0)) - g(x_0, z_0)) + u_0^\top(A'y_\delta'^*(x_0, u_0) - b)$$

$$- v_0^\top(A'z_0' - b) + \frac{\rho_1}{2}\|A'y_\delta'^*(x_0, u_0) - b\|^2 - \frac{\rho_2}{2}\|A'z_0' - b\|^2\Bigg]$$

$$\leq \frac{5\delta\Phi(x_0)}{4} + O(1)l_{f,1}(\|y_0 - y^*(x_0)\| + \|y_\delta^*(x_0) - y^*(x_0)\| + \|y_\delta^*(x_0, u_0) - y^*(x_0)\|)$$

$$+ O(1)C_g(\|z_0 - y_0\| + \|y_\delta^*(x_0) - z^*(x_0, v_0)\| + \|y_\delta^*(x_0, u_0) - z_0\|)$$

$$+ O(\|A'z_0' - b\| + \|A'z_0' - b\|^2 + \|A'y_0' - b\| + \|A'y_0' - b\|^2$$

$$+ \|A'z'^*(x_0, v_0) - b\| + \|A'z'^*(x_0, v_0) - b\|^2 + \|A'y_\delta'^*(x_0, u_0) - b\| + \|A'y_\delta'^*(x_0, u_0) - b\|^2)$$

$$\leq \frac{5\delta\Phi(x_0)}{4} + O(\delta)$$

where $G = \max\{g(x_0, y_0), g(x_0, z_0), g(x_0, y_\delta^*(x_0)), g(x_0, z^*(x_0, v_0)), g(x_0, y_\delta^*(x_0, u_0))\}$, $\mathcal{C} = \{y \in \mathbb{R}^{d_y}|g(x_0, y) \leq G\}$, $C_g = \sup_{y \in \mathcal{C}} \nabla_y g(x_0, y)$. Since $g(x_0, y)$ is strongly convex w.r.t $y$, its sub-level

set $\mathcal{C}$ is compact and convex. Moreover, since $g$ is Lipschitz smoothness, its gradient in this compact set $\mathcal{C}$ is upper bounded by an $O(1)$ constant $C_g$.

We can notice that

$$V_0 - \min_t V_t \leq \frac{5\delta[\Phi(x_0) - \Phi^*]}{4} + O(\delta) = O(\epsilon)$$

and

$$\frac{1}{T} \sum_{t=0}^{T-1} \|\nabla\Phi_\delta(x_t)\|^2 = \frac{1}{T\delta^2} \sum_{t=0}^{T-1} \|\nabla\phi_\delta(x_t)\|^2 \leq \frac{4}{T\eta_x\delta^2}(V_0 - \min_t V_t) = O\left(\frac{\epsilon^{-1}}{T}\right).$$

Therefore, we can find an $\epsilon$-stationary point of $\Phi_\delta(x)$ with a complexity of $O(\epsilon^{-3})$.

**Proof of Corollary 5.5**

For fixed $x_0$, define $W_t = \frac{1}{4}K(x_0, y'_t, z'_t, u_t, v_t) + 2q(x_0, v_t) - d(x_0, z'_t, u_t, v_t)$. According to (39), with appropriate parameters, we have

$$W_t - W_{t+1}$$
$$\geq \frac{1}{16\eta_y}\|y'_{t+1} - y'_t\|^2 + \frac{1}{8\eta_z}\|z'_{t+1} - z'_t\|^2$$
$$+ \frac{\eta_u}{4}\|A'y'^*_\delta(x_0, u_{t+1}) - b\|^2 + \frac{\eta_v}{2}\|A'z'^*(x_0, v_{t+1}) - b\|^2$$
$$+ \frac{\eta_u}{4}\|A'y'_t - b\|^2 + \frac{\eta_v}{4}\|A'z'_t - b\|^2$$

Thus, when $T = O(\epsilon^2)$, we can find $t \in T$ such that

$$\|y'_{t+1} - y'_t\| \leq \delta$$
$$\|A'y'^*_\delta(x_0, u_{t+1}) - b\| \leq \delta$$
$$\|A'y'_t - b\| \leq \delta$$
$$\|z'_{t+1} - z'_t\| \leq \delta$$
$$\|A'z'^*(x_0, v_{t+1}) - b\| \leq \delta$$
$$\|A'z'_t - b\| \leq \delta$$

Denote the active set at $y^*_\delta(x_0)$ as $\mathcal{I}^\alpha$, $\mathcal{J}^\alpha = [d_h]/\mathcal{I}$. Define $\Delta^\alpha = \min_{i\in\mathcal{J}^\alpha}|[\alpha^*_\delta(x_0)]_i|$. Denote the active set of $z^*(x_0)$ as $\mathcal{I}^\beta$, $\mathcal{J}^\beta = [d_h]/\mathcal{I}$. Define $\Delta^\beta = \min_{i\in\mathcal{J}^\beta}|[\beta^*(x_0)]_i|$. Set $\epsilon \leq \min\{\Delta^\alpha/(6\sigma_\alpha), \Delta^\alpha/(6\sigma_y), \Delta^\beta/(6\sigma_\beta), \Delta^\beta/(6\sigma_z)\}$. According to Lemma D.4, we have

$$\|u_{t+1} - u^*_\delta(x_0)\| \leq O(\delta)$$
$$\|v_{t+1} - v^*(x_0)\| \leq O(\delta)$$

Then, if we set $y'_0 = y'_t$, $z'_0 = z'_t$, $u_0 = u_{t+1}$, $v_0 = v_{t+1}$, with $x_0, y'_0, z'_0, u_0, v_0$ as initial points, according to Theorem 5.4, we can find an $\epsilon$-stationary point of $\Phi_\delta$ with a complexity of $O(\epsilon^{-3})$.

Thus, the total complexity is $O(\epsilon^{-3} + \epsilon^{-2}) = O(\epsilon^{-3})$.

**Lemma D.4.** *For a fixed $x_0$, denote the active set at $y^*_\delta(x_0)$ as $\mathcal{I}^\alpha$, $\mathcal{J}^\alpha = [d_h]/\mathcal{I}$. We have $[\alpha^*_\delta(x_0)]_{\mathcal{I}^\alpha} = 0$ and $[\alpha^*_\delta(x_0)]_{\mathcal{J}^\alpha} < 0$. Suppose $s^\alpha_{\min} = \sigma_{\min}(A_{\mathcal{I}^\alpha}) > 0$. Define $\Delta^\alpha = \min_{i\in\mathcal{J}^\alpha}|[\alpha^*_\delta(x_0)]_i|$. When $\|y'_{t+1} - y'_t\| \leq \Delta^\alpha/(6\sigma_\alpha), \|A'y'^*_\delta(x_0, u_{t+1}) - b\| \leq \Delta^\alpha/(6\sigma_y)$, we have*

$$\|u_{t+1} - u^*_\delta(x_0)\| \leq \sigma_{uyx0}\|y'_{t+1} - y'_t\| + \sigma_{u2x0}\|A'y'^*_\delta(x_0, u_{t+1}) - b\| + \sigma_{u1x0}\|A'y'_t - b\|$$

*where $\sigma_{uyx0} = \frac{1}{\eta_y} + \frac{1+\sigma_{\max}}{\eta_y s^\alpha_{\min}} + \frac{L_g\sigma_\alpha}{s^\alpha_{\min}}$, $\sigma_{u2x0} = \frac{L_g\sigma_y}{s^\alpha_{\min}}$, $\sigma_{u1x0} = \frac{\rho_1\sigma_{\max}(A)}{s^\alpha_{\min}}$.*

*Similarly, for a fixed $x_0$, denote the active set of $z^*(x_0)$ as $\mathcal{I}^\beta$, $\mathcal{J}^\beta = [d_h]/\mathcal{I}$. Suppose $s^\beta_{\min} = \sigma_{\min}(A_{\mathcal{I}^\beta}) > 0$. Define $\Delta^\beta = \min_{i\in\mathcal{J}^\beta}|[\beta^*(x_0)]_i|$. When $\|z'_{t+1} - z'_t\| \leq \Delta^\beta/(6\sigma_\beta), \|A'z'^*_\delta(x_0, v_{t+1}) - b\| \leq \Delta^\beta/(6\sigma_z)$, we have*

$$\|v_{t+1} - v^*(x_0)\| \leq \sigma_{uzx0}\|z'_{t+1} - z'_t\| + \sigma_{v2x0}\|A'z'^*(x_0, v_{t+1}) - b\| + \sigma_{v1x0}\|A'z'_t - b\|$$

*where $\sigma_{uzx0} = \frac{1}{\eta_z} + \frac{1+\sigma_{\max}}{\eta_z s^\beta_{\min}} + \frac{l_{g,1}\sigma_\beta}{s^\beta_{\min}}$, $\sigma_{v2x0} = \frac{l_{g,1}\sigma_z}{s^\beta_{\min}}$, $\sigma_{v1x0} = \frac{\rho_2\sigma_{\max}(A)}{s^\beta_{\min}}$.*

*Proof.* Denote the active set of $y_\delta^*(x_0)$ as $\mathcal{I}$, $\mathcal{J} = [d_h]/\mathcal{I}$. We have $[\alpha_\delta^*(x_0)]_\mathcal{I} = 0$ and $[\alpha_\delta^*(x_0)]_\mathcal{J} < 0$. Note that

$$\|\alpha_t - \alpha_\delta^*(x_0)\| \leq \|y_t' - y_\delta'^*(x_0)\| \leq \|y_t - y_\delta^*(x_0, u_{t+1})\| + \|y_\delta^*(x_0) - y_\delta^*(x_0, u_{t+1})\|$$
$$\leq \sigma_\alpha \|y_{t+1}' - y_t'\| + \sigma_y \|A'y_\delta'^*(x_0, u_{t+1}) - b\|$$

Define $\Delta = \min_{i \in \mathcal{J}} |[\alpha_\delta^*(x_0)]_i|$. When $\|y_{t+1}' - y_t'\| \leq \Delta/(6\sigma_\alpha)$, $\|A'y_\delta'^*(x_0, u_{t+1}) - b\| \leq \Delta/(6\sigma_y)$, we have

$$\|\alpha_t - \alpha_\delta^*(x_0)\| \leq \frac{1}{3}\Delta$$
$$\|\alpha_{t+1} - \alpha_t\| \leq \frac{1}{6}\Delta$$

Thus, for $\mathcal{J}$, we have

$$[\alpha_t]_\mathcal{J} < 0$$
$$[\alpha_{t+1}]_\mathcal{J} < 0$$

Therefore, there are no projection in the update of $\alpha_{t+1}$ and we have

$$\|[u_{t+1}]_\mathcal{J} - [u_\delta^*(x_0)]_\mathcal{J}\| = \|[u_{t+1}]_\mathcal{J}\| = \|\frac{1}{\eta_y}([\alpha_{t+1}]_\mathcal{J} - [\alpha_t]_\mathcal{J})\| \leq \frac{1}{\eta_y}\|\alpha_{t+1} - \alpha_t\|$$

Moreover, for $\mathcal{I}$, we have

$$\nabla_y g_\delta(x_0, y_\delta^*(x_0)) + A_\mathcal{I}^\top [u_\delta^*(x_0)]_\mathcal{I} = 0$$

Thus,

$$\nabla_y g_\delta(x_0, y_t) + A_\mathcal{I}^\top [u_{t+1}]_\mathcal{I} + A_\mathcal{J}^\top [u_{t+1}]_\mathcal{J} + \rho_1 A^\top (A'y_t' - b) = \frac{1}{\eta_y}(y_{t+1} - y_t)$$

Suppose $\sigma_{\min}(A_\mathcal{I}) = s_{\min}$, we have

$$\|[u_{t+1}]_\mathcal{I} - [u_\delta^*(x_0)]_\mathcal{I}\|$$
$$\leq \frac{1}{s_{\min}}[\frac{1}{\eta_y}\|y_{t+1} - y_t\| + L_g\|y_t - y_\delta^*(x_0)\| + \rho_1\sigma_{\max}(A)\|A'y_t' - b\| + \sigma_{\max}(A)\|u_{t+1}]_\mathcal{J}\|]$$
$$\leq \frac{1}{s_{\min}}\left[\left(\frac{1 + \sigma_{\max}}{\eta_y} + L_g\sigma_\alpha\right)\|y_{t+1}' - y_t'\| + L_g\sigma_y\|A'y_\delta'^*(x_0, u_{t+1}) - b\|\right.$$
$$\left. + \rho_1\sigma_{\max}(A)\|A'y_t' - b\|\right]$$

Thus,

$$\|u_{t+1} - u_\delta^*(x_0)\| \leq \sigma_{uyx0}\|y_{t+1}' - y_t'\| + \sigma_{u2x0}\|A'y_\delta'^*(x_0, u_{t+1}) - b\| + \sigma_{u1x0}\|A'y_t' - b\|$$

where $\sigma_{uyx0} = \frac{1}{\eta_y} + \frac{1 + \sigma_{\max}}{\eta_y s_{\min}} + \frac{L_g\sigma_\alpha}{s_{\min}}$, $\sigma_{u2x0} = \frac{L_g\sigma_y}{s_{\min}}$, $\sigma_{u1x0} = \frac{\rho_1\sigma_{\max}(A)}{s_{\min}}$

Similar conditions and conclusions also hold for $\|v_{t+1} - v^*(x_0)\|$.

$\square$

# E   Proofs of Theorem 5.2 and Corollary 5.3

In this section, we provide proofs for Theorem 5.2 and Corollary 5.3. We first introduce the additional notations and lemmas that will be used in this section.

**Notations**

$$K(x, y', z', u, v) = \phi_\delta(x, y, z) + u^\top(Bx + A'y' - b) - v^\top(Bx + A'z' - b)$$

$$+ \frac{\rho_1}{2}\|Bx + A'y' - b\|^2 - \frac{\rho_2}{2}\|Bx + A'z' - b\|^2$$

$$L_K = l_\delta + 2l_{g,1} + \max\{\rho_1, \rho_2\}\max\{\sigma_{\max}^2(A'), \sigma_{\max}^2(B), \sigma_{\max}(B)\sigma_{\max}(A')\}$$

$$\mu_y = \min\{\mu_g - l_\delta - \rho_1\sigma_{\max}^2(A), \frac{\rho_1}{2}\}$$

$$\mu_z = \min\{\mu_g - \rho_2\sigma_{\max}^2(A), \frac{\rho_2}{2}\}$$

$$y_\delta^*(x) = \min_{y \in \mathcal{Y}(x)} g_\delta(x, y)$$

$$z^*(x) = \operatorname*{argmin}_{z \in \mathcal{Y}(x)} g(x, z)$$

$$u_\delta^*(x) = \operatorname*{argmax}_{u \in \mathbb{R}_+} \min_{y \in \mathcal{Y}(x)} g_\delta(x, y) + u^\top(Bx + Ay - b)$$

$$v^*(x) = \operatorname*{argmax}_{v \in \mathbb{R}_+} \min_{z \in \mathcal{Y}(x)} g(x, y) + v^\top(Bx + Az - b)$$

$$y_\delta'^*(x, u) = \operatorname*{argmin}_{y' \in \mathcal{P}_y} K(x, y', z', u, v)$$

$$z'^*(x, v) = \operatorname*{argmax}_{z' \in \mathcal{P}_y} K(x, y', z', u, v)$$

$$[y_\delta^*(x, u)^\top, \alpha_\delta^*(x, u)^\top]^\top = y_\delta'^*(x, u)$$

$$[z^*(x, v)^\top, \beta^*(x, v)^\top]^\top = z'^*(x, v)$$

$$d(x, z', u, v) = K(x, y_\delta'^*(x, u), z', u, v)$$

$$q(x, v) = \phi_\delta(x, y_\delta^*(x), z^*(x, v)) - v^\top(Bx + A'z'^*(x, v) - b) - \frac{\rho_2}{2}\|Bx + A'z'^*(x, v) - b\|^2$$

$$V_t = \frac{1}{4}K(x_t, y_t', z_t', u_t, v_t) + 2q(x_t, v_t) - d(x_t, z_t', u_t, v_t)$$

**Lemma 5.1**

*When the LICQ condition (Definition 4.3) holds for $y^*(x)$ and $y_\delta^*(x)$, the optimal Lagrangian multipliers of $y^*(x)$ and $y_\delta^*(x)$ are unique and we have*

$$u_\delta^*(x) = \operatorname*{argmax}_{u \in \mathbb{R}_+} \min_{y \in \mathcal{Y}(x)} g_\delta(x, y) + u^\top h(x, y) = \operatorname*{argmax}_{u \in \mathbb{R}^{d_h}} \min_{y' \in \mathcal{P}_y} K(x, y', z', u, v),$$

$$v^*(x) = \operatorname*{argmax}_{v \in \mathbb{R}_+} \min_{z \in \mathcal{Y}(x)} g(x, z) + v^\top h(x, z) = \operatorname*{argmin}_{v \in \mathbb{R}^{d_h}} \max_{z' \in \mathcal{P}_y} K(x, y', z', u, v).$$

*Proof.* Suppose

$$v_1 = \operatorname*{argmax}_{v \in \mathbb{R}_+} \min_{z \in \mathcal{Y}(x)} g(x, y),$$

$$v_2 = \operatorname*{argmax}_{u \in \mathbb{R}^{d_h}} \min_{y' \in \mathcal{P}_y} K(x, y', z', u, v).$$

The KKT conditions for $v_1$ are

$$\nabla_y g(x, z^*(x)) + A^\top v_1 = 0$$
$$Bx + Az^*(x) - b \le 0$$
$$v_1 \ge 0$$
$$v_1^\top(Bx + Az^*(x) - b) = 0$$

The KKT conditions for $v_2$ are

$$\nabla_y g(x, z^*(x)) + A^\top v_1 + \rho_2(Bx + A'z'^*(x) - b) = 0$$
$$Bx + A'z'^*(x) - b = 0$$
$$\beta \le 0$$
$$v_1 \in \partial I_-(\beta)$$

Note that these two KKT conditions are equivalent. Moreover, since the LICQ condition (Definition 4.3) holds for $z^*(x)$, we have $v_1 = v_2$. Similar conditions and conclusions also hold for $u_\delta^*(x)$. $\qquad\square$

**Lemma E.1.** $K(x, y', z', u, v)$ *is* $\mu_y$-*strongly w.r.t.* $y'$, $\mu_z$-*strongly concave w.r.t.* $z'$, *and* $L_K$-*smooth w.r.t.* $x, y', z'$.

*Proof.* According to Lemma A.4, we know that $K(x, y', z', u, v)$ is $\mu_y$-strongly convex w.r.t. $y'$, $\mu_z$-strongly concave w.r.t. $z'$. Moreover

$$\nabla_x K(x, y', z', u, v) = \nabla_x \phi_\delta(x, y, z) + B^\top(u - v) + \rho_1 B^\top(Bx + A'y' - b)$$
$$- \rho_2 B^\top(Bx + A'z' - b)$$
$$\nabla_y K(x, y', z', u, v) = \nabla_y \phi_\delta(x, y, z) + A^\top u + \rho_1 A^\top(A'y' - b)$$
$$\nabla_z K(x, y', z', u, v) = \nabla_z \phi_\delta(x, y, z) - A^\top v - \rho_2 A^\top(A'z' - b)$$

Thus, $K(x, y', z', u, v)$ is $L_K$-smooth w.r.t. $x, y', z'$. $\qquad\square$

**Lemma E.2.**

$$\|u_{t+1} - u_\delta^*(x_t)\| \le \sigma_{uy}\|y'_{t+1} - y'_t\| + \sigma_{u1}\|Bx_t + A'y'_t - b\| + \sigma_{u2}\|Bx_t + A'y_\delta'^*(x_t, u_{t+1}) - b\|$$
$$\|v_{t+1} - v^*(x_t)\| \le \sigma_{vz}\|z'_{t+1} - z'_t\| + \sigma_{v1}\|Bx_t + A'z'_t - b\| + \sigma_{v2}\|Bx_t + A'z'^*(x_t, v_{t+1}) - b\|$$

*where* $\sigma_{uy} = \frac{1}{\sigma_{\min}(A)}\left(\frac{1}{\eta_y} + \sigma_\alpha l_{g,1}\right)$, $\sigma_{u1} = \rho_1$, $\sigma_{u2} = \frac{\sigma_y l_{g,1}}{\sigma_{\min}(A)}$, $\sigma_{vz} = \frac{1}{\sigma_{\min}(A)}\left(\frac{1}{\eta_z} + \sigma_\beta l_{g,1}\right)$, $\sigma_{v1} = \rho_2$, $\sigma_{v2} = \frac{\sigma_z l_{g,1}}{\sigma_{\min}(A)}$

*Proof.* By the optimality condition at $y_\delta^*(x_t)$, we have

$$\nabla_y g_\delta(x_t, y_\delta^*(x_t)) + A^\top u_\delta^*(x_t) = 0.$$

The update rule of $y_t$:

$$-\frac{y_{t+1} - y_t}{\eta_y} = \nabla_y g_\delta(x_t, y_t) + A^\top u_{t+1} + \rho_1 A^\top(Bx_t + Ay_t - b - \alpha_t),$$

Putting together, we have

$$A^\top(u_\delta^*(x_t) - u_{t+1})$$
$$= \frac{y_{t+1} - y_t}{\eta_y} + \nabla_y g_\delta(x_t, y_t) - \nabla_y g_\delta(x_t, y_\delta^*(x_t)) + \rho_1 A^\top(Bx_t + Ay_t - b - \alpha_t)$$

Since $A$ has full row rank, we have

$$u_\delta^*(x_t) - u_{t+1} = (AA^\top)^{-1} A\left[\frac{y_{t+1} - y_t}{\eta_y} + \nabla_y g_\delta(x_t, y_t) - \nabla_y g_\delta(x_t, y_\delta^*(x_t))\right]$$
$$+ \rho_1(Bx_t + Ay_t - b - \alpha_t),$$

and

$$\|u_{t+1} - u_\delta^*(x_t)\|$$
$$\le \frac{1}{\sigma_{\min}(A)\eta_y}\|y_{t+1} - y_t\| + \frac{l_{g,1}}{\sigma_{\min}(A)}\|y_t - y_\delta^*(x_t)\| + \rho_1\|Bx_t + Ay_t - b - \alpha_t\|$$

Moreover,
$$\|y_t - y_\delta^*(x_t)\| \leq \|y_t - y_\delta^*(x_t, u_{t+1})\| + \|y_\delta^*(x_t) - y_\delta^*(x_t, u_{t+1})\|$$
$$\leq \sigma_\alpha \|y_{t+1}' - y_t'\| + \sigma_y \|Bx_t + A'y_\delta'^*(x_t, u_{t+1}) - b\|$$

where the last equality is due to Lemma D.2.

Thus,
$$\|u_{t+1} - u_\delta^*(x_t)\| \leq \sigma_{uy}\|y_{t+1}' - y_t'\| + \sigma_{u1}\|Bx_t + A'y_t' - b\| + \sigma_{u2}\|Bx_t + A'y_\delta'^*(x_t, u_{t+1}) - b\|$$

where $\sigma_{uy} = \frac{1}{\sigma_{\min}(A)}(\frac{1}{\eta_y} + \sigma_\alpha l_{g,1})$, $\sigma_{u1} = \rho_1$, $\sigma_{u2} = \frac{\sigma_y l_{g,1}}{\sigma_{\min}(A)}$.

Similarly, we have
$$\|v_{t+1} - v^*(x_t)\| \leq \sigma_{vz}\|z_{t+1}' - z_t'\| + \sigma_{v1}\|Bx_t + A'z_t' - b\| + \sigma_{v2}\|Bx_t + A'z'^*(x_t, v_{t+1}) - b\|$$

where $\sigma_{vz} = \frac{1}{\sigma_{\min}(A)}(\frac{1}{\eta_z} + \sigma_\beta l_{g,1})$, $\sigma_{v1} = \rho_2$, $\sigma_{v2} = \frac{\sigma_z l_{g,1}}{\sigma_{\min}(A)}$.

$\square$

**Lemma E.3.**
$$\|z'^*(x_1) - z'^*(x_2)\| \leq \sigma_{zb}\|x_1 - x_2\|$$
$$\|y_\delta'^*(x_1) - z_\delta'^*(x_2)\| \leq \sigma_{yb}\|x_1 - x_2\|$$
$$\|v^*(x_1) - v^*(x_2)\| \leq \sigma_{vb}\|x_1 - x_2\|$$
$$\|u_\delta^*(x_1) - u_\delta^*(x_2)\| \leq \sigma_{ub}\|x_1 - x_2\|$$

*where* $\sigma_{zb} = \sigma_{\max}(A)\sigma_z + \sigma_{zx}$, $\sigma_{yb} = \sigma_{\max}(A)\sigma_y + \sigma_{yx}$, $\sigma_{vb} = \frac{l_{g,1}\sigma_{zb}}{\sigma_{\min}(A)}$, $\sigma_{ub} = \frac{L_g\sigma_{yb}}{\sigma_{\min}(A)}$.

*Proof.* Here, we introduce an additional notation: $z'^*(x; w) = \arg\min_{z' \in \mathcal{G}(w)} g(x, z) + \frac{\rho_2}{2}\|Bx + A'z' - b\|^2$, where $\mathcal{G}(w) = \{z' \in \mathcal{P}_y | Bw + A'z' - b = 0\}$. We can notice that $z'^*(x; x) = z'^*(x)$. According to Lemma A.3, we have $\|z'^*(x_1; x_1) - z'^*(x_1; x_2)\| \leq \sigma_z\|Ax_1 - Ax_2\|$. Moreover, we have $\|z'^*(x_1; x_2) - z'^*(x_2; x_2)\| \leq \sigma_{zx}\|x_1 - x_2\|$. Thus, $\|z'^*(x_1) - z'^*(x_2)\| \leq [\sigma_{\max}(A)\sigma_z + \sigma_{zx}]\|x_1 - x_2\|$. Moreover,
$$\nabla_y g(x, z^*(x_1)) + A^\top v^*(x_1) = 0$$
$$\nabla_y g(x, z^*(x_2)) + A^\top v^*(x_2) = 0$$

Thus,
$$\|v^*(x_1) - v^*(x_2)\| \leq \frac{l_{g,1}\sigma_{zb}}{\sigma_{\min}(A)}\|x_1 - x_2\|$$

Similarly, we have $\|y_\delta'^*(x_1) - y_\delta'^*(x_2)\| \leq [\sigma_{\max}(A)\sigma_y + \sigma_{yx}]\|x_1 - x_2\|$ and $\|u_\delta^*(x_1) - u_\delta^*(x_2)\| \leq \frac{L_g\sigma_{yb}}{\sigma_{\min}(A)}\|x_1 - x_2\|$ $\square$

### E.1 Potential function

In this subsection, we will prove the following descent lemma for $V_t$.

**Lemma E.4.** *When* $\delta \leq \mu_g/(2l_{f,1})$, $0 \leq \rho_1 \leq \frac{\mu_g - \delta l_{f,1}}{\sigma_{\max}^2(A)}$, $0 \leq \rho_2 \leq \frac{\mu_g}{\sigma_{\max}^2(A)}$, $\eta_y = 1/(4L_K)$, $\eta_z = 2/(L_K + 4L_d)$, $\eta_x = \min\{\eta_y\mu_y^2/(640L_K^2), \eta_z\mu_z^2/(640L_K^2), \eta_u/(240(\sigma_y^2 + \sigma_{u2}^2 + \sigma_{u1}^2)L_K^2), \eta_v/(240(\sigma_z^2 + \sigma_{v2}^2 + \sigma_{v1}^2)L_K^2), 2/(L_K + 4L_d + 8L_q), 1/(1920\eta_y L_K^2\sigma_{uy}^2), 1/(1920\eta_z L_K^2\sigma_{uz}^2)\}$, $\eta_u = \eta_y\mu_y^2/(256\sigma_{\max}^2(A))$, $\eta_v = \eta_z\mu_z^2/(256\sigma_{\max}^2(A))$, *we have*

$$V_t - V_{t+1} \geq \frac{1}{4\eta_x}\|x_{t+1} - x_t\|^2 + \frac{1}{16\eta_y}\|y_{t+1}' - y_t'\|^2 + \frac{1}{8\eta_z}\|z_{t+1}' - z_t'\|^2$$
$$+ \frac{\eta_u}{4}\|Bx + A'y_\delta'^*(x_t, u_{t+1}) - b\|^2 + \frac{\eta_v}{2}\|Bx + A'z'^*(x_t, v_{t+1}) - b\|^2$$
$$+ \frac{\eta_u}{4}\|Bx + A'y_t' - b\|^2 + \frac{\eta_v}{4}\|Bx + A'z_t' - b\|^2 + \frac{\eta_x}{4}\|\nabla\phi_\delta(x_t)\|^2 \qquad (42)$$

*Thus,*

$$\frac{1}{T}\sum_{t=0}^{T-1}\|\nabla\phi_\delta(x_t)\|^2 \le \frac{4}{T\eta_x}(V_0 - \min_t V_t) \tag{43}$$

*Proof.* First, for function $d$, we have

$$d(x_t, z_t', u_{t+1}, v_{t+1}) - d(x_t, z_t', u_t, v_t)$$
$$=K(x_t, y_\delta'^*(x_t, u_{t+1}), z_t', u_{t+1}, v_{t+1}) - K(x_t, y_\delta'^*(x_t, u_t), z_t', u_t, v_t)$$
$$\ge K(x_t, y_\delta'^*(x_t, u_{t+1}), z_t', u_{t+1}, v_{t+1}) - K(x_t, y_\delta'^*(x_t, u_{t+1}), z_t', u_t, v_t)$$
$$\ge -\eta_v\|Bx_t + A'z_t' - b\|^2 + \eta_u(Bx_t + A'y_\delta'^*(x_t, u_{t+1}) - b)^\top(Bx_t + A'y_t' - b)$$

Note that

$$\nabla_x d(x, z', u, v) = \nabla_x\phi_\delta(x, y_\delta^*(x, u), z) + B^\top(u - v) + \rho_1 B^\top(Bx + A'y_\delta'^*(x, u) - b)$$
$$- \rho_2 B^\top(Bx + A'z' - b)$$
$$\nabla_{z'} d(x, z', u, v) = \begin{bmatrix} \nabla_z g(x, z) - A^\top v - \rho_2 A^\top(Bx + A'z' - b) \\ v + (Bx + A'z' - b) \end{bmatrix}$$

Thus, according to Lemma D.2, we know that $\nabla_x d(x, z', u, v)$ is $(L_K + L_K\sigma_{yx})$-continuous w.r.t. $x, z'$ and $\nabla_z' d(x, z', u, v)$ is $L_K$-continuous w.r.t. $x, z'$. Define $L_d = \max\{L_K + L_K\sigma_{yx}, L_K\}$. We have

$$d(x_{t+1}, z_{t+1}', u_{t+1}, v_{t+1}) - d(x_t, z_t', u_{t+1}, v_{t+1})$$
$$\ge \langle\nabla_x K(x_t, y_\delta'^*(x_t, u_{t+1}), z_t', u_{t+1}, v_{t+1}), x_{t+1} - x_t\rangle + \langle\nabla_{z'} K(x_t, y_\delta'^*(x_t, u_{t+1}), z_t', u_{t+1}, v_{t+1}), z_{t+1}' - z_t'\rangle$$
$$- \frac{L_d}{2}(\|x_{t+1} - x_t\|^2 + \|z_{t+1}' - z_t'\|^2)$$
$$\ge \langle\nabla_x K(x_t, y_\delta'^*(x_t, u_{t+1}), z_t', u_{t+1}, v_{t+1}), x_{t+1} - x_t\rangle + \frac{1}{\eta_z}\|z_{t+1}' - z_t'\|^2$$
$$- \frac{L_d}{2}(\|x_{t+1} - x_t\|^2 + \|z_{t+1}' - z_t'\|^2)$$

Then, for function $q$, we have

$$q(x_t, v_t) - q(x_t, v_{t+1})$$
$$\ge K(x_t, y_\delta'^*(x_t), z'^*(x_t, v_t), u_t, v_t) - K(x_t, y_\delta'^*(x_t), z'^*(x_t, v_{t+1}), u_t, v_{t+1})$$
$$\ge K(x_t, y_\delta'^*(x_t), z'^*(x_t, v_{t+1}), u_t, v_t) - K(x_t, y_\delta'^*(x_t), z'^*(x_t, v_{t+1}), u_t, v_{t+1})$$
$$\ge \eta_v(Bx_t + A'z'^*(x_t, v_{t+1}) - b)^\top(Bx_t + A'z_t' - b)$$

Note that

$$\nabla_x q(x, v) = \nabla_x\phi_\delta(x, y_\delta^*(x), z^*(x, v)) + B^\top(u_\delta^*(x) - v) + \rho_1 B^\top(Bx + A'y_\delta^*(x) - b)$$
$$- \rho_2 B^\top(Bx + A'z'^*(x, v) - b)$$

Thus, according to Lemma D.2 and E.3, $q(\cdot, v)$ is $L_q = (L_K + L_K\sigma_{zx} + L_K\sigma_{yb} + \sigma_{max}(B)\sigma_{ub})$-smooth. We have

$$q(x_t, v_{t+1}) - q(x_{t+1}, v_{t+1})$$
$$\ge \langle\nabla_x K(x_t, y_\delta'^*(x_t), z'^*(x_t, v_{t+1}), u^*(x_t), v_{t+1}), x_t - x_{t+1}\rangle - \frac{L_q}{2}(\|x_{t+1} - x_t\|^2)$$

Finally, for function $K$, we have

$$K(x_t, y_t', z_t', u_t, v_t) - K(x_t, y_t', z_t', u_{t+1}, v_{t+1}) = -\eta_u\|Bx_t + A'y_t' - b\|^2 + \eta_v\|Bx_t + A'z_t' - b\|^2,$$

and

$$K(x_t, y_t', z_t', u_{t+1}, v_{t+1}) - K(x_{t+1}, y_{t+1}', z_{t+1}', u_{t+1}, v_{t+1})$$
$$\ge \frac{1}{\eta_x}\|x_{t+1} - x_t\|^2 + \frac{1}{\eta_y}\|y_{t+1}' - y_t'\|^2 - \frac{1}{\eta_z}\|z_{t+1}' - z_t'\|^2$$
$$- \frac{L_K}{2}(\|x_{t+1} - x_t\|^2 + \|y_{t+1}' - y_t'\|^2 + \|z_{t+1}' - z_t'\|^2).$$

Thus, for $V_t$, we have

$$
\begin{aligned}
&V_t - V_{t+1} \\
\geq & \langle \nabla_x K(x_t, y_\delta'^*(x_t, u_{t+1}), z_t', u_{t+1}, v_{t+1}), x_{t+1} - x_t \rangle + \frac{1}{\eta_z}\|z_{t+1}' - z_t'\|^2 \\
& + \eta_u(Bx_t + A'y_\delta'^*(x_t, u_{t+1}) - b)^\top(Bx_t + A'y_t' - b) - \eta_v\|Bx_t + A'z_t' - b\|^2 \\
& - \frac{L_d}{2}(\|x_{t+1} - x_t\|^2 + \|z_{t+1}' - z_t'\|^2) + 2\langle \nabla_x K(x_t, y_\delta'^*(x_t), z'^*(x_t, v_{t+1}), u^*(x_t), v_{t+1}), x_t - x_{t+1} \rangle \\
& + 2\eta_v(Bx_t + A'z'^*(x_t, v_{t+1}) - b)^\top(Bx_t + A'z_t' - b) - L_q(\|x_{t+1} - x_t\|^2) \\
& + \frac{1}{4\eta_x}\|x_{t+1} - x_t\|^2 + \frac{1}{4\eta_y}\|y_{t+1}' - y_t'\|^2 - \frac{1}{4\eta_z}\|z_{t+1}' - z_t'\|^2 - \frac{\eta_u}{4}\|Bx_t + A'y_t' - b\|^2 \\
& + \frac{\eta_v}{4}\|Bx_t + A'z_t' - b\|^2 - \frac{L_K}{8}(\|x_{t+1} - x_t\|^2 + \|y_{t+1}' - y_t'\|^2 + \|z_{t+1}' - z_t'\|^2) \\
\geq & -\|\nabla_x K(x_t, y_\delta'^*(x_t, u_{t+1}), z_t', u_{t+1}, v_{t+1}) - \nabla_x K(x_t, y_\delta'^*(x_t), z'^*(x_t, v_{t+1}), u^*(x_t), v_{t+1})\|\|x_{t+1} - x_t\| \\
& + \left(\frac{\eta_u}{2} - \frac{\eta_u}{4}\right)\|Bx_t + A'y_t' - b\|^2 + \frac{\eta_u}{2}\|Bx_t + A'y_\delta'^*(x_t, u_{t+1}) - b\|^2 - \frac{\eta_u}{2}\|A'y_t' - A'y_\delta'^*(x_t, u_{t+1})\|^2 \\
& + \left(\eta_v + \frac{\eta_v}{4} - \eta_v\right)\|Bx_t + A'z_t' - b\|^2 + \eta_v\|Bx_t + A'z'^*(x_t, v_{t+1}) - b\|^2 - \eta_v\|A'z_t' - A'z'^*(x_t, v_{t+1})\|^2 \\
& - \frac{\eta_x}{2}\|\nabla_x K(x_t, y_\delta'^*(x_t), z'^*(x_t, v_{t+1}), u^*(x_t), v_{t+1}) - \nabla_x K(x_t, y_t', z_t', u_{t+1}, v_{t+1})\|^2 \\
& + \frac{1}{2\eta_x}\|x_t - x_{t+1}\|^2 + \frac{\eta_x}{2}\|\nabla_x K(x_t, y_\delta'^*(x_t), z'^*(x_t, v_{t+1}), u^*(x_t), v_{t+1})\|^2 \\
& + \left(\frac{1}{4\eta_x} - \frac{L_K}{8} - \frac{L_d}{2} - L_q\right)\|x_{t+1} - x_t\|^2 \\
& + \left(\frac{1}{\eta_z} - \frac{1}{4\eta_z} - \frac{L_K}{8} - \frac{L_d}{2}\right)\|z_{t+1}' - z_t'\|^2 \\
& + \left(\frac{1}{4\eta_y} - \frac{L_K}{8}\right)\|y_{t+1}' - y_t'\|^2,
\end{aligned}
$$

and

$$
\begin{aligned}
&V_t - V_{t+1} \\
\geq & -\frac{1}{4\eta_x}\|x_{t+1} - x_t\|^2 - 3\eta_x L_K^2\|y_\delta'^*(x_t, u_{t+1}) - y_\delta'^*(x_t)\|^2 - 3\eta_x L_K^2\|z'^*(x_t, v_{t+1}) - z_t'\|^2 \\
& - 3\eta_x L_K^2\|u_{t+1} - u^*(x_t)\|^2 \\
& + \frac{\eta_u}{4}\|Bx_t + A'y_t' - b\|^2 + \frac{\eta_u}{2}\|Bx_t + A'y_\delta'^*(x_t, u_{t+1}) - b\|^2 - \frac{\eta_u \sigma_{\max}^2(A)}{2}\|y_\delta'^*(x_t, u_{t+1}) - y_t'\|^2 \\
& + \frac{\eta_v}{4}\|Bx_t + A'z_t' - b\|^2 + \eta_v\|Bx_t + A'z'^*(x_t, v_{t+1}) - b\|^2 - \eta_v \sigma_{\max}^2(A)\|z'^*(x_t, v_{t+1}) - z_t'\|^2 \\
& - 2\eta_x L_K^2\|y_\delta'^*(x_t) - y_\delta'^*(x_t, u_{t+1})\|^2 - 2\eta_x L_K^2\|y_\delta'^*(x_t, u_{t+1}) - y_t'\|^2 - 2\eta_x L_K^2\|u_{t+1} - u^*(x_t)\|^2 \\
& - 2\eta_x L_K^2\|z'^*(x_t, v_{t+1}) - z_t'\|^2 + \frac{1}{2\eta_x}\|x_t - x_{t+1}\|^2 \\
& + \frac{\eta_x}{4}\|\nabla_x K(x_t, y_\delta'^*(x_t), z'^*(x_t), u^*(x_t), v^*(x_t))\|^2 - \eta_x L_K^2\|z'^*(x_t) - z'^*(x_t, v_{t+1})\|^2 \\
& - \eta_x L_K^2\|v_{t+1} - v^*(x_t)\|^2 \\
& + \left(\frac{1}{4\eta_x} - \frac{L_K}{8} - \frac{L_d}{2} - L_q\right)\|x_{t+1} - x_t\|^2 \\
& + \left(\frac{3}{4\eta_z} - \frac{L_K}{8} - \frac{L_d}{2}\right)\|z_{t+1}' - z_t'\|^2 \\
& + \left(\frac{1}{4\eta_y} - \frac{L_K}{8}\right)\|y_{t+1}' - y_t'\|^2,
\end{aligned}
$$

and

$$V_t - V_{t+1}$$

$$\geq \left( \frac{1}{2\eta_x} + \frac{1}{4\eta_x} - \frac{1}{4\eta_x} - \frac{L_K}{8} - \frac{L_d}{2} - L_q \right) \|x_{t+1} - x_t\|^2$$

$$+ \left( \frac{1}{4\eta_y} - \frac{L_K}{8} \right) \|y'_{t+1} - y'_t\|^2 - \left( 5\eta_x L_K^2 + \frac{\eta_u \sigma_{\max}^2(A)}{2} \right) \|y_\delta'^*(x_t, u_{t+1}) - y'_t\|^2$$

$$+ \left( \frac{3}{4\eta_z} - \frac{L_K}{8} - \frac{L_d}{2} \right) \|z'_{t+1} - z'_t\|^2 - \left( 5\eta_x L_K^2 + \eta_v \sigma_{\max}^2(A) \right) \|z'^*(x_t, v_{t+1}) - z'_t\|^2$$

$$+ \frac{\eta_u}{4} \|Bx_t + A'y'_t - b\|^2 + \frac{\eta_u}{2} \|Bx_t + A'y_\delta'^*(x_t, u_{t+1}) - b\|^2 - 5\eta_x L_K^2 \|y_\delta'^*(x_t, u_{t+1}) - y_\delta'^*(x_t)\|^2$$

$$+ \frac{\eta_v}{4} \|Bx_t + A'z'_t - b\|^2 + \eta_v \|Bx_t + A'z'^*(x_t, v_{t+1}) - b\|^2 - \eta_x L_K^2 \|z'^*(x_t, v_{t+1}) - z'^*(x_t)\|^2$$

$$+ \frac{\eta_x}{4} \|\nabla_x K(x_t, y_\delta'^*(x_t), z'^*(x_t), u^*(x_t), v^*(x_t))\|^2 - 5\eta_x L_K^2 \|u_{t+1} - u^*(x_t)\|^2 - \eta_x L_K^2 \|v_{t+1} - v^*(x_t)\|^2$$

$$\geq \left( \frac{1}{2\eta_x} - \frac{L_K}{8} - \frac{L_d}{2} - L_q \right) \|x_{t+1} - x_t\|^2$$

$$+ \left( \frac{1}{4\eta_y} - \frac{L_K}{8} - \frac{20\eta_x L_K^2 + 2\eta_u \sigma_{\max}^2(A)}{\mu_y^2 \eta_y^2} - 15\eta_x L_K^2 \sigma_{uy}^2 \right) \|y'_{t+1} - y'_t\|^2$$

$$+ \left( \frac{3}{4\eta_z} - \frac{L_K}{8} - \frac{L_d}{2} - \frac{20\eta_x L_K^2 + 4\eta_v \sigma_{\max}^2(A)}{\mu_z^2 \eta_z^2} - 3\eta_x L_K^2 \sigma_{vz}^2 \right) \|z'_{t+1} - z'_t\|^2$$

$$+ \left( \frac{\eta_u}{2} - 5\sigma_y^2 \eta_x L_K^2 - 15\eta_x L_K^2 \sigma_{u2}^2 \right) \|Bx_t + A'y_\delta'^*(x_t, u_{t+1}) - b\|^2$$

$$+ \left( \eta_v - \sigma_z^2 \eta_x L_K^2 - 3\eta_x L_K^2 \sigma_{v2}^2 \right) \|Bx_t + A'z'^*(x_t, v_{t+1}) - b\|^2$$

$$+ \left( \frac{\eta_u}{4} - 15\eta_x L_K^2 \sigma_{u1}^2 \right) \|Bx_t + A'y'_t - b\|^2 + \left( \frac{\eta_v}{4} - 3\eta_x L_K^2 \sigma_{v1}^2 \right) \|Bx_t + A'z'_t - b\|^2 + \frac{\eta_x}{4} \|\nabla \phi_\delta(x)\|^2$$

where the last equality is due to Lemma D.2, E.2 and $\nabla_x K(x_t, y_\delta'^*(x_t), z'^*(x_t), u^*(x_t), v^*(x_t)) = \nabla \phi_\delta(x)$.

$\square$

**Proof of Theorem 5.2**

When $\delta \leq \mu_g/(2l_{f,1})$, $0 \leq \rho_1 \leq \frac{\mu_g - \delta l_{f,1}}{\sigma_{\max}^2(A)}$, $0 \leq \rho_2 \leq \frac{\mu_g}{\sigma_{\max}^2(A)}$, $\eta_y = 1/(4L_K)$, $\eta_z = 2/(L_K + 4L_d)$, $\eta_x = \min\{\eta_y \mu_y^2/(640L_K^2), \eta_z \mu_z^2/(640L_K^2), \eta_u/(240(\sigma_y^2 + \sigma_{u2}^2 + \sigma_{u1}^2)L_K^2), \eta_v/(240(\sigma_z^2 + \sigma_{v2}^2 + \sigma_{v1}^2)L_K^2), 2/(L_K + 4L_d + 8L_q), 1/(1920\eta_y L_K^2 \sigma_{uy}^2), 1/(1920\eta_z L_K^2 \sigma_{uz}^2)\}$, $\eta_u = \eta_y \mu_y^2/(256\sigma_{\max}^2(A))$, $\eta_v = \eta_z \mu_z^2/(256\sigma_{\max}^2(A))$, according to (43), we have

$$\frac{1}{T} \sum_{t=0}^{T-1} \|\nabla \phi_\delta(x_t)\|^2 \leq \frac{4}{T\eta_x} (V_0 - \min_t V_t) \tag{44}$$

Note that

$$V_t = \frac{1}{4} K(x_t, y'_t, z'_t, u_t, v_t) + 2q(x_t, v_t) - d(x_t, z'_t, u_t, v_t)$$

$$\geq 2q(x_t, v_t) - \frac{3}{4} d(x_t, z'_t, u_t, v_t) \geq \frac{5}{4} q(x_t, v_t) \geq \frac{5}{4} \phi_\delta(x_t) \geq \frac{5\delta \Phi^*}{4} - \frac{5\delta^2 l_{f,0}^2}{8\mu_g}$$

Therefore, when $\delta = \Theta(\epsilon)$, with $T = O(\epsilon^{-4})$, we have $t \in [T]$, such that $\|\nabla \Phi_\delta(x_t)\| = \|\frac{1}{\delta} \nabla \phi_\delta(x_t)\| \leq \epsilon$.

Moreover, if we have $x_0, y_0, z_0, u_0, v_0$ such that

$$\|y_0 - y_\delta^*(x_0)\| \le O(\delta)$$
$$\|u_0 - u_\delta^*(x_0)\| \le O(\delta)$$
$$\|z_0 - z^*(x_0)\| \le O(\delta)$$
$$\|v_0 - v^*(x_0)\| \le O(\delta)$$

Then, set $\alpha_0 = h(x_0, y_0)$, $\beta_0 = h(x_0, z_0)$, we can easily prove that

$$\|Bx_0 + A'y_\delta'^*(x_0, u_0) - b\| \le \|Bx_0 + A'y_\delta'^*(x_0, u_\delta^*(x_0)) - b\| + \sigma_{\max}(A')\sigma_{yu}\|u_0 - u_\delta^*(x_0)\| \le O(\delta)$$
$$\|Bx_0 + A'y_0' - b\| \le \sigma_{\max}(A')\|y_0' - y'^*(x_0)\| \le O(\delta)$$

and similarly,

$$\|Bx_0 + A'z'^*(x_0, v_0) - b\| \le O(\delta)$$
$$\|Bx_0 + A'z_0' - b\| \le O(\delta).$$

Then, we have

$$
\begin{aligned}
V_0 =& \frac{1}{4}[\delta f(x_0, y_0) + (g(x_0, y_0) - g(x_0, z_0)) + u_0^\top(Bx_0 + A'y_0' - b) - v_0^\top(Bx_0 + A'z_0' - b) \\
&+ \frac{\rho_1}{2}\|Bx_0 + A'y_0' - b\|^2 - \frac{\rho_2}{2}\|Bx_0 + A'z_0' - b\|^2] \\
&+ 2[\delta f(x_0, y_\delta^*(x_0)) + (g(x_0, y_\delta^*(x_0)) - g(x_0, z'^*(x_0, v_0))) - v_0^\top(Bx_0 + A'z'^*(x_0, v_0) - b) \\
&- \frac{\rho_2}{2}\|Bx_0 + A'z'^*(x_0, v_0) - b\|^2] - [\delta f(x_0, y_\delta'^*(x_0, u_0)) + (g(x_0, y_\delta'^*(x_0, u_0)) - g(x_0, z_0)) \\
&+ u_0^\top(Bx_0 + A'y_\delta'^*(x_0, u_0) - b) - v_0^\top(Bx_0 + A'z_0' - b) \\
&+ \frac{\rho_1}{2}\|Bx_0 + A'y_\delta'^*(x_0, u_0) - b\|^2 - \frac{\rho_2}{2}\|Bx_0 + A'z_0' - b\|^2] \\
\le& \frac{5\delta\Phi(x_0)}{4} + O(1)l_{f,1}(\|y_0 - y^*(x_0)\| + \|y_\delta^*(x_0) - y^*(x_0)\| + \|y_\delta^*(x_0, u_0) - y^*(x_0)\|) \\
&+ O(1)C_g(\|z_0 - y_0\| + \|y_\delta^*(x_0) - z^*(x_0, v_0)\| + \|y_\delta^*(x_0, u_0) - z_0\|) \\
&+ O(\|Bx_0 + A'z_0' - b\| + \|Bx_0 + A'z_0' - b\|^2 + \|Bx_0 + A'y_0' - b\| + \|Bx_0 + A'y_0' - b\|^2 \\
&+ \|Bx_0 + A'z'^*(x_0, v_0) - b\| + \|Bx_0 + A'z'^*(x_0, v_0) - b\|^2 \\
&+ \|Bx_0 + A'y_\delta'^*(x_0, u_0) - b\| + \|Bx_0 + A'y_\delta'^*(x_0, u_0) - b\|^2) \\
\le& \frac{5\delta\Phi(x_0)}{4} + O(\delta)
\end{aligned}
$$

where $G = \max\{g(x_0, y_0), g(x_0, z_0), g(x_0, y_\delta^*(x_0)), g(x_0, z^*(x_0, v_0)), g(x_0, y_\delta^*(x_0, u_0))\}, \mathcal{C} = \{y \in \mathbb{R}^{d_y} | g(x_0, y) \le G\}, C_g = \sup_{y \in \mathcal{C}} \nabla_y g(x_0, y)$. Since $g(x_0, y)$ is strongly convex w.r.t $y$, its sub-level set $\mathcal{C}$ is compact and convex. Moreover, since $g$ is Lipschitz smooth, its gradient in this compact set $\mathcal{C}$ is upper bounded by an $O(1)$ constant $C_g$.

We can notice that

$$V_0 - \min_t V_t \le \frac{5\delta[\Phi(x_0) - \Phi^*]}{4} + O(\delta) = O(\epsilon)$$

and

$$\frac{1}{T}\sum_{t=0}^{T-1}\|\nabla\Phi_\delta(x_t)\|^2 = \frac{1}{T\delta^2}\sum_{t=0}^{T-1}\|\nabla\phi_\delta(x_t)\|^2 \le \frac{4}{T\eta_x\delta^2}(V_0 - \min_t V_t) = O\left(\frac{\epsilon^{-1}}{T}\right).$$

Therefore, we can find an $\epsilon$-stationary point of $\Phi_\delta(x)$ with a complexity of $O(\epsilon^{-3})$.

### Proof of Corollary 5.3

When $h(x, y) = Bx + Ay - b$, $A$ is full row rank and under Assumption 3.1, 3.2, 3.3 and $\delta = O(\epsilon) \le \mu_g/(2l_{f,1})$ if we apply PGD for $\max_{v \in \mathbb{R}_+} \min_{z \in \mathbb{R}^{d_y}} g(x_0, y) + v^\top(Bx_0 + Az - b)$ with a fixed $x_0$, then according to Theorem 6 in [16], we can find $\hat{v}, \hat{z}$ such that

$$\|v_0 - v^*(x_0)\| \le \delta$$
$$\|z_0 - z^*(x_0)\| \le \delta$$

with a complexity of $O(\log(\epsilon^{-1}))$.

Set $y_0 = z_0 = \hat{z}$, $u_0 = v_0 = \hat{v}$, $\alpha_0 = h(x_0, y_0)$, $\beta_0 = h(x_0, z_0)$. We have

$$\|y_0 - y_\delta^*(x_0)\| \leq O(\delta)$$
$$\|Bx_0 + A'y_\delta'^*(x_0, u_0) - b\| \leq O(\delta)$$
$$\|Bx_0 + A'y_0' - b\| \leq O(\delta)$$
$$\|u_0 - u_\delta^*(x_0)\| \leq O(\delta)$$
$$\|Bx_0 + A'z'^*(x_0, v_0) - b\| \leq O(\delta)$$
$$\|Bx_0 + A'z_0' - b\| \leq O(\delta)$$

with $x_0, y_0', z_0', u_0, v_0$ as initial points and apply Algorithm 1, according to Theorem 5.2, we can find an $\epsilon$-stationary point of $\Phi_\delta$ with a complexity of $O(\epsilon^{-3})$.

Thus, the total complexity is $O(\epsilon^{-3} + \log(\epsilon^{-1})) = O(\epsilon^{-3})$.

## F  Detailed Experimental Settings and Additional Experiments

We adapt and modify the code from [16]. The experiments on the toy example and hyperparameter optimization for SVM are conducted on an AMD EPYC 9554 64-Core Processor. The experiments on transportation network design are conducted on an Intel(R) Xeon(R) Platinum 8375C CPU. For the toy example, we set the hyperparameters for our algorithm as $\delta = 0.1, \eta_x = \eta_y = \eta_z = \eta_u = \eta_v = 0.01, \rho_1 = \rho_2 = 1$.

### F.1  Hyperparameter optimization in SVM

Support Vector Machines (SVMs) construct a machine learning model by identifying the best possible hyperplane that maximizes the separation margin between data points from different classes. In a hard-margin SVM, no misclassification is allowed, ensuring that all samples are correctly classified. Conversely, soft-margin SVMs allow certain samples to be misclassified to accommodate cases where a perfect separation is not feasible. To achieve this, slack variables $\xi$ are introduced to quantify classification violations for each sample.

We consider the same problem formulation as in [16], which is

$$\min_c \quad \mathcal{L}_{\mathcal{D}_{\text{val}}}(w^*, b^*) = \sum_{(z_{\text{val}}, l_{\text{val}}) \in \mathcal{D}_{\text{val}}} \exp\left(1 - l_{\text{val}}\left(z_{\text{val}}^\top w^* + b^*\right)\right) + \frac{1}{2}\|c\|^2$$

$$w^*, b^*, \xi^* = \arg\min_{w, b, \xi} \quad \frac{1}{2}\|w\|^2$$
$$\text{s.t.} \quad l_{\text{tr},i}(z_{\text{tr},i}^\top w + b) \geq 1 - \xi_i, \quad \forall i \in \{1, \ldots, |\mathcal{D}_{\text{tr}}|\}$$
$$\xi_i \leq c_i, \quad \forall i \in \{1, \ldots, |\mathcal{D}_{\text{tr}}|\}.$$

where $\mathcal{D}_{\text{tr}} \triangleq \{(z_{\text{tr},i}, l_{\text{tr},i})\}_{i=1}^{|\mathcal{D}_{\text{tr}}|}$ is the training dataset, $\mathcal{D}_{\text{val}} \triangleq \{(z_{\text{val},i}, l_{\text{val},i})\}_{i=1}^{|\mathcal{D}_{\text{val}}|}$ is the validation dataset, with $z_{\text{tr},i}$ ($z_{\text{val},i}$) being the features of sample $i$ and $l_{\text{tr},i}$ ($l_{\text{val},i}$) being its corresponding labels. The hyperparameter $c_i$ is introduced to bound the soft margin violation $\xi_i$. Note that the LL problem is equivalent to

$$w^*, b^* = \arg\min_{w, b, \xi} \quad \frac{1}{2}\|w\|^2$$
$$\text{s.t.} \quad l_{\text{tr},i}(z_{\text{tr},i}^\top w + b) \geq 1 - c_i, \quad \forall i \in \{1, \ldots, |\mathcal{D}_{\text{tr}}|\}$$

Then upper level variables are $c$, and the lower level variables are $w, b$. Thus, we have a coupled linear constraint for the lower level problem.

We compared our algorithm SFLCB with GAM [46], LV-HBA [50], BLOCC [16], and BiC-GAFFA [49] on the diabetes dataset. For GAM, we follow the same implementation approach and hyper-parameters as [16], setting $\alpha = 0.05, \epsilon = 0.005$ and using a different formulation as introduced

in [46]. For BLOCC and SFLCB, we set the LL objective as $\frac{1}{2}\|w\|^2 + \frac{\mu}{2}b^2$, where $\mu = 0.01$ serves as a regularization term to make the LL problem strongly convex w.r.t $w, b$. For LV-HBA, we set $\alpha = 0.01, \gamma_1 = 0.1, \gamma_2 = 0.1, \eta = 0.001$. For BLOCC, we set $\gamma = 12, \eta = 0.01, T = 20, T_y = 20, \eta_{1g} = 0.001, \eta_{1F} = 0.00001, \eta_{2g} = 0.0001, \eta_{2F} = 0.0001$. For BiC-GAFFA, we set $p = 0.3, \gamma_1 = 10, \gamma_2 = 0.01, \eta = 0.01, \alpha = 0.001, \beta = 0.001, c_0 = 10, R = 10$. For our algorithm SFLCB, we set $\delta = 0.01, \eta_x = \eta_y = \eta_z = \eta_u = \eta_v = 0.001, \rho_1 = \rho_2 = 0.01$. The hyperparameters of LV-HBA, BLOCC are the same as those used in [16] and the hyperparameters of BiC-GAFFA are the same as those used in their original paper [49]. The experiments are conducted across 10 different random train-validation-test splits, and the average results along with one standard deviation are reported in Figure 2.

## F.2 Transportation network design

We consider the same setting as in [16]. In this setting, we act as the operator to design a new network that connects a set of stations $S$. Passengers then decide whether to use this network based on their rational decisions (lower level). The objective is to maximize the operator's benefit (upper level). The operator can select a set of potential links $\mathcal{A} \subseteq S \times S$, and for each link $(i, j) \in \mathcal{A}$, determine its capacity $x_{ij}$. A link is constructed if $x_{ij} > 0$; a larger $x_{ij}$ attracts more travelers, generating more revenue but incurring higher construction costs $c_{ij}$. Passenger demand is defined over a set of origin-destination pairs $\mathcal{K} \subseteq S \times S$. For each $(o, d) \in \mathcal{K}$, there is a known traffic demand $w_{od}$ and existing travel times $t_{od}^{\text{ext}}$. We assume a single existing network. The fraction of passengers choosing the new network for each $(o, d)$ pair is denoted by $y_{od}$, and $y_{od}^{ij}$ represents the proportion using link $(i, j)$.

We summarize the notation as follows. We keep the notation the same as in [16].

- $x_{ij} \in \mathbb{R}_+$, the capacity of the new network for the link $(i, j) \in \mathcal{A}$.

- $y^{od} \in [0, 1]$, the proportion of passengers from $(o, d) \in \mathcal{K}$ choosing the new network for their travel.

- $y_{ij}^{od} \in [0, 1]$, the proportion of passengers from $(o, d) \in \mathcal{K}$ choosing the new network and use the link $(i, j) \in \mathcal{A}$

- $x = \{x_{ij}\}_{\forall (i,j) \in \mathcal{A}}$ are the upper-level variables to be optimized.

- $\mathcal{X} = \mathbb{R}_+^{|\mathcal{A}|}$ represents the domain of $x$.

- $y = \{y^{od}, \{y_{ij}^{od}\}_{\forall (i,j) \in \mathcal{A}}\}_{\forall (o,d) \in \mathcal{K}}$ are the lower-level variables to be optimized

- $\mathcal{Y} = [\varepsilon, 1 - \varepsilon]^{|\mathcal{K}|} \times [\varepsilon, 1 - \varepsilon]^{|\mathcal{A}||\mathcal{K}|}$, where $\varepsilon$ is a small positive number, represents the domain of $y$.

- $w^{od}$, the total estimated demand for $(o, d) \in \mathcal{K}$.

- $m^{od}$, the revenue obtained by the operator from a passenger in $(o, d) \in \mathcal{K}$.

- $c_{ij}$, the construction cost per passenger for link $(i, j) \in \mathcal{A}$.

- $t_{ij}$, the travel time for link $(i, j) \in \mathcal{A}$.

- $t_{\text{ext}}^{od}$, travel time on the existing network for passengers in $(o, d) \in \mathcal{K}$.

- $\omega_t < 0$, the coefficient associated with the travel time for passengers.

With these notions, we can introduce the bilevel formulations of this problem. At the upper level, the network operator seeks to maximize the overall profit by attracting more passengers while minimizing construction costs; thus, its objective is

$$\min_{x \in \mathcal{X}} f(x, y_g^*(x)) := -\left( \underbrace{\sum_{\forall (o,d) \in \mathcal{K}} m^{od} y^{\text{od}*}(x)}_{profit} - \underbrace{\sum_{\forall (i,j) \in \mathcal{A}} c_{ij} x_{ij}}_{cost} \right), \tag{45}$$

where $y^{\text{od}*}(x)$ are optimal lower-level passenger flows. For the lower-level, the objective is defined as finding the flow variables that maximize passenger utility and minimize flow entropy cost.

$$\min_{y \in \mathcal{Y}} g(x, y) := - \Bigg( \underbrace{\sum_{(o,d) \in \mathcal{K}} \sum_{(i,j) \in \mathcal{A}} w^{od} \omega_t t_{ij} y_{ij}^{od} + \sum_{(o,d) \in \mathcal{K}} w^{od} \omega_t t_{\text{ext}}^{od}(1 - y^{od})}_{\text{passengers utility}} \qquad (46)$$

$$+ \underbrace{\sum_{(o,d) \in \mathcal{K}} w^{od} y^{od}(\ln(y^{od}) - 1) + \sum_{(o,d) \in \mathcal{K}} w^{od}(1 - y^{od})(\ln(1 - y^{od}) - 1)}_{\text{flow entropy cost}} \Bigg)$$

$$\text{s.t.} \quad \sum_{\forall j | (i,j) \in \mathcal{A}} y_{ij}^{od} - \sum_{\forall j | (j,i) \in \mathcal{A}} y_{ji}^{od} = \begin{cases} y^{od} & \text{if } i = o \\ -y^{od} & \text{if } i = d \\ 0 & \text{otherwise} \end{cases} \qquad \forall i, (o, d) \in \mathcal{S} \times \mathcal{K}$$

$$(47)$$

$$\sum_{\forall (o,d) \in \mathcal{K}} w^{od} y_{ij}^{od} \leq x_{ij} \qquad \forall (i, j) \in \mathcal{A} \qquad (48)$$

where (47) are the flow-conservation constraints and (48) are the capacity constraints.

We consider the same 3 nodes and 9 nodes settings as in [16] and use the same lower level feasibility criteria. The hyperparameter settings used in Figure 3 are listed below. For the 3 nodes setting, we set the hyperparameters of our method SFLCB as: $\delta = 0.1$, $\rho_1 = \rho_2 = 1000$, $\eta_x = \eta_y = \eta_z = \eta_u = \eta_v = 3e - 4$, and $T = 30000$. For the 9 nodes setting, we set the hyperparameters of our method SFLCB as: $\delta = 0.25$, $\rho_1 = \rho_2 = 50$, $\eta_x = \eta_y = \eta_z = \eta_u = \eta_v = 3e - 5$, and $T = 300000$. For BLOCC, we used the same hyperparameters as those used in [16].

