# OpenReview forum: "A Single-Loop First-Order Algorithm for Linearly Constrained Bilevel Optimization"
_NeurIPS.cc/2025/Conference — NeurIPS 2025 poster_

### Official Review · Reviewer_K5dx · 2025-06-09

**Clarity:** 1
**Significance:** 3
**Originality:** 3
**Rating:** 4
**Confidence:** 3

**Summary:**

This paper studies a bilevel optimization (BLO) problem with a linear inequality constraint in the lower-level (LL) problem under the assumption of strong convexity in the LL variable. The goal is to improve the known convergence rate from $ O(\epsilon^{-3} \ln(\epsilon^{-1}))$ to $O(\epsilon^{-3})$. The authors propose a novel penalty-based reformulation by transforming the original BLO problem (0) to a value-function-based formulation (1), then introducing a slackness parameter to convert inequality constraints to equality constraints (2), applying a Lagrangian formulation (3), and finally converting the problem to an augmented Lagrangian form (4), which is jointly strongly convex in the LL variable and slack variable.

The paper theoretically establishes function value and gradient equivalence between (0) and (1), then claims equivalence through steps (1)–(4). Based on this, the authors propose SFLCB, a fully single-loop algorithm with an O(\epsilon^{-3}) convergence guarantee. Experimental results on a toy problem, SVM parameter learning, and a transportation network demonstrate the method’s effectiveness.

**Questions:**

1.	Related Work Comparison:
In the table on page 3:
	•	[16] handles general domain constraints Y, not just LL inequality constraints. Please acknowledge this distinction.
	•	The comparison in line 112 is arguably unfair since [16] requires projection due to its more general setting.
	•	Please also include [20] in the table and discuss its relevance. This will strengthen the positioning of the paper.

2.	Equivalence Gaps:
The paper uses gradient equivalence between the original BLO and the value-function-based penalty formulation (original BLO problem (0) to penalty reformulation (1) $\phi_{\delta}(x)$), but the later equivalences (e.g., step 2 to 4) a at optimal points. Can the authors clarify this distinction and avoid using “equivalence” loosely? This impacts the logical flow of the methodology.

3.	Connections to Prior Work:
Several results appear to overlap with prior literature:
	•	Theorem 4.1 vs. Theorem 1.1 in [21] and Theorem 1/2 in [38]
	•	Lemma 4.4 vs. Lemma 2 in [16]
	•	Lemma 4.7 vs. results in [21]
	•	Theorem 4.9 vs. Theorem 1.1 in [21]
Please clarify the distinctions or cite appropriately instead of re-deriving them.

4.	Terminology and Notation:
	•	Replace “convex about y” with “convex in y” or “with respect to y”.
	•	Clarify the set notation: defining \mathcal{X} = \mathbb{R}^{d_x} after stating x \in \mathcal{X} can be misleading.
	•	Fix the confusion between “Lipschitz continuous” and “Lipschitz smooth” (e.g., Lines 134, 168).
	•	Please add numbering to key formulations (e.g., line 140) for clarity.

5.	Use of Saddle Envelopes:
In line 211, the paper introduces a Moreau-like or saddle-envelope formulation (see [53]). How is this justified as being equivalent to the original problem? More explanation is needed.

6.	Experimental Evaluation:
	•	For the transportation network problem, how long does the algorithm take to run?
	•	Can the authors provide convergence plots under a 24-hour time budget to demonstrate practical efficiency?
	•	In the SVM and transportation tasks, how well is the lower-level problem solved (e.g., optimality gap)?

**Ethical Concerns:**

["NO or VERY MINOR ethics concerns only"]

**Final Justification:**

I believe this paper presents a solid method with both theoretical and empirical strengths. The proposed approach is well-motivated, and the results are promising. However, I have two significant concerns that affect my overall assessment.

First, the writing quality is a major barrier. The language is often unclear and difficult to follow, which hinders readability and makes it challenging to fully evaluate the technical contributions. While the authors acknowledged this issue in the rebuttal and committed to improving the presentation, the current state of the writing makes it hard to assess whether the revisions will be sufficient to resolve the underlying clarity issues.

Second, the analysis techniques and experimental setup, while sound, appear fairly derivative. In addition, several known results are stated without appropriate citation in the main text, which weakens the positioning of the paper within the existing literature.

Given the strength of the method itself, I have decided to slightly raise my score. However, due to the concerns above — especially the difficulty in fully interpreting the contributions — I have lowered my confidence to reflect this uncertainty. I believe the paper could become a strong contribution with substantial revision.

**Limitations:**

Yes. The paper focuses on a specific BLO setting (LL strong convexity and linear inequality constraints) and clearly outlines the assumptions needed. The scope is narrower than in some related work, which should be acknowledged more clearly, but this limitation is not ignored. e.g. future potential improvement can be on a more generalized constraint setting or relaxing LL SC assumption to PL.

**Quality:**

3

**Strengths And Weaknesses:**

Strengths:
	•	Introduces a novel reformulation strategy using slack parameters and augmented Lagrangians for handling LL inequality constraints.
	•	Presents a theoretically grounded, fully single-loop algorithm (SFLCB) with a provable convergence rate of O(\epsilon^{-3}).
	•	Experimental evaluation on diverse tasks demonstrates practical performance.

Weaknesses:
	•	The language and writing clarity hinder readability and understanding of key concepts. Several expressions are nonstandard or unclear.
	•	The claimed equivalences between various reformulations are not always rigorously justified. Some gaps remain, especially between function-level and pointwise equivalence.
	•	The paper occasionally re-derives known results without citing or contrasting them with existing literature.
	•	Comparison to related work is somewhat unfair in cases where the settings differ (e.g., projection in [16] only because of its more complicated setting of considering domain constraint Y).

---

> ### Author Rebuttal · Authors · 2025-07-31
>
> We thank the reviewer for the valuable comments. Please see our responses below.
>
> >In the table on page 3: [16] handles general domain constraints Y, not just LL inequality constraints. Please acknowledge this distinction.
>
> Thanks for your advice. We will acknowledge this distinction in the revised paper. Actually, [16] assumed that $Y$ is a domain that is easy to project, and for any $y\in Y$ in a neighborhood of $y^*(x)$, its constraints satisfy the LICQ condition. Thus, [16] only used a simple projection to $Y$ to handle this domain constraint $Y$. If we adopt the same assumption on $Y$ as in [16], then in our algorithm we can simply add a projection onto $Y$ to handle this domain constraint, which would not cause any issues in theory or practice. The main theoretical focus of our work is on handling _coupled constraints_ of the form  $Bx+Ay-b\leq 0$, which are not easy to project onto. For such constraints, compared to [16], our work improves the double-loop algorithm to a single-loop one and provides a theoretical guarantee of better complexity. We believe these are significant results. Furthermore, in practice, single-loop algorithms are easier to implement. Our experiments also demonstrate that our method empirically outperforms [16].
>
> >The comparison in line 112 is arguably unfair since [16] requires projection due to its more general setting.
>
> We do acknowledge that [16] considers a more general setting, and we will highlight this in the revised paper to avoid misunderstandings, However, in line 112, what we stated was: "However, they [16] did not provide a discussion between the stationary point of the reformulated function $\Phi_\delta$ and the original hyper-objective $\Phi$." This is a factual description of their work [16]. While [16] considers a more general setting, it does not discuss the relationship between the stationary point of the reformulated problem and that of the original problem. Our work focuses on a more specific setting and provides a discussion on this relationship.
>
> >Please also include [20] in the table and discuss its relevance. This will strengthen the positioning of the paper.
>
> Thanks for your advice. The main reason we did not include [20] in Table is that [20] used a different stationary measure so that the complexity results of [20] are not directly comparable to our paper and [16, 21]. Our paper used the same reformulation as [16, 21], and as a result the complexity results of our paper and theirs are comparable. Thus, we mainly compared [16], [21] and our paper in Table and discussed [20] in the Related Works section. We will add [20] to Table and add some explanations in the revised paper.
>
> >The paper uses gradient equivalence between the original BLO and the value-function-based penalty formulation (original BLO problem (0) to penalty reformulation (1) ), but the later equivalences (e.g., step 2 to 4) a at optimal points. Can the authors clarify this distinction and avoid using “equivalence” loosely? This impacts the logical flow of the methodology.
>
> Thanks for your advice. We will clarify this distinction in the revised paper and avoid using “equivalence” loosely. Yes, the “equivalence” in Section 5 refers to the optimal points. Our derivation in Section 5 is intended to illustrate the idea behind the design of SFLCB. In fact, in Theorems 5.2 and 5.4, as well as Corollaries 5.3 and 5.5, we established the convergence results for stationary points of $\Phi_\delta$.
>
> >Several results appear to overlap with prior literature: Theorem 4.1 vs. Theorem 1.1 in [21] and Theorem 1/2 in [38]
>
> In Appendix line 824, we stated that "Similar results and proofs of Theorem 4.1 can also be found in [21, 16]." Thanks for your advice, we will add these citations in the main text.
>
> >Lemma 4.4 vs. Lemma 2 in [16]
>
> In Appendix line 840, we stated that our proof used the Lemma 2 and 3 in [16]. Thanks for your advice, we will add these citations in the main text.
>
> >Lemma 4.7 vs. results in [21]
>
> Our Lemma 4.7 is different from the results in [21]. [21] considers lower-level constraints  $y\in Y$ that are independent of $x$, whereas our work analyzes coupled constraints of the form $y\in Y(x)=$ { $y\in R^{d_y} | h(x,y)\leq 0$ }. Actually, in Appendix line 848, we stated that our Lemma 4.7 used Theorem 2 in [45]. We will add the citation in the main text.
>
> >Theorem 4.9 vs. Theorem 1.1 in [21]
>
> Our Theorem 4.9 is also very different from Theorem 1.1 in [21]. The analysis in [21] only considers lower-level constraints
> $y\in Y$ that are independent of $x$, whereas our work analyzes coupled constraints of the form $y\in Y(x)=$ { $y\in R^{d_y} | h(x,y)\leq 0$ }. Handling such coupled constraints requires estimating the Lagrangian multipliers in the implicit gradient formula, which makes our analysis more involved compared to [21]. Notably, the Future Directions section of [21] also emphasizes that analyzing lower-level $x$-dependent constraints is a valuable direction for future research, highlighting the value of our work.  In the main text line 191-194, we also stated the connections and differences between our results and those in [21].
>
> >Terminology and Notation
>
> Thanks for your advice. We will modify these in the revised paper.
>
> >In line 211, the paper introduces a Moreau-like or saddle-envelope formulation (see [53]). How is this justified as being equivalent to the original problem?
>
> Thanks for pointing out our imprecise use of terminology. In line 211, we mean $K$ and $L_\delta$ have the same optimal points and same optimal function value. Our derivation in Section 5 is intended to illustrate the idea behind the design of SFLCB. In fact, in Theorems 5.2 and 5.4, as well as Corollaries 5.3 and 5.5, we directly established the convergence results for stationary points of $\Phi_\delta$ (not for $K$). We hope this response can address your concern.
>
> >For the transportation network problem, how long does the algorithm take to run? Can the authors provide convergence plots under a 24-hour time budget to demonstrate practical efficiency?
>
> Thanks for your advice. We will add plots in the revised paper to better illustrate how the UL utility converges over time. Since the rebuttal does not support images, we present the following primary results in tables.
>
> 3 Node:
> |Time(s)|1|2|3|4|5|
> |---|---|---|---|---|---|
> |BLOCC|1.36|1.66|1.70|1.70|1.71|
> |SFLCB|1.04|1.69|1.84|1.84|1.84|
>
> 9 Node:
> |Time(s)|20|40|60|80|100|120|140|160|180|200|250|300|350|400|
> |---|----|----|----|----|----|----|----|----|----|----|----|----|----|----|
> |BLOCC|-420.87|-89.73|-23.66|-4.62|1.72|6.25|6.59|8.55|8.93|9.14|9.53|9.15|9.52|9.54|
> |SFLCB|2.57|12.09|15.30|18.03|19.95|20.52|20.92|21.18|21.19|21.20|21.06|20.79|20.67|20.40|
>
> In the 9-node setting, we set $\delta=0.25$ in SFLCB for better comparison with BLOCC (since in our reformulation, $\delta$ has a similar role as $1/\gamma$ in BLOCC and $\gamma=4$) and keep all other hyperparameters the same as those in our current paper. Note that the $\delta$ here is different from the one used in our paper. Thus, the results of the 9-node setting are different from the current paper.
>
> We can see that in both the 3-node and 9-node settings, SFLCB converges faster than BLOCC and achieves better UL utility.
>
> >In the SVM and transportation tasks, how well is the lower-level problem solved (e.g., optimality gap)?
>
> Thanks for this question. For SFLCB, suppose its final outputs are $x, y', u, z', v$. We keep its $x, y', u$ fixed, and only optimize $z'$ and its multiplier $v$ for additional 1000 interactions to get $z^{\ast}$ as its LL optimal point. Then, we calculate $||z-z^{\ast}||$ as its optimality gap. For BLOCC, suppose its final outputs are $x, y_g, \mu_g, y_F, \mu_F$. we also fix its final output $x$ and run the inner loop to optimize $y_g, \mu_g$ for additional 1000 interactions to get $y_g^{\ast}$ as its LL optimal point. We then calculate $||y_g-y_g^{\ast}||$ as its optimality gap. For SVM, we find that SFLCB's optimality gap is 3.6e-2, and BLOCC's is 1.9e-2. For the transportation 3-node task, the SFLCB's optimality gap is 6.6e-6, and BLOCC's is 2.3e-3. For the transportation 9-node task, the SFLCB's optimality gap is 9.7e-3, and BLOCC's is 5.5e-2. Overall, it can be observed that BLOCC and SFLCB have optimality gaps of similar magnitude and these optimality gaps are relatively small, indicating that the lower-level problem is well-solved.
>
> >The paper focuses on a specific BLO setting (LL strong convexity and linear inequality constraints) and clearly outlines the assumptions needed. The scope is narrower than in some related work, which should be acknowledged more clearly, but this limitation is not ignored. e.g. future potential improvement can be on a more generalized constraint setting or relaxing LL SC assumption to PL.
>
> Thank you for this constructive comment. We acknowledge that some related works [16, 48, 49] have considered more general settings, and future research can be done on more general settings. However, it is also worth noting that linear constraints already encompass a broad range of applications, including but not limited to distributed optimization [47, 18], adversarial training [52, 18], and hyperparameter optimization for some constrained learning tasks. Several related works have also concentrated on analyzing linear constraints with LL strong convexity, such as [41, 18, 20]. Actually, [18, 41] considered an even narrower setting ($Ay\leq b$) than ours. Compared to the double-loop structure in [16, 20] and the requirement of Hessian computations in [18, 41], our work provides a single-loop first-order algorithm for linearly constrained BLO with rigorous theoretical analyses. Our experiments also demonstrate the practical efficiency of our method. Therefore, we believe that our work has its own contributions and value. We will add some discussion and highlight that future potential improvement is worth exploring in the revised paper.

---

> ### Comment · Reviewer_K5dx · 2025-08-01
>
> **1. Setting and equivalence analysis of [16]**
>
> [16] consider both $y\in Y$ and $Ay+Bx+c\leq 0$ constraints. So it is not as the author response as "[16] considers $y\in Y$ easy to project but this paper consider $Ay+Bx+c\leq 0$ non-easy-to-project". Moreover, although [16] did not explicitly discuss on $\| \phi(x)-\phi_\delta(x)\|$, this result can be easily achieved directly following the proofs of [21]. And [16] established that the $O(\epsilon)$-local-solution of their reformulation is the $O(\epsilon)$-local-solution of the original problem, while this paper did not discuss in this or establishing distance bound for gradients between original problem and their final reformulation (4).
>
> **2. Comparison with [20]**
>
> Although it considers a different convergence. Comparison is still very important. I am not convinced with theoretical analysis.
>
> **3. Equivalence problem**
>
> As the author only argue the equivalence in optimal point, the comparison table does not convince me as it may converge to only stationary point of the reformualtion, and the gradient distance bound or minimal value distance bound is not studied.
>
> **4. Theoretical analysis similar to [21]**
>
> Although [21] only consider non coupled constraint setting, its analysis technique in proving the bound for $\|\phi - \phi_{\delta}\|$ and related can be easily generalized to coupled constraint setting without major modification. So the novalty of the analysis in this paper is still unclear.
>
> **General comment**
>
> I think the reformulation this paper considered is very interesting. However, the major drawback is:
>
> 1) Sub-optimal presentation: too much related work not cited properly and some language use issues. The missing reference and missing acknowlege of existing theorem and lemma can have plagiarism concern.
>
> 2) The highlight of the paper, the equivalence of the proposed reformulation, was not supported with theoretical analysis quatitatively. e.g. How will the $O(\epsilon)$-solution of your reformualation be in distance with the solution of the original bilevel question? or how will the gradient of K in distance with the gradient of $\phi(x)$?
>
> Although I am impressed by the experiment result, I believe **major editing is required to get this paper ready for publish**. It is a promising paper if it can provide rigorous theory regarding 2.

---

> ### Author Response · Authors · 2025-08-01
>
> Thank you very much for recognizing the experimental results of our work and considering it promising. However, we find it necessary to make the following clarifications.
>
> >1. Setting and equivalence analysis of [16]
>
> We stated very clearly in the rebuttal: "We do acknowledge that [16] considers a more general setting, and we will highlight this in the revised paper to avoid misunderstandings". What we intended to clarify in the rebuttal is that the main focus of our paper, as well as [16], is on tackling the coupled constraints that are not easy to project. If the domain constraint $Y$ is a domain that is easy to project onto, then it can be handled via projection, which is not the main technical focus or analytical concern of our paper (or [16]).
>
> > 2. Comparison with [20]
>
> We have clearly introduced and discussed the main results of [20] and compared them with our work in Lines 116-124 in our original paper.  And we stated in the rebuttal that we will add [20] to the comparison table and add some explanations in the revised paper.
>
> The following is the content of Lines 116–124 of our original paper: "Recently, [20] also proposed first-order methods for linearly constrained BLO. Especially, they proved a nearly optimal convergence rate $\tilde{O}(\epsilon^{-2})$ for linear equality constraints and proposed algorithms that can attain $(\delta, \epsilon)$-Goldstein stationarity for linear inequality constraints. However, their convergence rates for linear inequality constraints either have additional dependence on dimension $d$ (such as $\tilde{O}(d\delta^{-1}\epsilon^{-3})$) or need additional assumptions to access the _exact_ optimal dual variable (such as $\tilde{O}(\delta^{-1}\epsilon^{-4})$), while we do not need the exact optimal dual variable assumption. Compared to the double-loop algorithms in [20], our proposed single-loop one is easier to implement in practice. Moreover, our techniques are also different from theirs for linear inequality constraints, thereby highlighting the distinct contributions and independent interests of our work."
>
> >I am not convinced with theoretical analysis.
>
> If the reviewer believes there is any error in our proof, we would be very grateful if you could point it out.
>
> >while this paper did not discuss in this or establishing distance bound for gradients between original problem and their final reformulation (4). ... 3. Equivalence problem. As the author only argue the equivalence in optimal point, the comparison table does not convince me as it may converge to only stationary point of the reformualtion, and the gradient distance bound or minimal value distance bound is not studied. ... The highlight of the paper, the equivalence of the proposed reformulation, was not supported with theoretical analysis quatitatively. e.g. How will the $O(\epsilon)$-solution of your reformualation be in distance with the solution of the original bilevel question? or how will the gradient of (4) in distance with the gradient of $\phi(x)$?
>
> **We stated very clearly in the original paper and in the rebuttal that in Theorems 5.2 and 5.4, as well as Corollaries 5.3 and 5.5, we directly established the convergence results for stationary points of $\Phi_\delta$, not the reformulation (4).** The gradient distance bound and the minimal value distance bound between $\Phi_\delta$ and $\Phi$ are rigorously discussed in Section 4.
>
> > although [16] did not explicitly discuss on $\|\phi(x)-\phi_\delta(x)\|$, this result can be easily achieved directly following the proofs of [21] ... 4. Theoretical analysis similar to [21]. Although [21] only consider non coupled constraint setting, its analysis technique in proving the bound for $\|\phi(x)-\phi_\delta(x)\|$ and related can be easily generalized to coupled constraint setting without major modification. So the novalty of the analysis in this paper is still unclear.
>
> While the upper bound for $|\Phi(x)-\Phi_\delta(x)|$ is not that difficult to derive, our work additionally proved the upper bound for $||\nabla \Phi(x)-\nabla \Phi_\delta(x)||$ with the coupled constraint setting. Compared to [21], our analysis is more complex due to the additional need to estimate the gradients of the Lagrange multipliers in the implicit gradient formula because of the coupled constraints. We do not see how this could have been easily generalized without major modification, as the reviewer has suggested.
>
> >The missing reference and missing acknowlege of existing theorem and lemma can have plagiarism concern.
>
> Actually, as we stated in the rebuttal, we have already cited and acknowledged these existing theorems and lemmas **in the Appendix of the original paper**. We will move these citations and acknowledgments to the main text in the revised paper.
>
> We hope the above responses address your concern.

---

> > ### Comment · Reviewer_K5dx · 2025-08-03
> >
> > Thank you. I think that resolved most of my concerns. I will raise my score.

---

### Official Review · Reviewer_Fppg · 2025-06-29

**Clarity:** 2
**Significance:** 3
**Originality:** 3
**Rating:** 4
**Confidence:** 3

**Summary:**

This paper considers bilevel optimization problems where the inner problem is linearly constrained. The inner function is assumed to be strongly convex, and the outer function is smooth, possibly non-convex. Additionally, the inner constraint set depends on both the inner and outer variables. First, the authors provide some differentiability results about the hyperobjective. Then, they propose SFLCB, a single-loop fully first-order algorithm to solve the considered problem. This algorithm is based on the Lagrangian reformulation of the bilevel problem. The authors demonstrate non-asymptotic convergence of SFLCB towards a stationary point of the hyperobjective. Numerical experiments on a toy problem, hyperparameter selection for SVM, and transportation network design are provided.

**Questions:**

- **Q1**: How is SFLCB sensitive to the hyperparameters?

**Ethical Concerns:**

["NO or VERY MINOR ethics concerns only"]

**Final Justification:**

The paper makes a solid contribution to the resolution of linearly constrained bilevel optimization problems. However, as noticed by the other reviewers, the clarity of the writing can be improved. For this reason, I keep my score at 4.

**Limitations:**

yes

**Quality:**

3

**Strengths And Weaknesses:**

### Strengths

- **S1**: To my knowledge, the proposed method is the first fully first-order algorithm that handles inner constraint set that depends on the outer variable $x$.

- **S2**: The proposed algorithm is a single-loop algorithm, making it more practical than a two-loop method. This is an improvement upon prior work.

- **S3**: SFLCB is theoretically grounded with non-asymptotic theoretical results.

- **S4**: Numerical experiments show the practical interest of SFLCB.



### Weaknesses

- **W1**: There is a substantial number of typos, imprecisions and notational issues that have to be fixed. See the section Minor/Typos for details.

- **W2**: I personally find the paper, and in particular Section 5 hard to read. In particular there are many notations, and readers can easily get lost. The introduction of $\alpha$ and $\beta$ in Section 5 should be motivated. Also, stating the SFLCB is an adaptation of gradient descent ascent on $K$ would be beneficial for a better understanding of the algorithm.

- **W3**: Maybe I missed something but in the proof of Theorem 5.2, it is unclear to me how from the different inequalities the authors conclude that the complexity is in $\mathcal{O}(\epsilon^{-3})$. Could the author clarify this point?

- **W4**: SFLCB comes with many hyperparameters (the five different step sizes and $\delta$) which makes it less practical.



### Minor/Typo
- **Line 23**: "upper level" -> "upper-level"

- **Line 24-25**: "$\mathcal{X} = \mathbb{R}^{d_x}, \mathcal{Y}(x) = \{y \in\mathbb{R}^{d_y}\,\vert\, h(x,y)\leq 0\}, h(x,y):\mathbb{R}^{d_x}\times\mathbb{R}^{d_y}\to\mathbb{R}^{d_h}$" -> Avoid sentences full of mathematical symbols without any accompanying text.

- **Line 33-34**: "However, these implicit gradient-based methods necessitate computing the Hessian matrix of the lower-level problem." -> This is inaccurate. Even though the implicit gradient involves a linear system with the Hessian matrix, this system can be solved by only using Hessian-vector products (for instance, with the CG algorithm [1, 2]), which do not require computing the full Hessian matrix [3, 4].

- **Assumption 3.2 and 4.6, and Lines 966 and 1077**: "Lipschitz smoothness" -> "Lipschitz continuous"

- **Assumption on $\delta$**: The assumption $\delta\leq \frac{\mu_g}{2l_{f,1}}$ should be stated from the beginning of section 4. Otherwise, $y_\delta$ and $\Phi_\delta$ are not necessarily well-defined. It is also missing in Theorem 4.1.

- **Convex/concave about**: Avoid writing that a function is "convex/concave/strongly convex *about* $y$" as done lines 208, 209, 210, 212. Instead, write that a function is "convex/concave/strongly convex *with respect to* $y$".

- **Lines 841-843**: There are several notational issues and typos:
    - "$\nabla_x g(x, y^* (x)) = \nabla_x g(x, y^* (x)) + \nabla_x h(x,y^* (x))\lambda^* (x)$" -> This is unclear, but if I refer to Lemma 2 in [5], I assume that $\nabla_x g(x, y^* (x))$ on the left-hand side refers to $\nabla_x(x\mapsto \min_{y\in\mathcal{Y}(x)}g(x,y))$ and $\nabla_x g(x, y^* (x))$  on the right-hand side refers to the gradient of $g$ with respect to its first argument at the point $(x, y^* (x))$. This has to be clarified in the notation.
    - "$\nabla_x g_\delta(x, y_\delta^* (x)) = \delta \nabla_x f(x, y_\delta(x)) + \nabla_x g(x, y_\delta^* (x)) + \nabla_x h(x,y^* (x))\lambda^* (x)$" -> Merely the same remark. $\nabla_x g_\delta(x, y_\delta^* (x))$ in reality denotes the gradient of $x\mapsto \delta f(x, y_\delta^* (x)) + g(x, y_\delta^* (x))$ which is unclear from the notation.
    - "$\Phi(x) = ...$" -> "$\Phi_\delta(x) = ...$"


[1] [Pedregosa, F. *Hyperparameter optimization with approximate gradient.* ICML 2016.](https://arxiv.org/pdf/1602.02355)

[2] [Grazzi, R., Franceschi, L., Pontil, M., and Salzo, S. *On the iteration complexity of hypergradient computation*. ICML 2020.](https://arxiv.org/pdf/2006.16218)

[3] [Pearlmutter, B. A. *Fast exact multiplication by the hessian*. Neural computation, 1994](https://www.bcl.hamilton.ie/~barak/papers/nc-hessian.pdf)

[4] [Dagréou, M., Ablin, P., Vaiter, S., and Moreau, T. *How to compute hessian-vector products?* ICLR Blogposts 2024.](https://iclr-blogposts.github.io/2024/blog/bench-hvp/)

[5] [Jiang, L., Xiao, Q., Tenorio, V. M, Real-Rojas, F. Marques, A. and Chen, T. *A primal-dual-assisted penalty approach to bilevel optimization with coupled constraints*. arXiv:2406.10148 2024](https://arxiv.org/pdf/2406.10148)

---

> ### Author Rebuttal · Authors · 2025-07-31
>
> We thank the reviewer for the valuable comments. Please see our responses below.
>
> >I personally find the paper, and in particular Section 5 hard to read. In particular there are many notations, and readers can easily get lost. The introduction of $\alpha$ and $\beta$ in Section 5 should be motivated. Also, stating the SFLCB is an adaptation of gradient descent ascent on $K$ would be beneficial for a better understanding of the algorithm.
>
> Thanks for your advice. The introduction of $\alpha$ and $\beta$ is primarily for technical reasons. We reformulate the original constraints $h(x, y) \leq 0$ and $h(x, z) \leq 0$ as  $h(x, y) - \alpha = 0, \alpha \leq 0; h(x, z) - \beta = 0, \beta \leq 0$ respectively.  This reformulation allows us to handle the equality constraints $h(x, y) - \alpha = 0$ and $h(x, z) - \beta = 0$ using the augmented Lagrangian $K$, while the inequality constraints $\alpha \leq 0$ and $\beta \leq 0$ can be enforced via simple projection. We will add these explanations and state that the SFLCB is an adaptation of gradient descent ascent on $K$ in the revised paper to make the derivation and the motivations of SFLCB clearer.
>
> >Maybe I missed something but in the proof of Theorem 5.2, it is unclear to me how from the different inequalities the authors conclude that the complexity is in $O(\epsilon^{-3})$. Could the author clarify this point?
>
> Thanks for your question. Our theoretical analysis establishes the following result: $$
> \frac{1}{T} \sum_{t=0}^{T-1} || \nabla \phi_\delta(x_t) ||^2 \leq \frac{4}{T \eta_x} \left( V_0 - \min_t V_t \right)
> $$
> (line 959, 1069). Moreover, we prove that
> $$
> V_t \geq V_{\min} \geq \frac{5\delta \Phi^*}{4} - \frac{5\delta^2 l^2_{f,0}}{8\mu_g}
> $$
> (line 960, 1070).
>
> Since $\delta = \Theta(\epsilon)$ and $\phi_\delta = \delta \Phi_\delta$, an $\epsilon^2$-stationary point of $\phi_\delta$ is correspondingly an $\epsilon$-stationary point of $\Phi_\delta$. Therefore, we can find an $\epsilon$-stationary point of $\Phi_\delta$ with a complexity of $O(\epsilon^{-4})$. Note that $-V_t \leq O(\delta) = O(\epsilon)$.
>
> We additionally prove that with good $y'\_0$, $z'\_0$, $u_0$, $v_0$ (satisfying line 259–260 or line 273–274), we have
> $$
> V_0 \leq \frac{5\delta \Phi(x_0)}{4} + O(\delta) = O(\epsilon)
> $$
> (line 963, 1074).
>
> Therefore, with good $y'\_0$, $z'\_0$, $u_0$, $v_0$ (satisfying line 259–260 or line 273–274), SFLCB can find an $\epsilon$-stationary point of $\Phi_\delta$ with a complexity of $O(\epsilon^{-3})$. We will make the proof of Theorem 5.2 clearer in the revised paper.
>
> >SFLCB comes with many hyperparameters (the five different step sizes and $\delta$) which makes it less practical. How is SFLCB sensitive to the hyperparameters?
>
> We note that related works also typically require a similar number of hyperparameters. For example, BLOCC [16] similarly requires 5 step sizes for their $x$, $y_g$, $y_F$, $\mu_g$, $\mu_F$ and one hyperparameter $\lambda$, which plays a similar role as $\delta$ in our work. BiC-GAFFA [48] needs step sizes $\alpha$, $\eta$, proximal parameters $\gamma_1, \gamma_2$, and a penalty parameter $c_k$. LV-HBA [49] needs step sizes $\alpha$, $\beta$, $\eta$, a proximal parameter $\gamma$, and a penalty parameter $c_k$. In Appendix F.2.1, we conducted sensitivity experiments of $\delta$ for transportation network design and found that our algorithm performs well over a wide range of $\delta$ values (from 0.05 to 0.5). In our hyperparameter optimization experiments for SVM and transportation network design, we simply set all five step sizes ($\eta_x$, $\eta_y$, $\eta_z$, $\eta_u$, $\eta_v$) to the same value and still observed reasonably good performance. These experimental results indicate that SFLCB is not particularly sensitive to the choice of hyperparameters, and that good performance can be achieved with simple tuning of $\delta$ and the learning rate. We would like to add more sensitivity analysis experiments in the revised paper to better illustrate this point.
>
> >Minor/Typo: Line 23; Line 24-25; Assumption 3.2 and 4.6, and Lines 966 and 1077; Assumption on $\delta$; Convex/concave about
>
> Thanks for your advice. We will modify these typos in the revised paper.
>
> >Minor/Typo: Line 33-34: "However, these implicit gradient-based methods necessitate computing the Hessian matrix of the lower-level problem." -> This is inaccurate. Even though the implicit gradient involves a linear system with the Hessian matrix, this system can be solved by only using Hessian-vector products (for instance, with the CG algorithm [1, 2]), which do not require computing the full Hessian matrix [3, 4].
>
> Thanks for your advice. The phrase “these implicit gradient-based methods” in line 33 is intended to refer to the works mentioned in the previous sentence, namely [41, 18, 45, 44], which all use implicit gradient-based methods to handle constrained bilevel problems and involve computing the Hessian matrix. We will revise the paper to clarify this and avoid potential misunderstandings.
>
> >Minor/Typo: Lines 841-843: There are several notational issues and typos
>
> Thanks for pointing out these issues and typos. Actually, we will define $\psi(x) = g(x, y^{\ast}(x))$ and $\psi\_\delta(x) = g\_\delta(x, y^{\ast}\_\delta(x))$.
>
> Line 841 should be:
> $$
> \nabla_x \psi(x) = \nabla_x g(x, y^{\ast}(x)) + \nabla_x h(x, y^{\ast}(x)) \lambda^{\ast}(x)
> $$
>
> and
> $$
> \nabla_x \psi\_\delta(x) = \delta \nabla_x f(x, y^{\ast}\_\delta(x)) + \nabla_x g(x, y^{\ast}\_\delta(x)) + \nabla_x h(x, y^{\ast}\_\delta(x)) \lambda^{\ast}\_\delta(x)
> $$
>
> Line 842 should be:
> $$
> \Phi\_\delta(x) = f(x, y^{\ast}\_\delta(x)) + \frac{1}{\delta} \left[ g(x, y^{\ast}\_\delta(x)) - g(x, y^{\ast}(x)) \right]
> $$
>
> We will modify these in the revised paper.

---

> > ### Comment · Reviewer_Fppg · 2025-08-04
> >
> > I thank the authors for their answer. I keep my positive score.

---

### Official Review · Reviewer_ME2c · 2025-07-03

**Clarity:** 2
**Significance:** 3
**Originality:** 3
**Rating:** 4
**Confidence:** 2

**Summary:**

- This paper proposes a first-order method for bilevel optimization problems where the lower-level problem has a strongly convex objective function and linear constraints.
- The authors first reformulate the bilevel optimization problem as a single-level problem. They provide non-asymptotic bounds on the differences in function values and gradients between the original hyper-objective function and the reformulated one.
- They then construct a convex augmented Lagrangian function under linear constraints and design a Hessian-free first-order method based on this augmented Lagrangian.
- The paper analyzes the convergence of the proposed algorithm, demonstrating an improved convergence rate compared to existing Hessian-free single-loop methods.

**Questions:**

- Regarding the second weakness I raised: Is the observed speedup directly attributable to the improved convergence rate? I am also interested in whether the convergence of the objective function value improves with the number of iterations.
- Related to the first question, I am interested in the arithmetic computational cost per iteration of the proposed method. Is the arithmetic computational cost of the proposed method the same as that of existing single-loop methods?
- The authors mention that prior work [49] requires projection onto $\mathcal C$, which incurs high computational cost. However, in cases like this paper, where $h$ is a linear function of $x$ and $y$, projection is easily computable. Does the proposed method still offer a computational cost advantage over [49] even when $h$ is a linear function?

**Ethical Concerns:**

["NO or VERY MINOR ethics concerns only"]

**Final Justification:**

The authors have sincerely addressed all of my concerns and questions. While I recommend they revise the manuscript to incorporate these points, I believe the paper is suitable for publication.

**Limitations:**

yes

**Quality:**

2

**Strengths And Weaknesses:**

- Strengths
	- Bilevel optimization problems have diverse applications in the machine learning community, making the design of efficient solution methods for these problems highly significant.
	- In particular, existing single-loop methods could only discuss the asymptotic relationship between the reformulated problem and the original bilevel optimization problem. This work offers originality by providing a non-asymptotic analysis.
	- Furthermore, despite the limitation to linear constraints, the paper achieves an improved convergence rate, which makes it a valuable contribution to research on solution methods for bilevel optimization problems.
- Weaknesses
	- The clarity of Section 4 could be improved. Section 4 discusses the reformulation in a general case where the constraint function $h$ is not necessarily limited to being linear. This section should state more explicitly that $h$ is not limited to a linear function.
	- Regarding the computational experiments: While it's true that the proposed method obtains higher quality solutions in less time compared to existing single-loop methods, it's unclear if this should be directly attributed to the improved convergence rate.

---

> ### Author Rebuttal · Authors · 2025-07-30
>
> We thank the reviewer for the valuable comments. Please see our responses below.
>
> >The clarity of Section 4 could be improved. Section 4 discusses the reformulation in a general case where the constraint function  is not necessarily limited to being linear. This section should state more explicitly that  is not limited to a linear function.
>
> Thanks for your advice. We will state these more explicitly in the revised paper.
>
> >Regarding the computational experiments: While it's true that the proposed method obtains higher quality solutions in less time compared to existing single-loop methods, it's unclear if this should be directly attributed to the improved convergence rate. Regarding the second weakness I raised: Is the observed speedup directly attributable to the improved convergence rate? I am also interested in whether the convergence of the objective function value improves with the number of iterations.
>
> In transportation network design experiment, the observed speedup is attributable to the improved convergence rate and objective function value improves with the number of iterations. Actually, in the transportation network design problem, the UL utility is defined as the negative of the objective function value. Thus, a larger UL utility means a smaller objective function value. We provide the following results showing how the objective function value changes over time with BLOCC and SFLCB.
>
> 3 Node:
> | Time (s) | 1    | 2    | 3    | 4    | 5    |
> |----------|------|------|------|------|------|
> | BLOCC    | 1.36 | 1.66 | 1.70 | 1.70 | 1.71 |
> | SFLCB    | 1.04 | 1.69 | 1.84 | 1.84 | 1.84 |
>
> 9 Node:
> | Time (s) | 20     | 40     | 60     | 80     | 100    | 120   | 140   | 160   | 180   | 200   | 250   | 300   | 350   | 400   |
> |----------|--------|--------|--------|--------|--------|-------|-------|-------|-------|-------|-------|-------|-------|-------|
> | BLOCC    | -420.87| -89.73 | -23.66 | -4.62  | 1.72   | 6.25  | 6.59  | 8.55  | 8.93  | 9.14  | 9.53  | 9.15  | 9.52  | 9.54  |
> | SFLCB    | 2.57   | 12.09  | 15.30   | 18.03  | 19.95  | 20.52 | 20.92 | 21.18 | 21.19 | 21.20 | 21.06 | 20.79 | 20.67 | 20.40  |
>
> In the 9-node setting, we set $\delta=0.25$ in SFLCB for better comparison with BLOCC (since in our reformulation, $\delta$ has a similar role as $1/\gamma$ in BLOCC and $\gamma=4$) and keep all other hyperparameters the same as those in our current paper. Note that the $\delta$ here is different from the one used in our paper. Thus, the results of the 9-node setting are different from the current paper.
>
> We can see that in both the 3-node and 9-node settings, SFLCB converges faster than BLOCC and achieves better UL utility (i.e., smaller objective function value). We will add performance figures in the revised paper to show how the objective function value changes over time and the number of iterations to better illustrate the practical efficiency of our method.
>
> >Related to the first question, I am interested in the arithmetic computational cost per iteration of the proposed method. Is the arithmetic computational cost of the proposed method the same as that of existing single-loop methods?
>
> Yes, the arithmetic computational cost per iteration of our proposed method is at least the same as, if not better than, the existing single-loop methods. For example, let us adopt the same assumptions that were used in [16] which assumes the complexity of calculating a function is proportional to the dimension of the inputs. In this case, assume the complexity for $\nabla_x f$, $\nabla_x g$ are $O(d_x)$, and the complexity for $\nabla_y f$, $\nabla_x g$ are $O(d_y)$. Then, the arithmetic computational cost per iteration of our method is $O(d_xd_h+d_yd_h)$. The arithmetic computational cost per iteration of BiC-GAFFA [48] is also $O(d_xd_h+d_yd_h)$, which is the same as ours. As a second example, LV-HBA [49] requires projection onto $\{h(x,y)=Bx+Ay-b\leq 0\}$ in each iteration. This projection problem is a Quadratic Programming (QP) problem with linear inequality constraints, which generally does not have a closed-form solution and common methods used to solve this problem, such as interior-point methods (Nesterov, 1994), require a complexity of $O(p(d_h, d_x, d_y)\log(1/\epsilon))$ to obtain an $\epsilon$ accurate solution, where $p(\cdot)$ is some polynomial. Thus, the computational cost per iteration of LV-HBA is $O(d_xd_h+d_yd_h +p(d_h, d_x, d_y)\log(1/\epsilon))$, which is larger than ours. As a third example, BLOCC [16] for linear constraints has a double-loop structure and its arithmetic computational cost per outer-loop iteration is $O((d_xd_h+d_yd_h)\log(1/\epsilon))$, which is also larger than ours.
>
> >The authors mention that prior work [49] requires projection onto $C$, which incurs high computational cost. However, in cases like this paper, where $h$ is a linear function of $x$ and $y$, projection is easily computable. Does the proposed method still offer a computational cost advantage over [49] even when $h$ is a linear function?
>
> Yes, even when $h$ is a linear function, the projection to $\{h(x,y)=Bx+Ay-b\leq 0\}$ is a Quadratic Programming (QP) problem with linear inequality constraints, which generally does not have a closed-form solution and commonly needs iterative methods to solve. Common methods used to solve this problem, such as interior-point methods (Nesterov, 1994), require a complexity of $O(p(d_h, d_x, d_y)\log(1/\epsilon))$ to obtain an $\epsilon$ accurate solution, where $p(\cdot)$ is some polynomial. Thus, our method still offer a computational cost advantage over [49].
>
> __Additional References__:
>
> Nesterov, Yurii, and Arkadii Nemirovskii. Interior-point polynomial algorithms in convex programming. Society for industrial and applied mathematics, 1994.

---

> > ### Comment · Reviewer_ME2c · 2025-08-04
> >
> > Thank you for your rebuttal. The authors' responses have successfully addressed my concerns. I will maintain my positive evaluation score.

---

### Official Review · Reviewer_gHwN · 2025-07-03

**Clarity:** 3
**Significance:** 2
**Originality:** 2
**Rating:** 4
**Confidence:** 4

**Summary:**

This paper studies bilevel optimization problems where the lower-level objective is strongly convex and subject to general linear constraints, including both inequalities and equalities that may couple with upper-level variables. The authors first apply a penalty relaxation and derive sufficient conditions under which the relaxed problem remains approximately equivalent to the original formulation. They then introduce a surrogate strongly convex loss function for the relaxed (but still constrained) problem. Finally, they propose an algorithm, SFLCB, to optimize the surrogate loss, analyze its convergence, and relate the resulting solution back to an approximate stationary point of the penalized problem.

**Questions:**

See Weaknesses

**Ethical Concerns:**

["NO or VERY MINOR ethics concerns only"]

**Final Justification:**

Still somewhat borderline, but a technically correct paper.

**Limitations:**

Overall, this paper feels borderline, but it addresses a relevant problem for the bilevel optimization community. Sharpening the emphasis on novelty and highlighting more surprising or non-trivial insights would help improve readability.

**Quality:**

3

**Strengths And Weaknesses:**

**Strengths**

-	The writing is generally clear and the contribution is positioned well in the context of existing work.

-	The paper builds on prior landscape analysis results and offers a nontrivial extension.

-	Convergence guarantees are provided for the proposed algorithm.


**Weaknesses**

-	The landscape analysis appears more general than previous results, but it primarily builds on [21], and the novelty seems limited.

-	The transition from Section 4 to Section 5 feels abrupt. The restriction to linear constraints in the algorithmic part needs more justification.

-	The sample complexity discussion is somewhat unclear. In particular:

1)	If $\delta = O(\epsilon)$, then we are already close to a stationary point — calling this a “good initialization” seems misleading.

2)	In many settings, initialization quality does not affect the final convergence rate. Where does the difference come from?

-	The claimed $\log(1/\epsilon)$ improvement over [16] needs clarification. Is this a fundamental improvement, or could a sharper analysis of [16] yield the same rate?

---

> ### Author Rebuttal · Authors · 2025-07-30
>
> We thank the reviewer for the valuable comments. However, we believe that the reviewer has some misunderstandings about the initialization and the sample complexity discussed in our paper. Actually, the $O(\epsilon^{-3})$ convergence rate in our paper does _not_ depend on the choice of initial point $x_0$. One can randomly choose $x_0\in X$ and set $\delta=\Theta(\epsilon)$ (Note that setting some constants to be $O(\epsilon)$ (or $O(\epsilon^{-1})$) is commonly used in related works [16, 20]). Then, according to Corollary 5.3 and Corollary 5.5, one can easily find $y'_0, z'_0, u_0, v_0$ that satisfy line 259-260 (or line 273-274) with a complexity of $O(\log(1/\epsilon))$ (or $O(\epsilon^{-2})$). Then with these $\delta, x\_0, y'_0, z'_0, u_0, v_0$, our SFLCB algorithm can converge with a complexity of $O(\epsilon^{-3})$. Therefore, with random initial point $x_0$, the overall complexity is $O(\epsilon^{-3}+\log(1/\epsilon))=O(\epsilon^{-3})$ (or $O(\epsilon^{-3}+\epsilon^{-2})=O(\epsilon^{-3})$). We hope these explanations clarify these misunderstandings. Below, we provide more detailed responses.
>
> >The landscape analysis appears more general than previous results, but it primarily builds on [21], and the novelty seems limited.
>
> The analysis in [21] only considers lower-level constraints  $y\in Y$ that are independent of $x$, whereas our work analyzes _coupled constraints_ of the form $y\in Y(x)=$ { $y\in R^{d_y} | h(x,y)\leq 0 $ }. Handling such coupled constraints requires estimating the Lagrangian multipliers in the implicit gradient formula, which makes our analysis more involved compared to [21]. Notably, the Future Directions section of [21] also emphasizes that analyzing lower-level $x$-dependent constraints is a valuable direction for future research, highlighting the value of our work. Therefore, we believe our work offers both novelty and significance.
>
> >The transition from Section 4 to Section 5 feels abrupt. The restriction to linear constraints in the algorithmic part needs more justification.
>
> The focus of our paper is on linearly constrained bilevel optimization, and therefore Section 5 primarily analyzes the linear constraints. It is worth noting that linear constraints encompass a broad range of applications, including distributed optimization [47, 18], adversarial training [52, 18], and hyperparameter optimization for constrained learning tasks such as hyperparameter tuning in SVMs. Moreover, several related works also concentrate on analyzing the linear constraints [41, 18, 20]. We appreciate your advice, and will add more justifications in the revised paper to make the transition smoother.
>
> >If $\delta=O(\epsilon)$, then we are already close to a stationary point — calling this a “good initialization” seems misleading.
>
> Setting $\delta=\Theta(\epsilon)$ does not suggest that we are close to a stationary point. Setting some constants to be $O(\epsilon)$ (or $O(\epsilon^{-1})$) is commonly used in related works [16, 20]. In our paper, our goal is to find an $\epsilon$ stationary point $\hat x$ satisfies $||\Phi_\delta(\hat x)||\leq \epsilon$. However, initial $x_0\in X$ can be randomly selected, and there is no guarantee that it will be close to a stationary point. In this paper, we say $y'_0, z'_0, u_0, v_0$ are "good initializations" if they satisfy line 259-260 (or line 273-274). Actually, we prove that for any fixed $\delta=\Theta(\epsilon)$ and $x_0\in X$, these "good initializations" are not difficult to achieve. In Corollary 5.3, we prove that for any initial point $x_0$, for $Bx + Ay-b$ constraints and $A$ is full row rank, we can find "good initializations" with a complexity of $O(\log(1/\epsilon))$ and in Corollary 5.5, we prove that for any initial point $x_0$, for $Ay-b$ constraints, we can find "good initializations" with a complexity of $O(\epsilon^{-2})$. Thus, the overall complexity is $O(\epsilon^{-3}+\log(1/\epsilon))=O(\epsilon^{-3})$ (or $O(\epsilon^{-3}+\epsilon^{-2})=O(\epsilon^{-3})$).
>
> >In many settings, initialization quality does not affect the final convergence rate. Where does the difference come from?
>
> Actually, the initial point $x_0$ does not affect the final convergence rate in our algorithm. Please refer to our general response before. We further add that the main reason why $y'\_0$, $z'\_0$, $u\_0$, $v\_0$ affect the convergence rate is that our theoretical analysis establishes the following result:
> $$
> \frac{1}{T} \sum_{t=0}^{T-1} || \nabla \phi_\delta(x_t) ||^2 \leq \frac{4}{T \eta_x} \left( V_0 - \min_t V_t \right)
> $$
> (line 959, 1069). Moreover, we prove that
> $$
> V_t \geq V_{\min} \geq \frac{5\delta \Phi^*}{4} - \frac{5\delta^2 l^2_{f,0}}{8\mu_g}
> $$
> (line 960, 1070).
>
> Since $\delta = \Theta(\epsilon)$ and $\phi_\delta = \delta \Phi_\delta$, an $\epsilon^2$-stationary point of $\phi_\delta$ is correspondingly an $\epsilon$-stationary point of $\Phi_\delta$. Therefore, we can find an $\epsilon$-stationary point of $\Phi_\delta$ with a complexity of $O(\epsilon^{-4})$. Note that $-V_t \leq O(\delta) = O(\epsilon)$.
>
> We additionally prove that with good $y'\_0$, $z'\_0$, $u_0$, $v_0$ (satisfying line 259–260 or line 273–274), we have
> $$
> V_0 \leq \frac{5\delta \Phi(x_0)}{4} + O(\delta) = O(\epsilon)
> $$
> (line 963, 1074).
>
> Therefore, with good $y'\_0$, $z'\_0$, $u_0$, $v_0$ (satisfying line 259–260 or line 273–274), SFLCB can find an $\epsilon$-stationary point of $\Phi_\delta$ with a complexity of $O(\epsilon^{-3})$.
>
> >The claimed $\log(1/\epsilon)$ improvement over [16] needs clarification. Is this a fundamental improvement, or could a sharper analysis of [16] yield the same rate?
>
> Compared with [16], the $\log(1/\epsilon)$ improvement achieved by our algorithm is a fundamental advancement. The additional $\log(1/\epsilon)$ complexity in [16] arises from their double-loop structure. Specifically, their inner loop involves solving a constrained strongly convex minimization problem, which inevitably incurs an extra $\log(1/\epsilon)$ complexity. In contrast, our algorithm adopts a single-loop structure, which theoretically avoids this additional overhead, leading to better convergence complexity. Moreover, single-loop algorithms are generally easier to implement in practice, and our experiments further demonstrate that our method empirically outperforms [16]. We believe that one of the main contributions of our paper compared to [16] is the improvement from a double-loop to a single-loop algorithm. In the field of minimax optimization -- a related area closely connected to bilevel optimization -- such improvements from double-loop to single-loop (Zhang 2020; Xu, 2023) are also commonly regarded as significant contributions.
>
> __Additional References__:
>
> Zhang, Jiawei, et al. "A single-loop smoothed gradient descent-ascent algorithm for nonconvex-concave min-max problems." Advances in neural information processing systems (2020).
>
> Xu, Zi, et al. "A unified single-loop alternating gradient projection algorithm for nonconvex–concave and convex–nonconcave minimax problems." Mathematical Programming (2023).

---

> ### Author Response · Authors · 2025-08-07
> **Follow-up on Rebuttal Response**
>
> Dear Reviewer gHwN,
>
> As the rebuttal phase is nearing its end (with less than two days remaining), we would like to kindly follow up regarding our responses to your comments. We would greatly appreciate it if you could let us know whether our reply has addressed your concerns adequately, or if there are any remaining questions or clarifications you'd like us to provide in the limited time remaining.
>
> Thank you again for your time and effort in reviewing our work.
>
> Best regards,
>
> Authors of Submission 18593

---

### Note · Authors · 2025-08-13

Dear Reviewers and Area Chairs,

We sincerely thank all reviewers for the valuable comments and discussions. In particular, we are grateful to Reviewers ME2c, Fppg, and K5dx for acknowledging that we have addressed most of their concerns and for their positive feedback. Regarding Reviewer gHwN’s concern about the sample complexity and initialization, we have clarified in the rebuttal that the $O(\epsilon^{-3})$ convergence rate in our paper does not depend on the choice of the initial point $x_0$. One can randomly choose $x_0 \in X$, and then, following the procedures in Corollary 5.3 and Corollary 5.5, an $\epsilon$-stationary point can be found with an overall complexity of $O(\epsilon^{-3})$. Compared to [16], in the linearly constrained setting, our algorithm achieves an improvement from a double-loop to a single-loop structure and improves the complexity from $O(\epsilon^{-3}\log(1/\epsilon))$ to $O(\epsilon^{-3})$. Single-loop algorithms are easier to implement in practice, and our experiments also demonstrate the empirical efficiency of our method. In the field of minimax optimization—a related area closely connected to bilevel optimization—such improvements from double-loop to single-loop (Zhang, 2020; Xu, 2023) are also widely recognized as significant contributions.

We also appreciate the reviewers’ feedback on minor typos and improvement suggestions, and we will incorporate these changes in the revised paper. Thanks again for your time and effort in reviewing our work.

Additional References:

Zhang, Jiawei, et al. "A single-loop smoothed gradient descent-ascent algorithm for nonconvex-concave min-max problems." Advances in neural information processing systems (2020).

Xu, Zi, et al. "A unified single-loop alternating gradient projection algorithm for nonconvex–concave and convex–nonconcave minimax problems." Mathematical Programming (2023).

Best regards,

Authors of Submission 18593

---

### Decision · Program_Chairs · 2025-09-17

**Decision:**

Accept (poster)

**Comment:**

Based on my reading the reviews, author responses and discussion, I see the following strengths of this manuscript:

- **S1.** The manuscript presents a reformulation of a bilevel optimization problem with coupled lower level linear constraints, converting it to a single level problem using the value-function based formulation. Based on this, the paper proposes a first-order Hessian-free optimization algorithm, and studies the convergence of the reformulated problem. Furthermore, the difference between the solution of the reformulated problem and the original bilevel problem is also bounded.

- **S2.** The convergence rate provides improvement over existing double loop solvers, highlighting the benefit of the single loop solution.


I also see the following limitations of this manuscript:

- **L1.** The presentation of the manuscript is quite confusing in parts and requires significant revision to better highlight the novel contributions of this work. On a related note, as highlighted by reviewers, the authors need to better position the contributions with respect to existing literature. Overall, it seems that the manuscript would need a significant revision.

- **L2.** (minor) The actual algorithm requires multiple hyperparameters to tune though that seems to be standard in bilevel optimization literature. Furthermore, the authors shared some additional results in the rebuttal claiming that the proposed scheme is not sensitive to the choices of the hyperparameters.

Based on the author-reviewer discussion, it seems like all reviewers agree that their concerns have been addressed. However, this manuscript needs some revision with better presentation of contributions and better positioning against existing literature.

I am recommending an accept but I hope that the authors incorporate all the feedback provided by the reviewers.